# Least Squares Regression Can Exhibit Under-Parameterized Double Descent

**Xinyue Li**
Applied Math, Yale University
xinyue.li.xl728@yale.edu

**Rishi Sonthalia**
Math, Boston College
rishi.sonthalia@bc.edu

## Abstract

The relationship between the number of training data points, the number of parameters, and the generalization capabilities of models has been widely studied. Previous work has shown that double descent can occur in the over-parameterized regime and that the standard bias-variance trade-off holds in the under-parameterized regime. These works provide multiple reasons for the existence of the peak. We postulate that the location of the peak depends on the technical properties of both the spectrum as well as the eigenvectors of the sample covariance. We present two simple examples that provably exhibit double descent in the under-parameterized regime and do not seem to occur for reasons provided in prior work.

## 1 Introduction

This paper demonstrates interesting new phenomena that suggest that our understanding of the relationship between the number of data points, the number of parameters, and the generalization error is incomplete, even for simple linear models. The classical bias-variance theory postulates that the generalization risk versus the number of parameters for a fixed number of training data points is U-shaped (Figure 1a[1]). However, modern machine learning has shown that if we keep increasing the number of parameters, the generalization error eventually starts decreasing again [2, 3] (Figure 1b[2]). This second descent has been termed as *double descent* and occurs in the *over-parameterized regime*, which is when the number of parameters exceeds the number of data points. *Understanding the location and the cause of such peaks in the generalization error is of significant importance.*

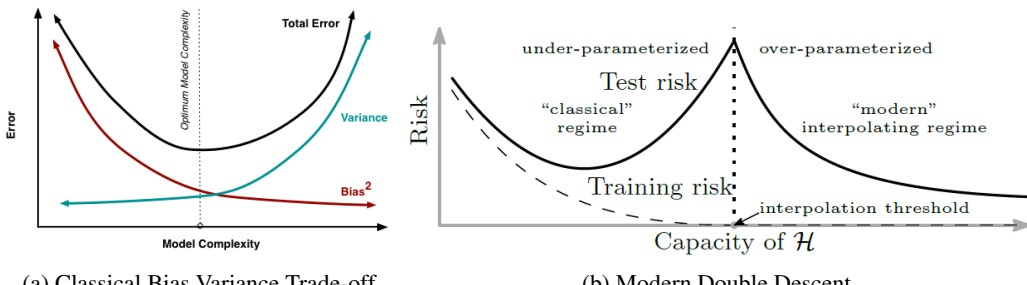

(a) Classical Bias Variance Trade-off.      (b) Modern Double Descent.

Figure 1: Bias-variance trade-off and double descent.

Many different theories have been postulated for the appearance of the peak. The prevalent theory is that when the model is under-parameterized, the learning is constrained. This constraint on the

---

[1]Image source [1]
[2]Image source [3]

38th Conference on Neural Information Processing Systems (NeurIPS 2024).

learning results in increased variance until the interpolation point. After this point, there exists a high dimensional space of solutions, and learning methods, such as gradient descent, pick solutions that generalize well. This conjecture has been empirically validated for deep neural networks. Due to the challenges of analyzing deep neural networks, theoretical understanding of this phenomenon has focused on linear models - linear regression [4–13] and kernelized regression [14–22]. These works show that there exists a peak at the *boundary between the under and over-parameterized regimes*. Hence validating the above postulated theory for their setting. Careful theoretical analysis then shows that the generalization error can be decomposed into various terms, one of which is the norm of the estimator. Specifically, it has been shown that the curve for the norm of the estimator versus the dimension of the data also exhibits double descent, with the peak occurring at the same point as the peak in the generalization error curve. In most cases, this is the only term in the decomposition that exhibits double descent. This leads to a second theory for the occurrence of the peak, that is, *the peak in the generalization error for linear models occurs due to the norm of the estimator blowing up.*

**Contributions.** Since understanding the reasons that peaks occur is of critical importance, it is crucial that we have a robust theory for their appearance. However, most work focuses on the over-parameterized regime and ignores the under-parameterized regime. This is because it is commonly believed that the variance is monotonic in the under-parameterized regime. We show that this is not true and present two simple examples that exhibit double descent in the *under-parameterized regime.*

- **Why does the Peak Occur**. We argue that the location of the peak depends on two factors: the alignment between the targets $y$ and singular vectors $V$ of the training data matrix and the spectrum of the data. We show that by modifying these quantities appropriately, we can move the peak into the underparameterized regime.
- **Modifying the Alignment**. For the first example, we consider a spiked covariate model, where one eigendirection dominates, and the regression target only depends on the dominant eigendirection. For this model, we consider the ridge regularized problem with ridge parameter $\mu^2$ and show that the ridge parameter $\mu^2$ controls the alignment between the targets $y$ and the singular vectors $V$. We show that for $\mu > 0$ the peak occurs in the under-parameterized regime (Theorem 2). Specifically, when the ratio of the dimension to the number of training data points $c$ is equal to $(1 + \mu^2)^{-1}$ ($c := d/n = 1/(1 + \mu^2)$).
- **Modifying the Spectrum** For the second example, we consider training data that is a mixture of isotropic Gaussian vectors and vectors from along a fixed direction $z$. By varying the mixture proportions $(\pi_1, \pi_2)$, we can modify the spectrum of the covariance matrix. We show that the expected risk displays under-parameterized double descent (Theorem 4), with the peak occurring when $c := d/n = \pi_1$).
- **Norm of the estimator**. We further analyze the first example and show that if we fix the number of training data points $n$ and vary the dimension $d$ of the problem, then for large values of $\mu$, the risk curve does not display a double descent. However, the curve for the norm of the estimator does display descent. Thus, the peak in the norm of the estimator does not imply a peak in the generaliation error.

**Organization.** The rest of the paper is organized as follows. Section 2 presents a quick overview of prior work on double descent for linear models. Section 3 highlights two less-studied properties that influence the location of the peak. Section 4 presents the first of the two examples of under-parameterized double descent. This model also shows that a double descent in the norm of the estimator does not translate to a double descent in the risk. Finally, Section 5 presents the second example of under-parameterized double descent.

## 2 Prior Work on Double Descent

In this section, we present the current prevailing theories for the occurrence of local maximums in the risk curve. Concretely, consider the following simple linear model that is a special case of the general models studied in [5, 8, 11, 23] amongst many other works. Let $x_i \sim \mathcal{N}(0, I_d)$ and let $\beta \in \mathbb{R}^d$ be a linear model with $\|\beta\| = 1$. Let $y_i = \beta^T x_i + \xi_i$ where $\xi \sim \mathcal{N}(0, 1)$. Then, let $\beta_{opt}$ be the minimum norm solution to $\arg\min_{\tilde{\beta}} \|\beta^T X_{trn} - \tilde{\beta}^T X_{trn} + \xi_{trn}\|$, where $\xi_{trn} \in \mathbb{R}^{n \times 1}$. One important quantity is the aspect ratio of $X$. Specifically, for a $d \times n$ matrix, the aspect ratio is $c := d/n$. With this terminology, we see that a model is under-parameterized if $c < 1$ and over-parameterized when $c > 1$.

Table 1: Table showing various assumptions on the data and the location of the double descent peak for linear regression and denoising. We only present a subset of references for each problem setting. For the low rank setting in this paper, see Appendix F.

| Noise | Ridge Reg. | Dim. | Peak Location | Reference |
|---|---|---|---|---|
| Input | Yes | 1 | Under-parameterized | This paper. |
| Output | No | Full | Under-parameterized | This paper |
| Input | No | Low | Interpolation point | [24, 37] |
| Input | Yes | Full | Interpolation point | [38] |
| Output | No | Full | Over-parameterized/interpolation point | [5, 8, 11] |
| Output | Yes | Full | Over-parameterized/interpolation point | [11, 23] |
| Output | No | Low | Over-parameterized/interpolation point | [39] |
| Output | Yes | Low | Over-parameterized/interpolation point | [40] |
| Output | No | Low | No peak | [41] |

Finally, the interpolation point, i.e., the point at which we can exactly fit the training data, is $c = 1$, assuming we have full-rank data.

Then, the excess risk $\mathcal{R}$ and the expected norm of $\beta_{opt}$ can be expressed as follows:

$$\mathcal{R} = \begin{cases} \frac{c}{1-c} & c < 1 \\ \frac{c-1}{c} + \frac{1}{c-1} & c > 1 \end{cases} \quad \text{and} \quad \beta_{opt} = \begin{cases} 1 + \frac{c}{1-c} & c < 1 \\ \frac{1}{c} + \frac{1}{c-1} & c > 1 \end{cases}.$$

In this model, there are a few important features that are ubiquitous in many prior double descent studies for linear models:

1. The peak happens at $c = 1$, on the border between the under and over-parameterized regimes.
2. Further, at $c = 1$ the training error equals zero. Hence, this is the interpolation point.
3. The peak occurs due to the expected norm of the estimator $\beta_{opt}$ blowing up near the interpolation point.

This is further validated by works that study ridge regularized regression [23–26]. Works such as [23] have shown that optimally regularized regression no longer exhibits double descent. Further, increasing the amount of regularization from zero to the optimal amount of regularization results in the magnitude of the peak in the generalization getting smaller until a peak no longer exists. *However, the location of the peak does not change by changing the amount of regularization.* Further, Chen, Min, Belkin, and Karbasi [27] proved that double descent cannot take place in the under-parameterized regime for the above model.

Subsequently, works such as [10, 23, 28–31] show that there can be multiple descents in the over-parameterized regime. Specifically, d'Ascoli, Sagun, and Biroli [30] show that the first peak in triple descent is due to the norm of the estimator peaking and that the second peak is due to the initialization of the random features. Their results, Figure 3 in [30], show that the peaks only occur if the model is over-parameterized. Further Chen, Min, Belkin, and Karbasi [27] show that by considering a variety of product data distributions, any shaped risk curve can be observed in the *over-parameterized* regime. Finally, Curth, Jeffares, and van der Schaar [32] says that the peak occurs at the point of effective dimensionality of the model and not the true dimensionality. Here, we see that there are three other reasons provided for the occurrence of peaks in the risk curve.

1. Regularization can reduce the effective dimensionality of the model and move the peak to the right into the over-parameterized regime [32].
2. For random feature models, we see that the random initialization results in a second peak in the over-parameterized regime [30].
3. Due to the data having a complex covariance structure, any shaped risk curve is possible in the over-parameterized regime [27].

Other works [33–36] have considered the problem for other loss models and shown a variety of different risk curves can exist. Table 1 summarizes some of the prior work.

**Double Descent with Input Noise.** There has also been prior work that studies double descent for models with input noise rather than output noise [24, 37, 38]. From these Sonthalia and Nadakuditi [24] and Kausik, Srivastava, and Sonthalia [37] consider the unregularized problem and show that the peak occurs at the boundary. Dhifallah and Lu [38] considers ridge regularization with isotropic Gaussian data and again sees that the peak occurs at the boundary.

## 3 Spectral Properties of the Data Affect the Peak Location

In this section, we identify two important spectral properties that govern the location of the peak. In later sections, we delve into two examples that modify these properties and move the peak into the under-parameterized regime. We begin with definitions and notations. Throughout the paper, we assume that training data $X = [x_1 \ldots x_n] \in \mathbb{R}^{d \times n}$ and $y = [y_1 \ldots, y_n] \in \mathbb{R}^{k \times n}$. We are interested in the standard ridge regularized least squares problem.

$$\min_{\hat{\beta}} \|y - \hat{\beta}^T X\|_F^2 + \mu^2 \|\hat{\beta}\|^2$$

In the unregularized case, the minimum norm solution is given by $\hat{\beta}^T = yX^\dagger$, where $X^\dagger$ is the Moore-Penrose Pseudoinverse of $X$. Prior work on linear models has shown that a double descent in the risk is due to a double descent in the norm of the estimator. Suppose $X = U\Sigma V^T$ is the SVD, $\hat{\beta} \in \mathbb{R}^{d \times 1}$, then using unitary invariance, we have that

$$\|\hat{\beta}\|^2 = \sum_{i=1}^{\text{rank}(X)} \frac{(yV)_i^2}{\sigma_i^2}$$

where $\sigma_i$ is the $i$th singular value. Hence, this is controlled by

1. The alignment between $y$ and $V$. Here $V$ are the eigenvectors of the data gram matrix.
2. The spectrum of the gram matrix $X^T X$.

Many prior works deal with the alignment in one of two ways. If $y = \beta^T X + \xi$, with $\xi$ having an isotropic distribution, then prior works either assume that $\beta$ has an isotropic distribution [4, 5, 42] or they assume that $X$ is isotropic Gaussian or that $x_i = \check{\Sigma}^{1/2} z_i$, where $z_i$ is from an isotropic Gaussian [23, 43] and $\check{\Sigma}$ is a deterministic matrix. For example, if $\beta$ has an isotropic distribution, taking the expectation with respect to $\beta, \xi$ we get that

$$\mathbb{E}_{\beta,\xi}\left[\|\hat{\beta}\|^2\right] = \mathbb{E}[\beta_1^2]\|XX^\dagger\|_F^2 + \mathbb{E}[\xi_1^2]\sum_{i=1}^{\text{rank}(X)} \frac{1}{\sigma_i^2}.$$

This quantity is then studied by looking at the distribution of the spectrum as $d, n \to \infty$.

**Definition 1.** *Given a random matrix $A$, if $\lambda_1, \ldots, \lambda_k$ are its eigenvalues. Then the **empirical spectral distribution** (ESD) is the following sum of Dirac delta measure*

$$\nu^k = \frac{1}{k}\sum_{i=1}^{k}\delta_{\lambda_i}$$

*and the limiting spectral distribution $\nu$ is a measure such that $\nu^k \to \nu$ weakly almost surely.*

In general, the limiting distribution $\nu_c$ depends on the limiting aspect ratio (*i.e.,* $d/n \to c$).

**Definition 2.** *Given a measure $\nu_c$ that is supported on the interval $J \subset \mathbb{R}$, the Stieltjes transform is*

$$m_{\nu_c}(\zeta) = \mathbb{E}_{\lambda \sim \nu_c}\left[\frac{1}{\lambda - \zeta}\right], \quad \zeta \in \mathbb{C} \setminus J.$$

One common assumption is that the limiting distribution of the empirical spectral distribution is the Marchenko-Pastur distribution [44]. Other limiting distributions have been considered in [5, 45, 46]. For the Marchenko-Pastur distribution, it is known (see, for example, Lemma 5 in [24]) that

$$m_{\nu_c}(0) = \mathbb{E}_{\lambda \sim \nu_c}\left[\frac{1}{\lambda}\right] = \begin{cases} \frac{c}{1-c} & c < 1 \\ \frac{1}{c-1} & c > 1 \end{cases}.$$

Hence, the risk is governed by the value of the Stieltjes transform of the limiting spectrum at $\zeta = 0$. In particular, for the above example, the location of the peak of the risk as a function of $c$ depended on the location of the peak of

$$c \mapsto m_{\nu_c}(0) =: G(c).$$

Hence the risk depends on both the spectrum and the alignment between $y$ and $V$. Thus, the peak occurs at $c = 1$ because of the following two conditions.

> **Alignment of $y$ and right singular vectors of $X$**
>
> **Assumption 1.** *If $X = U\Sigma V^T$ is the SVD, then $yV$ is isotropic.*

> **Stieltjes Transform Peak Assumption**
>
> **Assumption 2.** *The function $c \mapsto m_{\nu_c}(0) =: G(c)$ has a local maximum at $c = 1$.*

In this paper, we show that violating either one of the above two assumptions can move the peak from the interpolation point into the **under-parameterized regime.**

## 4 Alignment Mismatch

In this section, we present the first example that exhibits double descent in the under-parameterized regime. This model violates Assumption 1.

### 4.1 Model Assumptions

For any $k \leq d$, let $\beta \in \mathbb{R}^{d \times k}$ be fixed such that the operator norm $||\beta^T||$ is $\Theta(1)$. Let $X_{trn} \in \mathbb{R}^{d \times n}$ be the signal matrix and $A_{trn} \in \mathbb{R}^{d \times n}$ be the noise matrix. Then, the ridge regularized least square estimator $W_{opt}$ is the minimum norm solution to

$$W_{opt} := \underset{W}{\arg\min} \, \|\beta^T X_{trn} - W(X_{trn} + A_{trn})\|_F^2 + \mu^2 \|W\|_F^2. \tag{1}$$

Given test data $X_{tst} + A_{tst}$, the mean squared generalization error is given by

$$\mathcal{R}(W_{opt}) = \mathbb{E}_{A_{trn}, A_{tst}} \left[ \frac{\|\beta^T X_{tst} - W_{opt}(X_{tst} + A_{tst})\|_F^2}{n_{tst}} \right]. \tag{2}$$

**Assumption 3.** *Let $\mathcal{U} \subset \mathbb{R}^d$ be a one dimensional space with a unit basis vector $u$. Then let $X_{trn} = \sigma_{trn} u v_{trn}^T \in \mathbb{R}^{d \times n}$ and $X_{tst} = \sigma_{tst} u v_{tst}^T \in \mathbb{R}^{d \times n_{tst}}$ be the respective SVDs for the training data and test data matrices. We further assume that $\sigma_{trn} = O(\sqrt{n})$ and $\sigma_{tst} = O(\sqrt{n_{tst}})$.*

There are no assumptions on the distribution of $v_{trn}, v_{tst}$ besides having unit norm. First, we see that the data $X + A$ has a spiked covariance, with the dominant eigendirection closely aligned with $u$. Since the targets only depend on $X$, we consider $A$ to represent noise. Even with the rank 1 assumption, the model captures many different scenarios. If $k = 1$, then the problem is similar to error-in-variables regression. If $k = d$ and $\beta = I$, then this is the supervised denoising problem. If the columns of $X_{trn}$ are all $\pm u$ and $\beta = u$, then this captures the binary classification problem (with MSE loss) for two Gaussian clusters centered at $u$ and $-u$ with labels $\pm 1$.

**Assumptions about $A$.** The analysis works for general assumptions in [24]. For simplicity, we assume that the matrix $A$ has I.I.D. entries drawn from a normalized Gaussian.

**Assumption 4.** *The entries of the matrices $A \in \mathbb{R}^{d \times n}$ are I.I.D. from $\mathcal{N}(0, 1/d)$.*

### 4.2 Expected Risk and Peak Location

We begin by providing a formula for the generalization error given by Equation 2 for the least squares solution given by Equation 1. All proofs are in Appendix E.

**Theorem 1** (Generalization Error Formula). *Suppose the training data $X_{trn}$ and test data $X_{tst}$ satisfy Assumption 3 and the noise $A_{trn}, A_{tst}$ satisfy Assumption 4. Let $\mu$ be the regularization parameter. Then for the under-parameterized regime (i.e., $c < 1$) for the solution $W_{opt}$ to Problem 1, the generalization error or risk given by Equation 2 is given by*

$$\mathcal{R}(c,\mu) = \frac{c\sigma_{trn}^2(\sigma_{trn}^2+1))}{2d\tau^2} \frac{1+c+\mu^2c}{\sqrt{(1-c+\mu^2c)^2+4\mu^2c^2}} - \tau^{-2}\frac{c\sigma_{trn}^2(\sigma_{trn}^2+1))}{2d} + \tau^{-2}\frac{\sigma_{tst}^2}{n_{tst}} + o\left(\frac{1}{d}\right),$$

*where*

$$\frac{1}{\tau} = \frac{2\|\beta^T u\|}{2+\sigma_{trn}^2(1+c+\mu^2c-\sqrt{(1-c+\mu^2c)+4\mu^2c^2})}.$$

*Sketch.* The proof proceeds in four key steps. First, we use the Sherman-Morrison formula for pseudoinverses [47]. Next, we decompose the error into constituent dependent quadratic forms. Through random matrix theory and concentration of measure arguments, we demonstrate that each quadratic form concentrates around a deterministic value characterized by the Stieltjes transform of the limiting empirical spectral distribution. Finally, we establish concentration bounds for the products and sums of these dependent forms, yielding the desired error rate. □

Since the focus is on the under-parameterized regime, Theorem 1 only presents the under-parameterized case. The over-parameterized case can be found in Appendix E.3. Due to the complexity of the expression, it is difficult to discern how the risk scales with respect to the training data signal strength $\sigma_{trn}^2$, the regularization strength $\mu$, or the aspect ratio $c$. Since the focus of the paper is the scaling with respect to $c$, we present the connection between the risk curve and $c$ in the main text. However, the shape of the risk curve with respect to the other parameters is also interesting and can be found in Appendix D.

To understand the shape of the risk curve as $c$ varies, we first consider that *data scaling regime*. That is, fix $d$ and change $n$. The following theorem 2 shows that the risk curve is theoretically guaranteed to have a peak at $c = \frac{1}{1+\mu^2}$.

**Theorem 2** (Under-Parameterized Peak). *Let $\mu \in \mathbb{R}_{>0}$, $\sigma_{trn}^2 = n = d/c$ and $\sigma_{tst}^2 = n_{tst}$, and $d$ is sufficiently large, so that the error term $o(1/d)$ is small, then the risk $\mathcal{R}(c)$ from Theorem 1, as a function of $c$, has a local maximum in the under-parameterized regime at $c = \frac{1}{1+\mu^2}$.*

Theorem 2 contrasts with prior works, in which double descent occurs in the over-parameterized regime or on the boundary between the two regimes. We numerically verify the predictions from Theorems 1 and 2. Figure 2 shows that the theoretically predicted risk matches the numerical risk, thus *verifying that double descent occurs in the under-parameterized regime.*[3]

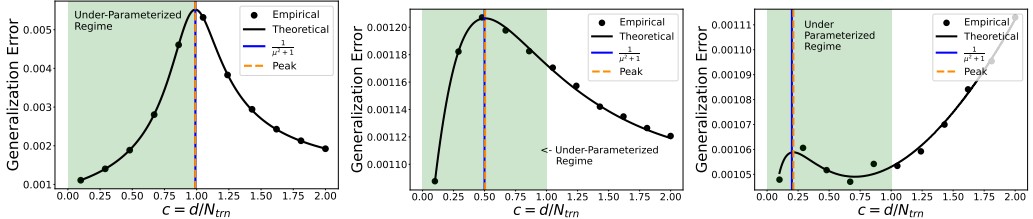

Figure 2: Figure showing the theoretical risk curve from Theorem 1 and empirical values in the data scaling regime for different values of $\mu$ [(L) $\mu = 0.1$, (C) $\mu = 1$, (R) $\mu = 2$]. Here $\sigma_{trn} = \sqrt{n}, \sigma_{tst} = \sqrt{n_{tst}}, d = 1000, n_{tst} = 1000$. For each empirical point, we ran at least 100 trials. More details can be found in Appendix G.

---

[3] All code for the experiments can be found at https://github.com/rsonthal/Under-Parameterized-Double-Descent

## 4.3 The Peak Occurs Due to Alignment Mismatch

We now show that the peak occurs due to a mismatch between the target vector and the right singular vectors of the input data. To begin, note that the ridge regularized problem can be written as follows

$$\|\beta^T \underbrace{[X_{trn}\ \mathbf{0}_{d\times d}]}_{\hat{X}_{trn}} - W([X_{trn}\ \mathbf{0}_{d\times d}] + \underbrace{[A_{trn}\ \mu I]}_{\hat{A}_{trn}})\|_F^2.$$

In this expression, $y = \beta^T \hat{X}_{trn} = (\beta^T u)[v_{trn}^T\ \mathbf{0}_p]$. Hence the direction is given by $\hat{v}_{trn}^T = [v_{trn}^T\ \mathbf{0}_p]$. The right singular vectors of $\hat{X}_{trn} + \hat{A}_{trn}$ are more difficult to compute, thus we use proxies. Since $\hat{X}_{trn}$ is rank 1, we use the right singular vectors of $\hat{A}_{trn}$ as a proxy. Lemma 5 in the appendix, shows that if $A = U\Sigma V^T$, then we can express $\hat{A}_{trn}$ as $\hat{U}\hat{\Sigma}\hat{V}^T$, where $\hat{U} = U$, $\hat{\Sigma}^2 = \Sigma^2 + \mu^2 I$, and $\hat{V} = \begin{bmatrix} V_{1:p}\Sigma\hat{\Sigma}^{-1} \\ \mu U\hat{\Sigma}^{-1} \end{bmatrix} \in \mathbb{R}^{n+d\times d}$. Here $V_{1:d}$ are the first $d$ columns of $V$. Then,

$$y\hat{V} = (\beta^T u)(\hat{v}_{trn}^T V_{1:d})\Sigma\hat{\Sigma}^{-1}.$$

Since $V_{1:d}$ came from a Gaussian random matrix, $(\hat{v}_{trn}^T V_{1:d})$ has isotropic entries. However, the diagonal matrix $\Sigma\hat{\Sigma}^{-1}$ results in the entries of $y\hat{V}$ not being isotropic. Note when $\mu = 0$, $\Sigma\hat{\Sigma}^{-1} = I$, hence it is isotropic. Hence, $\mu$ controls the deviation from isotropy.

We use $\hat{\Sigma}^T\hat{\Sigma}I$ as a proxy for the spectrum of the sample covariance. By Lemma 5, we have that $\hat{\Sigma}^T\hat{\Sigma} = \Sigma^T\Sigma + \mu^2$. We know that the limiting spectrum for $\Sigma^T\Sigma$ is the Marchenko-Pastur distribution for which the map $G(c) = m_{\nu_c}(0)$ has a maximum for $c = 1$. Shifting the spectrum changes the magnitude of the peak but does not change the location. This suggests that the peak occurs due to the misalignment between the target vector and the right singular vectors of the input data.

**Ablation experiment**    To verify that the location of the peak is due to the misalignment, we conduct two experiments. First, we solve the unregularized problem. However, we change the spectrum of the noise matrix $A_{trn}$. That is, instead of using $A_{trn} = U\Sigma V^T$, we use $\tilde{A} = U(\Sigma^2 + \mu^2 I)^{1/2}V^T$. If the spectrum was the primary factor determining the location of the peak, the peak should occur at $c = 1/(1 + \mu^2)$. However, as seen in Figure 3a it still occurs at $c = 1$. Second, we replace $\hat{V}$ with a uniformly random orthogonal matrix $Q$ [4]. Clearly, $yQ$ is now isotropic. Figure 3b shows that, in this case, the peak moves to the over-parameterized regime.

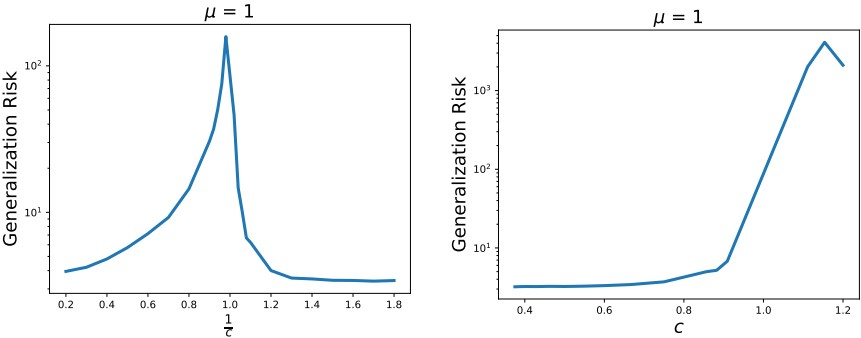

Figure 3: Risk for the ablation experiment. Left: Empirical Expected Risk when using $\tilde{A}$ for the noise. Right: Empirical risk when we replace $\hat{V}$ with a random orthogonal matrix.

**Connection to Prior Double Descent Theory**    Prior double descent theory postulates that the peak for the ridge regularized model occurs at the interpolation point for the unregularized model or further to the right into the over-parameterized regime. Hence, this model goes against prior expectations, with the peak moving to the left into the under-parameterized regime. However, there might still be some connection between the training error and the location of the peak. For example, the peak may correspond to a local minimum of the training loss. We explore this in Appendix C and see only a weak connection with the third derivative of the training error.

---

[4]We obtain such a matrix by computing the full SVD of a Gaussian random vector in Pytorch

### 4.4 Peak in the Norm of the Estimator Does Not Imply a Peak in the Risk

Prior double descent theories suggest that double descent occurs due to the norm of the estimator increasing and then decreasing. This is true for the above model where we fixed $d$ and varied $n$. However, the connection breaks if we fix $n$ and vary $d$ instead (*parameter scaling regime*). This difference between the two regimes is due to the normalization considered and has been observed before [30].

Figure 4 shows that for the parameter scaling regime, for small values of $\mu$, we see underparameterized double descent. However, as we increase $\mu$, the risk curve becomes monotonic. Nevertheless, as shown in Figure 5, for larger values of $\mu$, *there is still a peak* in the curve for the norm of the estimator $\|W_{opt}\|_F^2$. Hence, the curve for the norm of the estimator exhibits underparameterized double descent even if the risk does not. This is further highlighted in Figure 6. Here, we see that even though the variance is non-monotonic, the risk is dominated by the bias term. Thus, we show that a peak in the generalization error for linear models does not imply a peak in the norm of the estimator. The following theorem provides a local maximum in the $\mathbb{E}\left[\|W_{opt}\|_F^2\right]$ curve for $c < 1$.

**Theorem 3** ($\|W_{opt}\|_F$ Peak). *If $\sigma_{tst} = \sqrt{n_{tst}}$, $\sigma_{trn} = \sqrt{n}$ and $\mu$ is such that $p(\mu) < 0$, then for fixed $n$ that is sufficiently large enough, we have that $\mathbb{E}\left[\|W_{opt}\|_F\right]$ versus $c = d/n$ curve has a local maximum in the under-parameterized regime at $c = (\mu^2 + 1)^{-1}$.*

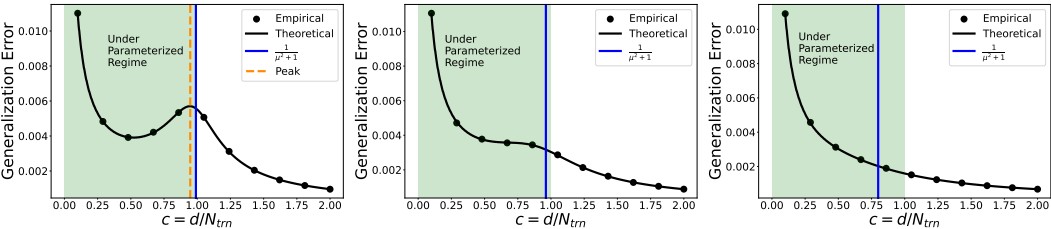

Figure 4: Figure showing the theoretical risk curve from Theorem 1 and empirical values in the parameter scaling regime for different values of $\mu$ [(L) $\mu = 0.1$, (C) $\mu = 0.2$, (R) $\mu = 0.5$]. Here, only $\mu = 0.1$ has a local peak. Here $n = n_{tst} = 1000$ and $\sigma_{trn} = \sigma_{tst} = \sqrt{1000}$. Each empirical point is an average of 100 trials.

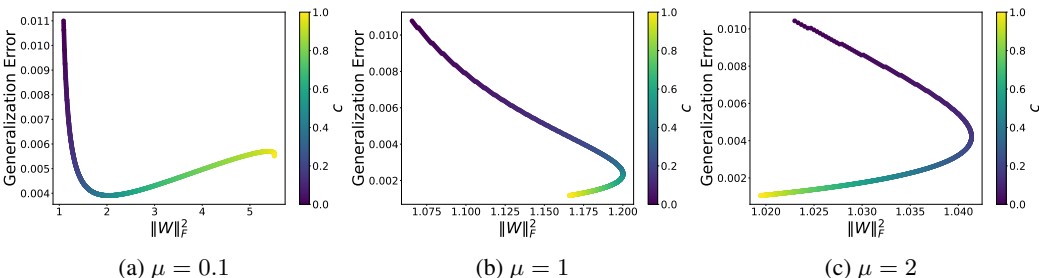

(a) $\mu = 0.1$      (b) $\mu = 1$      (c) $\mu = 2$

Figure 5: Figure showing generalization error versus $\mathbb{E}\left[\|W_{opt}\|_F^2\right]$ for the parameter scaling regime for three different values of $\mu$.

## 5 Shifting Local Maximum for Stieltjes Transform as a Function of $c$

In this section, we present the next example of under-parameterized double descent that violates Assumption 2. In particular, the maximum of the map $c \mapsto m_{\nu_c}(0)$ does not occur at $c = 1$. We show that the maximum can be chosen to be any value in $(0, 1)$. We consider the following mixture model. Let $\pi_1, \pi_2$ be mixture weights such that $\pi_1 + \pi_2 = 1$. Then, with probability $\pi_1$, the data is sampled from $\mathcal{N}(0, \frac{1}{d}I)$ and with probability $\pi_2$, the data point is $\alpha z$ for fixed $z \in \mathbb{R}^d$ and $\alpha \sim \mathcal{N}(0, 1)$. For this model, the uncentered covariance matrix is given by

$$\mathbb{E}[xx^T] = \pi_1 \mathbb{E}_{x \sim \mathcal{N}(0, \frac{1}{d}I)}[xx^T] + \pi_2 \mathbb{E}[\alpha^2 zz^T] = \frac{\pi_1}{d}I + \pi_2 zz^T.$$

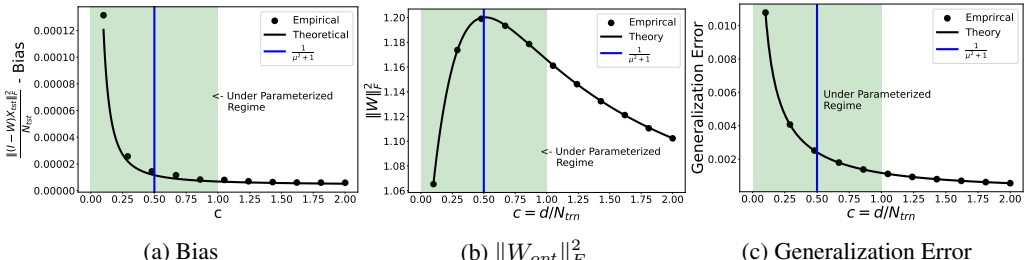

| (a) Bias | (b) $\|W_{opt}\|_F^2$ | (c) Generalization Error |

Figure 6: Figure showing the $\mathbb{E}\left[\|W_{opt}\|_F^2\right]$, and the generalization error in the parameter scaling regime for $\mu = 1$, $\sigma_{trn} = \sqrt{n}$, and $\sigma_{tst} = \sqrt{n_{tst}}$. Here $n = 1000$ and $n_{tst} = 1000$. For each empirical data point, we ran at least 100 trials. More details can be found in Appendix G.

Then, the expected excess risk for a solution $\hat{\beta}$ compared to $\beta$ is

$$\mathcal{R} = \mathbb{E}[\|\beta^T x - \hat{\beta}^T x\|^2 | X] = \mathbb{E}\left[\frac{\pi_1}{d}\|\beta^T - \hat{\beta}^T\|^2 + \pi_2\|(\beta - \hat{\beta})^T z\|^2 | X\right].$$

Let $X_{trn} = [A \ zv^T] \in \mathbb{R}^{d \times n}$, where the $A \in \mathbb{R}^{d \times n-k}$ with each column a data point sampled I.I.D. from $\mathcal{N}(0, \frac{1}{d}I)$ and $v \in \mathbb{R}^k$ is the vector with random coefficients in front of $z$. Let $\beta \in \mathbb{R}^d$ be the target regressor function and let $y^T = \beta^T X_{trn} + \xi_{trn}^T$, where $\xi_{trn}^T$ has I.I.D. entries from a standard normal. Let $\beta_{opt}^T = y^T X_{trn}^T (X_{trn} X_{trn}^T)^{-1}$ be the minimum norm. Then, Theorem 4 shows that the peak occurs when $d/n = c = \pi_1 < 1$. To experimentally verify Theorem 4, we consider two cases, one where we enforce $\beta \perp z$ and one where we do not. As seen in Figure 7, Theorem 4 is accurate for both cases. This suggests that the $\beta \perp z$ assumption seems to only be needed for simplifying the proof.

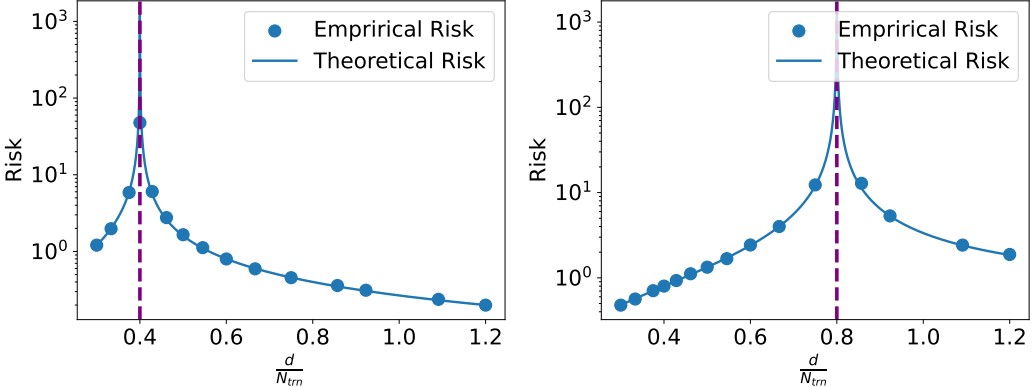

Figure 7: Figure showing under-parameterized double descent. (Left) We have $\beta = d \cdot z$. (Right) We have $\beta \perp z$. The solid blue line represents the theoretical estimate from Theorem 4 and the scatter points are from empirical experiments with $d = 600$. For the empirical points, we average over 50 trials. The dashed vertical purple line is $\pi_1$.

**Theorem 4** (Under-parametrized Peak). *For the above model, if $k/n \to \pi_2$, and $d/n \to c$, then the expected risk is given by* $\mathcal{R} = \begin{cases} \frac{\pi_1 c}{\pi_1 - c} & c < \pi_1 \\ \pi_1 \left(\frac{\pi_1}{c - \pi_1} + \left(1 - \frac{\pi_1}{c}\right)\left(\frac{\|\beta\|^2}{d} - \frac{(\beta^T z)^2}{\|z\|^2 d}\right)\right) & c > \pi_1 \end{cases}$.

Theorem 4 is quite surprising as it shows that the *only* peak in the risk curve occurs in the under-parameterized regime. One might assume that is due to the low rankness of the data from the second mixture. While this is true, prior work does not indicate that this is the reason. Specifically, Huang, Hogg, and Villar [41] shows that projecting onto low-dimensional versions of the data acts as a regularizer and removes the peak altogether. Xu and Hsu [39], also looks at a similar problem, but they consider isotropic Gaussian data and project onto the first $p$ components. In this case, the data is artificially high-dimensional (since only the first $k$ coordinates are non-zero).

They again see a peak at the interpolation point ($n = p$). Wu and Xu [40] also looks at a version of Principal Component Regression in which the data dimension is reduced. That is, the data is not embedded in high-dimensional space anymore. Wu and Xu [40] sees a peak at the boundary between the under and over-parameterized regions. Finally, Sonthalia and Nadakuditi [24] and Kausik, Srivastava, and Sonthalia [37] look at the denoising problem for low-dimensional data and have peaks at $c = 1$. Therefore, prior work does not immediately imply that low dimensional data results in under-parameterized double descent. If we had only low-dimensional data, then the peak "should" move into the over-parameterized regime. This is because if the true dimensionality of the data is $r < d$. Then, one might think that the peak occurs when the number of training data points $n$ equals $r$ since that is the interpolation point[5]. We are in the over-parameterized regime since $d > r = n$.

We can understand this phenomenon as follows. The data from the second mixture does not affect the smallest eigenvalue of the covariance matrix. This is because the second mixture lives in a one-dimensional space. Hence, it only affects the top eigenvalue. Since the Stieltjes transform at 0 is dominated by the behavior of the smallest non-zero eigenvalue, data from the second mixture has very little effect on the Stieltjes transform of the ESD at 0. We expect the above intuition to hold, even when replacing rank 1 with rank $r$ for any fixed small $r$.

**Connection to Prior Double Descent Theory**    For this example, it is easy to show that there is a strong connection to prior double descent theory. Specifically, even though we cannot interpolate the data, the minimum training error will occur at $c = \pi_1$. Further, we see that the blow-up in the excess risk is due to the norm of the estimator blowing up.

Additionally, we see that comparing with the result from [11] (which is the case when $\pi_2 = 0$), we see that the $\pi_2 > 0$ results in sifting the peak to $\pi_1$ and rescales the variance by $\pi_1$. However, we also see an additional correction term in the overparameterized regime:

$$\left(1 - \frac{\pi_1}{c}\right)\left(-\frac{(\beta^T z)^2}{\|z\|^2 d}\right)$$

Here we see that the term depends on the alignment between the target $\beta$ and the spike direction $z$.

## 6   Conclusion

This paper presents two simple models with double descent in the under-parameterized regime. While such peaks seem limited to special cases, understanding the cause is important for a complete understanding of the double descent phenomenon. Our analysis reveals that the location of peaks depends critically on two properties: the alignment between targets and the eigenvectors of the training data gram matrix and the behavior of the Stieltjes transform of the limiting empirical spectral distribution.

We demonstrate that violating either of these properties can shift the peak into the under-parameterized regime. The first model shows that ridge regularization can create a misalignment between targets and singular vectors, leading to a peak at $c = 1/(1 + \mu^2)$. The second model, using a mixture of isotropic Gaussian vectors and directional vectors, demonstrates that modifying the spectrum can result in a peak at $c = \pi_1$. These findings challenge several prevailing theories about double descent. They show that peaks need not occur at or beyond the interpolation threshold and that a peak in the estimator's norm does not necessarily imply a peak in the generalization error.

Investigating the interaction between spectral properties and generalization in deep neural networks to provide a general theory of double descent is an important avenue for future work. Understanding whether similar phenomena occur in other architectures and the implications for model selection and regularization remain open questions.

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

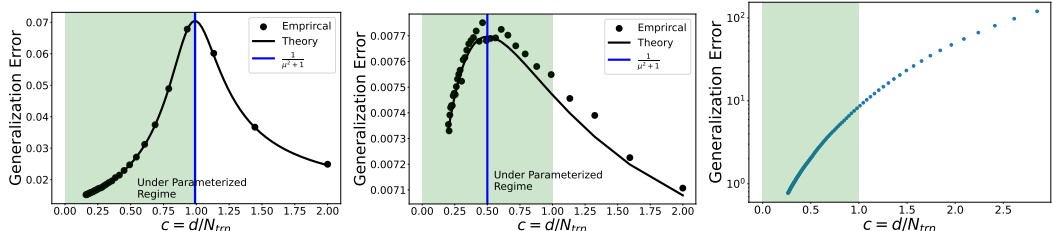

Figure 8: Generalization Error for low-dimension MNIST using a linear denoiser. For the left figure, we use 10 dimensions and $\mu = 0.1$. For the central figure, we use 5 dimensions and $\mu = 1$. For the right figure, we use 784 dimensions and $\mu = 1$.

## A  Higher Rank for Denoising Model

One might be led to believe that the restriction that the data lie on a line embedded in high dimensional space is crucial to the appearance of this phenomenon. However, this is not true. As long as the rank of the data is relatively low, for the input noise setting, we can see this phenomenon. Hence, we extend our results beyond the one-dimensional case to the low-dimensional case. Due to space constraints, the conjectured formula for the risk is in Appendix F.[6] We verify the conjectured formula as well as the role of low dimensionality using MNIST data. Specifically, we project the data onto a $r$-dimensional subspace. We then add noise to the low-dimensional representation and then solve the denoising problem. The left two figures in Figure 8 show that the phenomenon exists for low-dimensional data. That is, we see that a peak occurs in the under-parameterized regime, and the location of the peak seems to occur at $\frac{1}{\mu^2+1}$. However, running the same experiment without projecting to a low-dimensional space (right figure) results in very different phenomena. We no longer see double descent at all. Hence we see that if a peak occurs, then its location does not depend on the dimension. However, the occurrence of the peak does depend on the dimension. Thus, we see that this complements the results in [38].

---

[6]While these are currently presented as a conjecture. This is because we only computed the expectation terms. Careful analysis of the variance would allow us to formalize the statement.

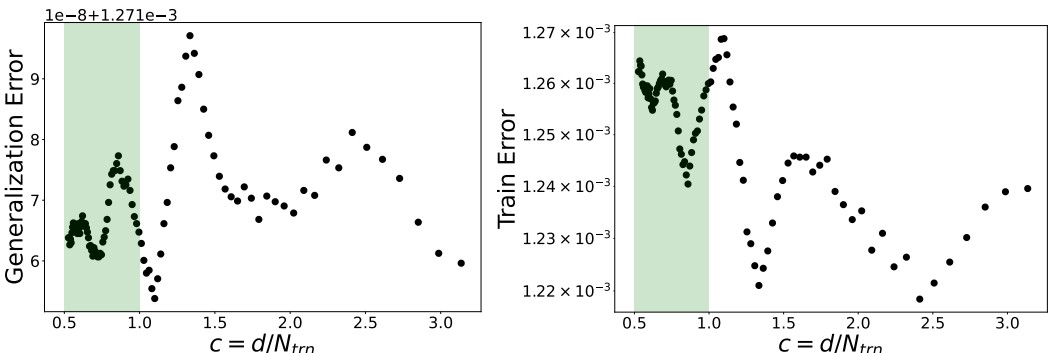

Figure 9: Generalization Risk and Training error for denoising a low-dimensional version of MNIST using a 1-hidden layer neural network. Here $d = 2 \cdot 784$.

# B    Non-Linear Model

To show that under-parameterized peak occurs for non-linear models. We conducted an experiment on MNIST with a 1-hidden layer neural network with a width of 784. The network has no bias, so has $2 \cdot 784^2$ parameters. Let $\mathcal{V}$ be a random 10-dimensional subspace. We project our data onto $\mathcal{V}$, add noise to the low dimensional representation, and train our network to remove this noise. We use full batch gradient descent with weight decay of $0.001$ to train the model for 500 epochs with a learning rate of $2 \times 10^{-2}$. We test it on the complete MNIST test data. Figure 9 shows the training error and generalization error as a function of the number of training data points. As seen in the figure, we have multiple peaks in the under-parameterized regime, and the peaks in the risk correspond to local minimums in the training error. Interestingly, the risk curve here exhibits 4 peaks!

There are, however, many crucial differences between the peaks for this neural network case and the linear model case. First, the location of the peak does not seem to depend on the strength of the ridge regularization. Second, the peak seems to directly correspond to the local minimums of the training error. Hence, while we see peaks in the under-parameterized regime, the mechanisms that create these peaks are likely to be different.

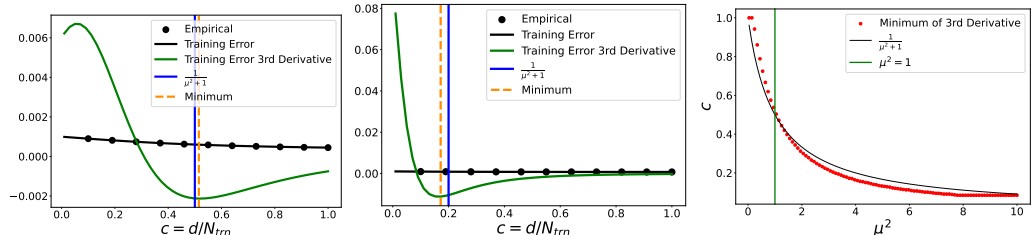

Figure 10: Figure showing the training error, the third derivative of the training error, and the location of the peak of the generalization error for different values of $\mu$ [(L) $\mu = 1$, (C) $\mu = 2$] for the data scaling regime. (R) shows the location of the local minimum of the third derivative and $\frac{1}{\mu^2+1}$.

## C    Training Error

As seen in the prior section, the peak occurs in the interior of the under-parameterized regime and not on the border between the under-parameterized and over-parameterized regimes. We have also seen that it does not necessarily occur whenever there is a peak in the norm of the estimator. The final postulate for where the peak occurs is that it occurs when we first hit zero training error.

In this section, we explore the connection between the training error and the risk. Theorem 5 derives a formula for the training error in the under-parameterized regime.

**Theorem 5** (Training Error). *Let $\tau$ be as in Theorem 1. The training error for $c < 1$ is given by*

$$\mathbb{E}_{A_{trn}}[\|X_{trn} - W_{opt}(X_{trn} + A_{trn})\|_F^2] = \tau^{-2}\left(\sigma_{trn}^2\left(1 - c \cdot T_1\right) + \sigma_{trn}^4 T_2\right) + o(1),$$

*where* $T_1 = \dfrac{\mu^2}{2}\left(\dfrac{1 + c + \mu^2 c}{\sqrt{(1 - c + \mu^2 c)^2 + 4\mu^2 c^2}} - 1\right) + \dfrac{1}{2} + \dfrac{1 + \mu^2 c - \sqrt{(1 - c + \mu^2 c)^2 + 4c^2\mu^2}}{2c}$,

*and*

$$T_2 = (\mu^2 c + c - 1 - \sqrt{(1 - c + \mu^2 c)^2 + 4c^2\mu^2})^2\left(\frac{\mu^2 c + c + 1}{2\sqrt{(1 - c + \mu^2 c)^2 + 4c^2\mu^2}} + \frac{1}{2}\right).$$

Since we are studying the ridge regularized problem, it is impossible for the training error to be exactly equal to 0. Hence, we may expect the peak to correspond to other features of the training error curve. Given the analytical formula for the training error, we can compute the derivatives. We found that the training error curve does not seem to signal the location of the peak in the generalization error curve.

Since we are studying the ridge regularized problem, it is impossible for the training error to be exactly equal to 0. Hence, we may expect the peak to correspond to other features of the training error curve. For example, the peak could correspond to a local minimum of the training error, or it could correspond to a point where the training error or its derivatives suddenly change. The first, a local minimum, is easy to see from the plot of the training error, but the second can be more subtle as we do not usually have access to the derivatives. However, since we have an analytical formula for the training error, we can compute the derivatives.

Figure 10 plots the location of the peak and the training error. Here, the figure shows that the training error curve does not seem to signal the location of the peak in the generalization error curve. However, it shows that for the data scaling regime, the peak roughly corresponds to a local minimum of the third derivative of the training error. While the minimum of the third derivative is difficult to interpret, as we can see from the third plot in Figure 10, the minimum seems to closely track the location of the peak.

# D Regularization Trade-off

It has been seen in prior work that the amount of noise added to the input data can be viewed as a regularizer [24, 49]. In our setup, we have two different regularizers: the amount of noise added to the data (since we are dealing with linear models, this is equivalent to the strength of the signal $\sigma_{trn}$) and the strength of the ridge regularizer $\mu$. It is interesting to analyze the trade-off between the two regularizers and the generalization error.

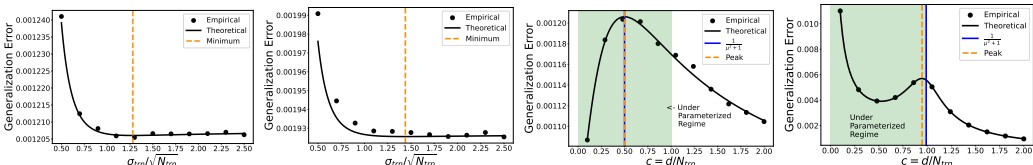

Figure 11: The first two figures show the $\sigma_{trn}$ versus risk curve for $c = 0.5, \mu = 1$ and $c = 2, \mu = 0.1$ with $d = 1000$. The second two figures show the risk when training using the optimal $\sigma_{trn}$ for the data scaling and parameter scaling regimes.

**Optimal $\sigma_{trn}$**  First, we fix $\mu$ and determine the optimal $\sigma_{trn}$. Figure 11 displays the generalization error versus $\sigma_{trn}^2$ curve. The figure shows that the error is initially large but then decreases until the optimal generalization error. The generalization error when using the optimal $\sigma_{trn}$ is also shown in Figure 11. Here, unlike [23], picking the optimal value of $\sigma_{trn}$ does not mitigate double descent.

**Proposition 1** (Optimal $\sigma_{trn}$). *The optimal value of $\sigma_{trn}^2$ for $c < 1$ is given by*

$$\sigma_{trn}^2 = \frac{\sigma_{tst}^2 d[2c(\mu^2+1)^2 - 2T(c\mu^2+c+1) + 2(c\mu^2-2c+1)] + N_{tst}(\mu^2 c^2 + c^2 + 1 - T)}{N_{tst}(c^3(\mu^2+1)^2 - T(\mu^2 c^2 + c^2 - 1) - 2c^2 - 1)}.$$

Additionally, it is interesting to determine how the optimal value of $\sigma_{trn}$ depends on both $\mu$ and $c$. Figure 12 shows that for small values of $\mu \in (0.1, 0.5)$, as $c$ changes, there exists an (inverted) double descent curve for the optimal value of $\sigma_{trn}$. However, for the data scaling regime, the minimum of this double descent curve *does not match the location for the peak of the generalization error*. Further, as the amount of ridge regularization increases, the optimal amount of noise regularization decreases proportionally; optimal $\sigma_{trn}^2 \approx d\mu^2$. Thus, for higher values of ridge regularization, it is preferable to have higher-quality data.

**Optimal Value of $\mu$**  We now explore the effect of fixing $\sigma_{trn}$ and then changing $\mu$. Figure 13, shows a $U$ shaped curve for the generalization error versus $\mu$, suggesting that there is an optimal value of $\mu$, which should be used to minimize the generalization error. Next, we compute the optimal value of $\mu$ using grid search and plot it against $c$. Figure 14 shows double descent for the optimal value of $\mu$ for small values of $\sigma_{trn}$. Thus, for low SNR data, we see a double descent, but we do not for high SNR data.

Finally, for a given value of $\mu$ and $c$, we compute the optimal $\sigma_{trn}$. We then compute the generalization error (when using the optimal $\sigma_{trn}$) and plot the generalization error versus $\mu$ curve. Figure 15 displays a very different trend from Figure 13. Instead of having a $U$-shaped curve, we have a

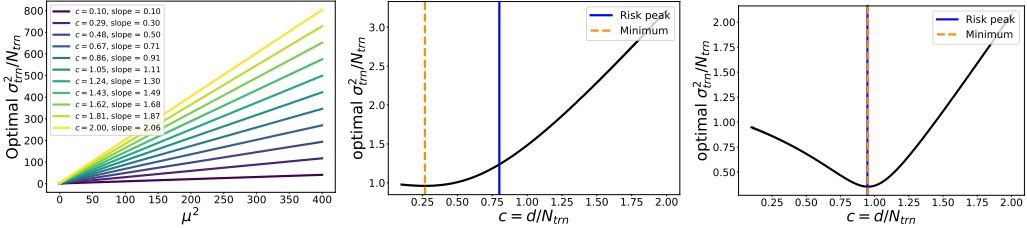

Figure 12: The first figure plots the optimal $\sigma_{trn}^2/n$ versus $\mu$ curve. The middle figure plots the optimal $\sigma_{trn}^2/n$ versus $c$ in the data scaling regime for $\mu = 0.5$, and the last figure plots the optimal $\sigma_{trn}^2/n$ versus $c$ in the parameter scaling regime for $\mu = 0.1$.

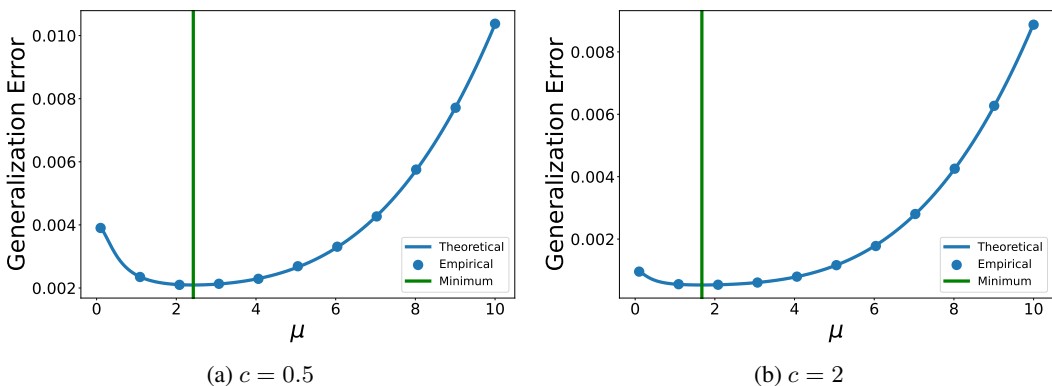

(a) $c = 0.5$              (b) $c = 2$

Figure 13: Figure showing the generalization error versus $\mu$ for $\sigma_{trn}^2 = n$ and $\sigma_{tst}^2 = N_{tst}$.

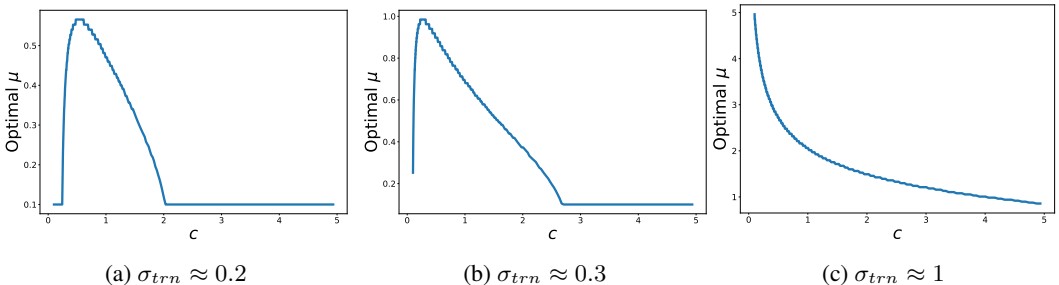

(a) $\sigma_{trn} \approx 0.2$       (b) $\sigma_{trn} \approx 0.3$       (c) $\sigma_{trn} \approx 1$

Figure 14: Figure for the optimal value of $\mu$ verses for different values of $\sigma_{trn}$

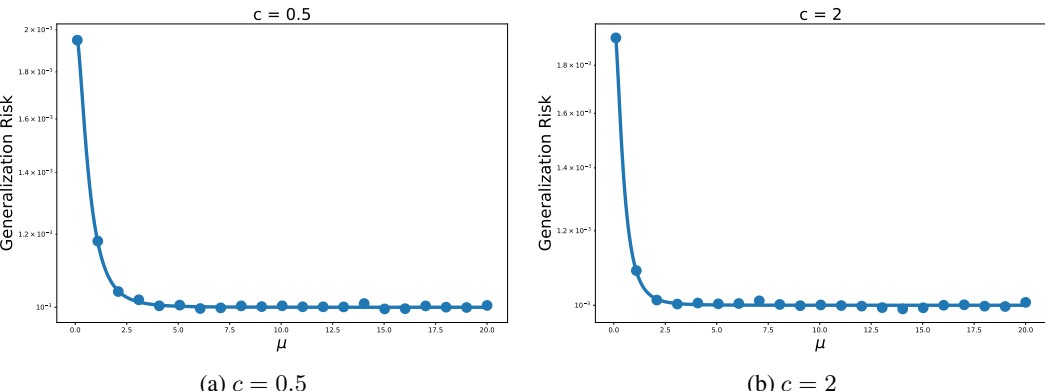

(a) $c = 0.5$              (b) $c = 2$

Figure 15: Figure showing the generalization error versus $\mu$ for the optimal $\sigma_{trn}^2$ and $\sigma_{tst}^2 = N_{tst}$.

monotonically decreasing generalization error curve. *This suggests that we can improve generalization by using higher-quality training while compensating for this by increasing the amount of ridge regularization.*

**Interaction Between the Regularizers** The optimal values of $\mu$ and $\sigma_{trn}$ are jointly computed using grid search for $\mu \in (0, 100]$ and $\sigma_{trn}/\sqrt{n} \in (0, 10]$. Figure 16 shows the results. Specifically, $\sigma_{trn}$ is at the highest possible value (so best quality data), and then the model regularizes purely using the ridge regularizer. This results in a monotonically decreasing generalization error curve. Thus, in the data scaling model, *there is an implicit bias that favors one regularizer over the other.* Specifically, the model's implicit bias *is to use higher quality data while using ridge regularization to regularize the model appropriately.* It is surprising that the two regularizers are not balanced.

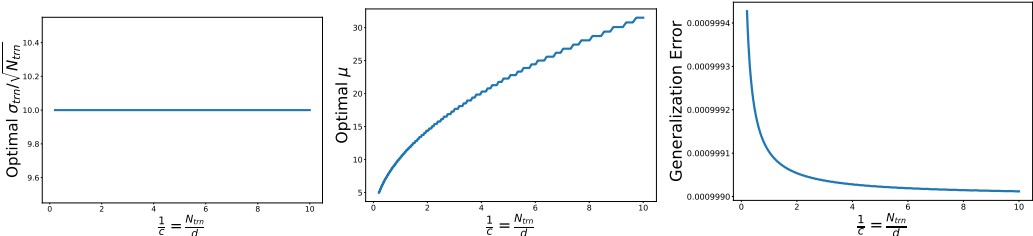

Figure 16: Trade-off between the regularizers. The left column is the optimal $\sigma_{trn}$, the central column is the optimal $\mu$, and the right column is the generalization error for these parameter restrictions.

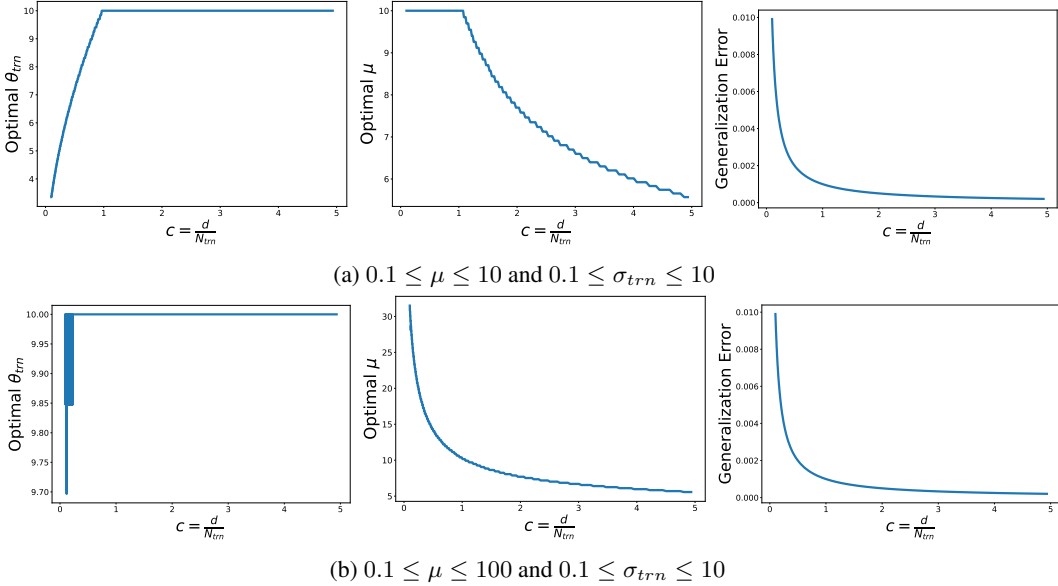

(a) $0.1 \leq \mu \leq 10$ and $0.1 \leq \sigma_{trn} \leq 10$

(b) $0.1 \leq \mu \leq 100$ and $0.1 \leq \sigma_{trn} \leq 10$

Figure 17: Trade-off between the regularizers. The left column is the optimal $\sigma_{trn}$, the central column is the optimal $\mu$, and the right column is the generalization error for these parameter restrictions

Next we look at the trade-off between $\sigma_{trn}$ and $\mu$ for the parameter scaling regime. We again see, Figure 17, that the model implicitly prefers regularizing via ridge regularization and not via input data noise regularizer.

# E    Proofs

## E.1    Linear Regression

We begin by noting,

$$\beta^T = (\beta_{opt}^T X + \xi_{trn}) X_{trn}^\dagger.$$

Thus, we have,

$$\|\beta\|^2 = \text{Tr}(\beta^T \beta)$$
$$= \text{Tr}(\beta_{opt}^T X_{trn} X_{trn}^\dagger (X_{trn}^\dagger)^T X_{trn} \beta_{opt}) + \text{Tr}(\xi_{trn} X_{trn}^\dagger (X_{trn}^\dagger)^T \xi_{trn}^T) + 2 \text{Tr}(\beta_{opt}^T X_{trn} X_{trn}^\dagger X_{trn}^\dagger)^T \xi_{trn}^T.$$

Taking the expectation, with respect to $\xi_{trn}$, we see that the last term vanishes.

Letting $X_{trn} = U_X \Sigma_X V_X^T$. We see that using the rotational invariance of $X$, $U_X, V_X$ are independent and uniformly random. Thus, $s := \beta_{opt}^T U_X$ is a uniformly random unit vector.

Thus, we see,

$$\mathbb{E}_{X_{trn}, \xi_{trn}} \left[ \text{Tr}(\beta_{opt}^T X_{trn} X_{trn}^\dagger (X_{trn}^\dagger)^T X_{trn} \beta_{opt}) \right] = \sum_{i=1}^{\min(d,n)} \mathbb{E}[s_i^2] = \min\left(1, \frac{1}{c}\right)$$

Similarly, we see,

$$\mathbb{E}_{X_{trn}, \xi_{trn}} \left[ \xi_{trn} X_{trn}^\dagger (X_{trn}^\dagger)^T \xi_{trn}^T \right] = \sum_{i=1}^{\min(d,n)} \mathbb{E}\left[ \frac{1}{\sigma_i (X_{trn})^2} \right]$$

Multiplying and dividing by $d$, normalizes the singular values squared of $X_{trn}$ so that the limiting distribution is the Marchenko Pastur distribution with shape $c$. Thus, we can estimate using Lemma 5 from Sonthalia and Nadakuditi [24] to get,

$$\begin{cases} \frac{c}{1-c} + o(1) & c < 1 \\ \frac{1}{c-1} + o(1) & c > 1 \end{cases}.$$

Finally, the cross-term has an expectation equal to zero. Thus,

$$\mathbb{E}_{X_{trn}, \xi_{trn}}[\|\beta_{opt}\|^2] = \begin{cases} 1 + \frac{c}{1-c} & c < 1 \\ \frac{1}{c} + \frac{1}{c-1} & c > 1 \end{cases}$$

Then we have,

$$\beta^T \beta_{opt} = \beta_{opt}^T X_{trn} X_{trn}^\dagger \beta_{opt} + \xi_{trn} X_{trn}^\dagger \beta_{opt}$$

The second term has an expectation equal to zero, and the first term is similar to before and has an expectation equal to $\min\left(1, \frac{1}{c}\right)$.

## E.2    Proof for Output Noise Model

**Theorem 4** (Under-parametrized Peak). *For the above model, if $k/n \to \pi_2$, and $d/n \to c$, then the expected risk is given by* $\mathcal{R} = \begin{cases} \frac{\pi_1 c}{\pi_1 - c} & c < \pi_1 \\ \pi_1 \left( \frac{\pi_1}{c - \pi_1} + \left(1 - \frac{\pi_1}{c}\right) \left( \frac{\|\beta\|^2}{d} - \frac{(\beta^T z)^2}{\|z\|^2 d} \right) \right) & c > \pi_1 \end{cases}.$

*Proof.* We begin with $c < \pi_1$. Here, since $AA^T$ is invertible, we can see that

$$\hat{\beta}^T = \left( \beta^T \begin{bmatrix} A & zv^T \end{bmatrix} + \xi^T \right) \begin{bmatrix} A^T \\ vz^T \end{bmatrix} \left( AA^T + \|v\|^2 zz^T \right)^{-1}$$

Let us focus on the term without the $\xi$. Using the Sherman Morrison formula and the orthogonality of $\beta$ and $z$, we see that

$$\left( \beta^T \begin{bmatrix} A & zv^T \end{bmatrix} \right) \begin{bmatrix} A^T \\ vz^T \end{bmatrix} \left( AA^T + \|v\|^2 zz^T \right)^{-1} = \beta^T$$

Thus, we get that

$$\hat{\beta}^T = \beta^T + \xi^T \begin{bmatrix} A^T \\ vz^T \end{bmatrix} \left(AA^T + \|v\|^2 zz^T\right)^{-1}$$

For this model, the uncentered covariance matrix is given by

$$\mathbb{E}[xx^T] = \pi_1 \mathbb{E}_{x \sim \mathcal{N}(0, \frac{1}{d}I)}[xx^T] + \pi_2 \mathbb{E}[\alpha^2 zz^T] = \frac{\pi_1}{d}I + \pi_2 zz^T.$$

Then, the expected excess risk for a solution $\hat{\beta}$ compared to $\beta$ is

$$\mathcal{R} = \mathbb{E}[\|\beta^T x - \hat{\beta}^T x\|^2 | X] = \mathbb{E}\left[\frac{\pi_1}{d}\|\beta^T - \hat{\beta}^T\|^2 + \pi_2 \|(\beta - \hat{\beta})^T z\|^2 | X\right].$$

The first term is given by

$$\mathbb{E}[\|\beta - \hat{\beta}^T\|^2 | X] = \mathbb{E}\left[\left\|\xi^T \begin{bmatrix} A^T \\ vz^T \end{bmatrix} \left(AA^T + \|v\|^2 zz^T\right)^{-1}\right\|^2 | X\right]$$

Taking the expectation over $\xi$, we get that the expected excess risk is given by

$$\text{Tr}\left(\begin{bmatrix} A^T \\ vz^T \end{bmatrix} \left(AA^T + \|v\|^2 zz^T\right)^{-2} \begin{bmatrix} A & zv^T \end{bmatrix}\right) = \text{Tr}\left(\left(AA^T + \|v\|^2 zz^T\right)^{-1}\right)$$

Again, using the Sherman-Morrison formula, we get that

$$\text{Tr}((AA^T + \|v\|zz^T)^{-1}) = \text{Tr}((AA^T)^{-1}) - \frac{\|v\|^2}{1 + \|v\|^2 z^T(AA^T)^{-1}z} \text{Tr}\left(z^T(AA^T)^{-2}z\right)$$

Suppose $\nu$ is the limiting distribution of the empirical spectral distribution for the non-zero eigenvalues. Then using the concentration results from [24], we see that the error can be expressed

$$\text{Tr}((AA^T)^{-1}) - \frac{\|v\|^2}{1 + \|v\|^2 z^T(AA^T)^{-1}z} \text{Tr}\left(z^T(AA^T)^{-2}z\right) = d \cdot \mathbb{E}_{\lambda \sim \nu}\left[\frac{1}{\lambda}\right] - \frac{\|v\|^2}{1 + \|v\|^2 \mathbb{E}_{\lambda \sim \nu}\left[\frac{1}{\lambda}\right]} \mathbb{E}_{\lambda \sim \nu}\left[\frac{1}{\lambda^2}\right]$$

The second term of the expected risk is

$$\left\|\xi^T \begin{bmatrix} A^T \\ vz^T \end{bmatrix} \left(AA^T + \|v\|zz^T\right)^{-1} z\right\|^2$$

Again, if we take the expectation over $\xi$ and write as a trace, we get

$$\text{Tr}(z^T \left((AA^T)^{-1} - \frac{(AA^T)^{-1}\|v\|^2 zz^T(AA^T)^{-1}}{1 + \|v\|^2 z^T(AA^T)^{-1}z}\right) z) = \frac{z^T(AA^T)^{-1}z}{1 + \|v\|^2 z^T(AA^T)^{-1}z}$$

Again, this can expressed as

$$\frac{\mathbb{E}_{\lambda \sim \nu}\left[\frac{1}{\lambda}\right]}{1 + \|v\|^2 \mathbb{E}_{\lambda \sim \nu}\left[\frac{1}{\lambda}\right]}$$

Putting it all together, the expected excess risk is

$$\pi_1 \mathbb{E}_{\lambda \sim \nu}\left[\frac{1}{\lambda}\right] + \pi_2 \frac{\mathbb{E}_{\lambda \sim \nu}\left[\frac{1}{\lambda}\right]}{1 + \|v\|^2 \mathbb{E}_{\lambda \sim \nu}\left[\frac{1}{\lambda}\right]} - \frac{\pi_1}{d} \frac{\|v\|^2}{1 + \|v\|^2 \mathbb{E}_{\lambda \sim \nu}\left[\frac{1}{\lambda}\right]} \mathbb{E}_{\lambda \sim \nu}\left[\frac{1}{\lambda^2}\right]$$

If we then send $n, k, d$ to infinity, while noting that $\|v\| \to \infty$, then we get then limiting risk

$$\pi_1 \mathbb{E}_{\lambda \sim \nu}\left[\frac{1}{\lambda}\right] = \frac{\pi_1 c}{\pi_1 - c}$$

For the $d > n - k$ case, $AA^T$ is no longer invertible. Hence, we need to replace the inverse with pseudoinverse. Hence, we have that

$$\hat{\beta}^T = \left(\beta^T \begin{bmatrix} A & zv^T \end{bmatrix} + \xi^T\right) \begin{bmatrix} A^T \\ vz^T \end{bmatrix} \left(AA^T + \|v\|^2 zz^T\right)^\dagger$$

We now expand the pseudoinverse using Theorem 1 from [47]. To write is succinctly, let $M = AA^T$, $P = (I - MM^\dagger)$, $\gamma = z^T Pz$, and $\tau = z^T M^\dagger z$ to get that

$$(AA^T + \|v\|^2 zz^T)^\dagger = (AA^T)^\dagger - \frac{1}{\gamma}\left((AA^T)^\dagger zz^T P + Pzz^T(AA^T)^\dagger\right) + \frac{(1 + \|v\|^2)\tau}{\|v\|^2\gamma^2}Pzz^T P$$

Note that $P$ is the projection onto the orthogonal complement of the range of $A$. Hence we see that $MP = M^\dagger P = 0$. Thus, multiplying through, and setting terms to zero, we see that

$$(M + \|v\|^2 zz^T)(M + \|v\|^2 zz^T)^\dagger = MM^\dagger - \frac{1}{\gamma}MM^\dagger zz^T P + \|v\|^2 zz^T M^\dagger$$

$$- \frac{1}{\gamma}\|v\|^2 zz^T M^\dagger zz^T P - \frac{1}{\gamma}\|v\|^2 zz^T Pzz^T M^\dagger$$

$$+ \frac{(1 + \|v\|^2\tau)}{\gamma^2}zz^T Pzz^T P$$

Using the fact that $z^T Pz = \gamma$, $z^T M^\dagger z = \tau$ and cancelling, we get

$$(M + \|v\|^2 zz^T)(M + \|v\|^2 zz^T)^\dagger = MM^\dagger - \frac{1}{\gamma}MM^\dagger zz^T P$$

$$- \frac{1}{\gamma}\|v\|^2 zz^T M^\dagger zz^T P + \frac{(1 + \|v\|^2\tau)}{\gamma}zz^T P$$

$$= MM^\dagger - \frac{1}{\gamma}MM^\dagger zz^T P$$

$$- \frac{1}{\gamma}\|v\|^2 z\tau z^T P + \frac{(1 + \|v\|^2\tau)}{\gamma}zz^T P$$

$$= MM^\dagger - \frac{1}{\gamma}MM^\dagger zz^T P + \frac{1}{\gamma}zz^T P$$

$$= MM^\dagger + \left[(I - MM^\dagger)\frac{1}{\gamma}zz^T P\right]$$

$$= MM^\dagger + \frac{1}{\gamma}Pzz^T P$$

Thus, the first two terms in the error are

$$\frac{\pi_1}{d}\|\beta - \hat{\beta}\|^2 + \pi_2\|\beta^T z - \hat{\beta}^T z\|^2$$

Let us look at the second term first. Here we see that

$$\hat{\beta}^T z = \beta^T\left(MM^\dagger + \frac{1}{\gamma}Pzz^T P\right)z = \beta^T(MM^\dagger z + Pz)$$

Thus, we see that

$$\beta^T z - \hat{\beta}^T z = \beta^T z - \beta^T(MM^\dagger z + Pz)$$
$$= \beta^T(I - MM^\dagger)z - \beta^T(I - MM^\dagger)z$$
$$= 0$$

For the first term, we first note that

$$\beta^T - \hat{\beta}^T = \beta^T - \beta^T\left(MM^\dagger + \frac{1}{\gamma}Pzz^T P\right)$$

$$= \beta^T P - \frac{1}{\gamma}\beta^T Pzz^T P$$

To compute the norm, we expand and get

$$\|\beta^T P\|^2 + \frac{1}{\gamma^2}\operatorname{Tr}\left(\beta^T Pzz^T PPzz^T P\beta\right) - \frac{2}{\gamma}\operatorname{Tr}\left(\beta^T Pzz^T PP\beta\right)$$

Noting that $P$ is a projection matrix, so $PP = P$ and using $z^T P Z = \gamma$, we get that this term simplifies to

$$\|\beta^T P\|^2 - \frac{1}{\gamma}\operatorname{Tr}\left(\beta^T P z z^T P \beta\right)$$

Note that the subspace for $P$ comes from a Gaussian random matrix. Hence is uniformly random. Hence

$$\mathbb{E}\left[\|\beta^T P\|^2\right] = \left(1 - \frac{n-k}{d}\right)\|\beta\|^2 = \left(1 - \frac{\pi_1}{c}\right)\|\beta\|^2$$

Similarly,

$$\mathbb{E}\left[-\frac{1}{\gamma}\operatorname{Tr}\left(\beta^T P z z^T P \beta\right)\right] = \left(1 - \frac{\pi_1}{c}\right)\frac{(\beta^T z)^2}{\|z\|^2}$$

For the next two terms, we need to first simplify

$$\begin{bmatrix} A^T \\ v z^T \end{bmatrix}(A A^T + \|v\|^2 z z^T)^\dagger$$

Substituting in our formulas and noticing that

$$A^T P = 0 \text{ and } A^T M^\dagger = A^\dagger$$

we get that

$$\begin{bmatrix} A^T \\ v z^T \end{bmatrix}(A A^T + \|v\|^2 z z^T)^\dagger = \begin{bmatrix} A^\dagger - \frac{1}{\gamma}A^\dagger z z^T P - 0 + 0 \\ v z^T M^\dagger - \frac{1}{\gamma}\tau v z^T P - v z^T M^\dagger + \frac{1+\|v\|^2\tau)}{\|v\|^2\gamma}v z^T P \end{bmatrix}$$

$$= \begin{bmatrix} A^\dagger - \frac{1}{\gamma}A^\dagger z z^T P \\ -\frac{1}{\gamma}\tau v z^T P + \frac{1}{\gamma}\tau v z^T P + \frac{1}{\|v\|^2\gamma}v z^T P \end{bmatrix}$$

$$= \begin{bmatrix} A^\dagger - \frac{1}{\gamma}A^\dagger z z^T P \\ \frac{1}{\|v\|^2\gamma}v z^T P \end{bmatrix}$$

Then we have that

$$\mathbb{E}\left[\left\|\xi^T\begin{bmatrix} A^T \\ v z^T \end{bmatrix}(A A^T + \|v\|^2 z z^T)^\dagger\right\|^2\right] = \left\|\begin{bmatrix} A^T \\ v z^T \end{bmatrix}(A A^T + \|v\|^2 z z^T)^\dagger\right\|^2$$

$$= \operatorname{Tr}\left(\begin{bmatrix} A^\dagger - \frac{1}{\gamma}A^\dagger z z^T P \\ \frac{1}{\|v\|^2\gamma}v z^T P \end{bmatrix}^T\begin{bmatrix} A^\dagger - \frac{1}{\gamma}A^\dagger z z^T P \\ \frac{1}{\|v\|^2\gamma}v z^T P \end{bmatrix}\right)$$

$$= \operatorname{Tr}\left(M^\dagger - \frac{1}{\gamma}m^\dagger z z^T P - \frac{1}{\gamma}P z z^t M^\dagger + \frac{1+\|v\|^2\tau}{\|v\|^2\gamma^2}P z z^T P\right)$$

$$= \operatorname{Tr}\left(M^\dagger\right) - \operatorname{Tr}\left(\frac{1}{\gamma}M^\dagger z z^T P\right) - \operatorname{Tr}\left(\frac{1}{\gamma}P z z^T M^\dagger\right) + \operatorname{Tr}\left(\frac{1+\|v\|^2\tau}{\|v\|^2\gamma^2}P z z^T P\right)$$

Using the cycling invariance of the trace and the fact that $M^\dagger P = 0$, we get that

$$\mathbb{E}\left[\left\|\xi^T\begin{bmatrix} A^T \\ v z^T \end{bmatrix}(A A^T + \|v\|^2 z z^T)^\dagger\right\|^2\right] = \operatorname{Tr}\left(M^\dagger\right) + \frac{1+\|v\|^2\tau}{\|v\|^2\gamma}$$

The last equality is obtained using cyclic invariance, the fact that $P^2 = P$, and $\gamma = z^T P z$. Using Lemma 6 for $p = n - k$ and $q = d$, we get that asymptotically,

$$\frac{\pi_1}{d}\operatorname{Tr}(M^\dagger) = \frac{\pi_1^2}{c - \pi_1}.$$

Note that we similarly get that asymptotically,

$$\frac{\pi_1}{d}\tau = \frac{1}{d}\frac{\pi_1^2}{c - \pi_1} \to 0.$$

Thus, the contribution to the risk from this term is $\frac{\pi_1^2}{c-\pi_1}$.

Finally, for the last term, we have that

$$\begin{bmatrix} A^T \\ vz^T \end{bmatrix} (AA^T + \|v\|^2 zz^T)^\dagger z = \begin{bmatrix} A^\dagger z - \frac{1}{\gamma} A^\dagger zz^T Pz \\ \frac{1}{\|v\|^2 \gamma} vz^T Pz \end{bmatrix}$$

$$= \begin{bmatrix} 0 \\ \frac{v}{\|v\|^2} \end{bmatrix}$$

Thus,

$$\mathbb{E}\left[\left\|\xi^T \begin{bmatrix} A^T \\ vz^T \end{bmatrix} (AA^T + \|v\|^2 zz^T)^\dagger z\right\|^2\right] = \frac{1}{\|v\|^2}$$

Note that $\|v\|^2 \approx k$ and hence this term is aympotitcally zero. Putting it all together, we get the needed result. $\qquad\square$

### E.3 Proofs for Theorem 1

The proof structure closely follows that of [24].

#### E.3.1 Step 1: Decompose the error into bias and variance terms.

First, we decompose the error. Since we are not in the supervised learning setup, we do not have standard definitions of bias/variance. However, we will call the following terms the bias/variance of the model. First, we recall the following from [24].

**Lemma 1** (Sonthalia and Nadakuditi [24]). *If $A_{tst}$ has mean 0 entries and $A_{tst}$ is independent of $X_{tst}$ and $W$, then*

$$\mathbb{E}_{A_{tst}}[\|X_{tst} - WY_{tst}\|_F^2] = \underbrace{\mathbb{E}_{A_{tst}}[\|X_{tst} - WX_{tst}\|_F^2]}_{Bias} + \underbrace{\mathbb{E}_{A_{tst}}[\|WA_{tst}\|_F^2]}_{Variance}. \tag{3}$$

#### E.3.2 Step 2: Formula for $W_{opt}$

Here, we compute the explicit formula for $W_{opt}$ in Problem 1. Let $\hat{A}_{trn} = [A_{trn} \quad \mu I]$, $\hat{X}_{trn} = [X_{trn} \quad 0]$, and $\hat{Y}_{trn} = \hat{X}_{trn} + \hat{A}_{trn}$. Then solving $\arg\min_W \|X_{trn} - WY_{trn}\|_F^2 + \mu^2\|W\|_F^2$ is equivalent to solving $\arg\min_W \|\hat{X}_{trn} - W\hat{Y}_{trn}\|_F^2$. Thus, $W_{opt} = \arg\min_W \|\hat{X}_{trn} - W\hat{Y}_{trn}\|_F^2 = \hat{X}_{trn}\hat{Y}_{trn}^\dagger$. Expanding this out, we get the following formula for $\hat{W}$. Let $\hat{u}$ be the left singular vector and $\hat{v}_{trn}$ be the right singular vectors of $\hat{X}_{trn}$. Note that the left singular does not change after ridge regularization, so $\hat{u} = u$. Let $\hat{h} = \hat{v}_{trn}^T \hat{A}_{trn}^\dagger$, $\hat{k} = \hat{A}_{trn}^\dagger u$, $\hat{s} = (I - \hat{A}_{trn}\hat{A}_{trn}^\dagger)u$, $\hat{t} = \hat{v}_{trn}(I - \hat{A}_{trn}^\dagger\hat{A}_{trn})$, $\hat{\gamma} = 1 + \sigma_{trn}\hat{v}_{trn}^T \hat{A}_{trn}^\dagger u$, $\hat{\tau} = \sigma_{trn}^2\|\hat{t}\|^2\|\hat{k}\|^2 + \hat{\gamma}^2$.

**Proposition 2.** *If $\hat{\gamma} \neq 0$ and $A_{trn}$ has full rank then*

$$W_{opt} = \frac{\sigma_{trn}\hat{\gamma}}{\hat{\tau}} u\hat{h} + \frac{\sigma_{trn}^2\|\hat{t}\|^2}{\hat{\tau}} u\hat{k}^T \hat{A}_{trn}^\dagger.$$

*Proof.* Here we know that $u$ is arbitrary. We have that $\hat{A}_{trn}$ has full rank. Thus, the rank of $\hat{A}_{trn}$ is $d$, and the range of $\hat{A}_{trn}$ is the whole space. Thus, $u$ lives in the range of $\hat{A}_{trn}$. In this case, we want Theorem 3 from [47]. We define

$$\hat{p} = -\frac{\sigma_{trn}^2\|\hat{k}\|^2}{\hat{\gamma}}\hat{t}^T - \sigma_{trn}\hat{k} \text{ and } \hat{q}^T = -\frac{\sigma_{trn}\|\hat{t}\|^2}{\hat{\gamma}}\hat{k}^T \hat{A}_{trn}^\dagger - \hat{h}.$$

Then we have,

$$(\hat{A}_{trn} + \sigma_{trn}u\hat{v}_{trn}^T)^\dagger = \hat{A}_{trn}^\dagger + \frac{\sigma_{trn}}{\hat{\gamma}}\hat{t}^T\hat{k}^T \hat{A}_{trn}^\dagger - \frac{\hat{\gamma}}{\hat{\tau}}\hat{p}\hat{q}^T.$$

Note that, by our assumptions, we have $\hat{t} = \hat{v}_{trn}(I - \hat{A}_{trn}^\dagger\hat{A}_{trn})$, and $(I - \hat{A}_{trn}^\dagger\hat{A}_{trn})$ is a projection matrix, thus

$$\hat{v}_{trn}^T \hat{t}^T = \hat{v}_{trn}^T (I - \hat{A}_{trn}^\dagger \hat{A}_{trn})^T \hat{v}_{trn}^T$$
$$= \hat{v}_{trn}^T (I - \hat{A}_{trn}^\dagger \hat{A}_{trn})^T (I - \hat{A}_{trn}^\dagger \hat{A}_{trn})^T \hat{v}_{trn}^T.$$

To compute $W_{opt} = \hat{X}_{trn}(\hat{X}_{trn} + \hat{A}_{trn})^\dagger = \sigma_{trn} u \hat{v}_{trn}^T (\hat{A}_{trn} + \sigma_{trn} u \hat{v}_{trn}^T)^\dagger$, using $\hat{\gamma} - 1 = \sigma_{trn} \hat{v}_{trn}^T \hat{A}_{trn}^\dagger u = \sigma_{trn} \hat{h} u$, we multiply this through.

$$\sigma_{trn} u \hat{v}_{trn}^T (\hat{A}_{trn} + \sigma_{trn} u \hat{v}_{trn}^T)^\dagger = \sigma_{trn} u \hat{v}_{trn}^T \left( \hat{A}_{trn}^\dagger + \frac{\sigma_{trn}}{\hat{\gamma}} \hat{t}^T \hat{k}^T \hat{A}_{trn}^\dagger - \frac{\hat{\gamma}}{\hat{\tau}} \hat{p} \hat{q}^T \right)$$

$$= \sigma_{trn} u \hat{h} + \frac{\sigma_{trn}^2 \|\hat{t}\|^2}{\hat{\gamma}} u \hat{k}^T \hat{A}_{trn}^\dagger$$

$$+ \frac{\sigma_{trn} \hat{\gamma}}{\hat{\tau}} u \hat{v}_{trn}^T \left( \frac{\sigma_{trn}^2 \|\hat{k}\|^2}{\hat{\gamma}} \hat{t}^T + \sigma_{trn} \hat{k} \right) \hat{q}^T$$

$$= \sigma_{trn} u \hat{h} + \frac{\sigma_{trn}^2 \|\hat{t}\|^2}{\hat{\gamma}} u \hat{k}^T \hat{A}_{trn}^\dagger + \frac{\sigma_{trn}^3 \|\hat{k}\|^2 \|\hat{t}\|^2}{\hat{\tau}} u \hat{q}^T$$

$$+ \frac{\sigma_{trn} \hat{\gamma}(\hat{\gamma} - 1)}{\hat{\tau}} u \hat{q}^T.$$

Then we have,

$$\frac{\sigma_{trn}^3 \|\hat{k}\|^2 \|\hat{t}\|^2}{\hat{\tau}} u \hat{q}^T = \frac{\sigma_{trn}^3 \|\hat{k}\|^2 \|\hat{t}\|^2}{\hat{\tau}} u \left( -\frac{\sigma_{trn} \|\hat{t}\|^2}{\hat{\gamma}} \hat{k}^T \hat{A}_{trn}^\dagger - \hat{h} \right)$$

$$= -\frac{\sigma_{trn}^4 \|\hat{k}\|^2 \|\hat{t}\|^4}{\hat{\tau} \hat{\gamma}} u \hat{k}^T \hat{A}_{trn}^\dagger - \frac{\sigma_{trn}^3 \|\hat{k}\|^2 \|\hat{t}\|^2}{\hat{\tau}} u \hat{h}$$

and

$$\frac{\sigma_{trn} \hat{\gamma}(\hat{\gamma} - 1)}{\hat{\tau}} u \hat{q}^T = \frac{\sigma_{trn} \hat{\gamma}(\hat{\gamma} - 1)}{\hat{\tau}} u \left( -\frac{\sigma_{trn} \|\hat{t}\|^2}{\hat{\gamma}} \hat{k}^T \hat{A}_{trn}^\dagger - \hat{h} \right)$$

$$= -\frac{\sigma_{trn}^2 \|\hat{t}\|^2 (\hat{\gamma} - 1)}{\hat{\tau}} u \hat{k}^T \hat{A}_{trn}^\dagger - \frac{\sigma_{trn} \hat{\gamma}(\hat{\gamma} - 1)}{\hat{\tau}} u \hat{h}.$$

Substituting back in and collecting like terms, we get,

$$\sigma_{trn} u \hat{v}_{trn}^T (\hat{A}_{trn} + \sigma_{trn} u \hat{v}_{trn}^T)^\dagger = \sigma_{trn} \left( 1 - \frac{\sigma_{trn}^2 \|\hat{k}\|^2 \|\hat{t}\|^2}{\hat{\tau}} - \frac{\hat{\gamma}(\hat{\gamma} - 1)}{\hat{\tau}} \right) u \hat{h} +$$

$$\sigma_{trn}^2 \left( \frac{\|\hat{t}\|^2}{\hat{\gamma}} - \frac{\sigma_{trn}^2 \|\hat{k}\|^2 \|\hat{t}\|^4}{\hat{\tau} \hat{\gamma}} - \frac{\|\hat{t}\|^2 (\hat{\gamma} - 1)}{\hat{\tau}} \right) u \hat{k}^T \hat{A}_{trn}^\dagger.$$

We can then simplify the constants as follows.

$$1 - \frac{\sigma_{trn}^2 \|\hat{k}\|^2 \|\hat{t}\|^2}{\hat{\tau}} - \frac{\hat{\gamma}(\hat{\gamma} - 1)}{\hat{\tau}} = \frac{\hat{\tau} - \sigma_{trn}^2 \|\hat{k}\|^2 \|\hat{t}\|^2 - \gamma^2 + \gamma}{\hat{\tau}} = \frac{\hat{\gamma}}{\hat{\tau}}$$

and

$$\frac{\|\hat{t}\|^2}{\hat{\gamma}} - \frac{\sigma_{trn}^2 \|\hat{k}\|^2 \|\hat{t}\|^4}{\hat{\tau} \hat{\gamma}} - \frac{\|\hat{t}\|^2 (\hat{\gamma} - 1)}{\hat{\tau}} = \frac{\|\hat{t}\|^2 \left( \hat{\tau} - \sigma_{trn}^2 \|\hat{k}\|^2 \|\hat{t}\|^2 - \hat{\gamma}^2 + \hat{\gamma} \right)}{\hat{\tau} \hat{\gamma}} = \frac{\|\hat{t}\|^2}{\hat{\tau}}.$$

This gives us the result. $\qquad\square$

### E.3.3 Step 3: Decompose the terms into a sum of various trace terms.

For the bias and variance terms, we have the following two Lemmas.

**Lemma 2.** *If $W_{opt}$ is the solution to Equation 1, then*

$$X_{tst} - W_{opt}X_{tst} = \frac{\hat{\gamma}}{\hat{\tau}}X_{tst}.$$

*Proof.* To see this, note that we have $n + M > M$.

$$X_{tst} - W_{opt}X_{tst} = X_{tst} - \frac{\sigma_{trn}\hat{\gamma}}{\hat{\tau}}u\hat{h}uv_{tst}^T - \frac{\sigma_{trn}^2\|\hat{t}\|^2}{\hat{\tau}}u\hat{k}^T\hat{A}_{trn}^\dagger uv_{tst}^T$$

$$= X_{tst} - \frac{\hat{\sigma}_{trn}\hat{\gamma}}{\hat{\tau}}u\hat{v}_{trn}^T\hat{A}_{trn}^\dagger uv_{tst}^T - \frac{\sigma_{trn}^2\|\hat{t}\|^2}{\hat{\tau}}u\hat{k}^T\hat{A}_{trn}^\dagger uv_{tst}^T.$$

Note that $\hat{\gamma} = 1 + \sigma_{trn}\hat{v}_{trn}^T\hat{A}_{trn}^\dagger u$. Thus, we have that $\sigma_{trn}\hat{v}_{trn}^T\hat{A}_{trn}^\dagger u = \hat{\gamma} - 1$. Substituting this into the second term, we get,

$$X_{tst} - W_{opt}X_{tst} = X_{tst} - \frac{\hat{\gamma}(\hat{\gamma}-1)}{\hat{\tau}}uv_{tst}^T - \frac{\sigma_{trn}^2\|\hat{t}\|^2}{\hat{\tau}}u\hat{k}^T\hat{A}_{trn}^\dagger uv_{tst}^T.$$

For the third term, since $\hat{k} = \hat{A}_{trn}^\dagger u$, $\hat{k}^T\hat{A}_{trn}^\dagger u = \hat{k}^T\hat{k} = \|\hat{k}\|^2$. Substituting this into the expression, we get that

$$X_{tst} - W_{opt}X_{tst} = X_{tst} - \frac{\hat{\gamma}(\hat{\gamma}-1)}{\hat{\tau}}uv_{tst}^T - \frac{\sigma_{trn}^2\|\hat{t}\|^2\|\hat{k}\|^2}{\hat{\tau}}uv_{tst}^T.$$

Since $X_{tst} = uv_{tst}^T$, we get,

$$X_{tst} - W_{opt}X_{tst} = X_{tst}\left(1 - \frac{\hat{\gamma}(\hat{\gamma}-1)}{\hat{\tau}} - \frac{\sigma_{trn}^2\|\hat{t}\|^2\|\hat{k}\|^2}{\hat{\tau}}\right).$$

Simplify the constants using $\hat{\tau} = \sigma_{trn}^2\|\hat{t}\|^2\|\hat{k}\|^2 + \hat{\gamma}^2$, we get,

$$\frac{\hat{\tau} + \hat{\gamma} - \hat{\gamma}^2 - \sigma_{trn}^2\|\hat{t}\|^2\|\hat{k}\|^2}{\hat{\tau}} = \frac{\hat{\gamma}}{\hat{\tau}}.$$

$\square$

**Lemma 3** (Sonthalia and Nadakuditi [24]). *If the entries of $A_{tst}$ are independent with mean 0, and variance $1/d$, then we have that $\mathbb{E}_{A_{tst}}[\|W_{opt}A_{tst}\|^2] = \frac{N_{tst}}{d}\|W_{opt}\|^2$.*

**Lemma 4.** *If $\hat{\gamma} \neq 0$ and $A_{trn}$ has full rank, then we have that*

$$\|W_{opt}\|_F^2 = \frac{\sigma_{trn}^2\hat{\gamma}^2}{\tau^2}\operatorname{Tr}(\hat{h}^T\hat{h}) + 2\frac{\sigma_{trn}^3\|\hat{t}\|^2\hat{\gamma}}{\hat{\tau}^2}\operatorname{Tr}(\hat{h}^T\hat{k}^T\hat{A}_{trn}^\dagger) + \frac{\sigma_{trn}^4\|\hat{t}\|^4}{\hat{\tau}^2}\underbrace{\operatorname{Tr}((\hat{A}_{trn}^\dagger)^T\hat{k}\hat{k}^T\hat{A}_{trn}^\dagger)}_{\rho}.$$

*Proof.* We have

$$\|W_{opt}\|_F^2 = \operatorname{Tr}(W_{opt}^TW_{opt})$$

$$= \operatorname{Tr}\left(\left(\frac{\sigma_{trn}\hat{\gamma}}{\hat{\tau}}u\hat{h} + \frac{\sigma_{trn}^2\|\hat{t}\|^2}{\hat{\tau}}u\hat{k}^T\hat{A}_{trn}^\dagger\right)^T\left(\frac{\sigma_{trn}\hat{\gamma}}{\hat{\tau}}u\hat{h} + \frac{\sigma_{trn}^2\|\hat{t}\|^2}{\hat{\tau}}u\hat{k}^T\hat{A}_{trn}^\dagger\right)\right)$$

$$= \frac{\sigma_{trn}^2\hat{\gamma}^2}{\hat{\tau}^2}\operatorname{Tr}(\hat{h}^Tu^Tu\hat{h}) + 2\frac{\sigma_{trn}^3\|\hat{t}\|^2\hat{\gamma}}{\hat{\tau}^2}\operatorname{Tr}(\hat{h}^Tu^Tu\hat{k}^T\hat{A}_{trn}^\dagger)$$

$$+ \frac{\sigma_{trn}^4\|\hat{t}\|^4}{\hat{\tau}^2}\operatorname{Tr}((\hat{A}_{trn}^\dagger)^T\hat{k}u^Tu\hat{k}^T\hat{A}_{trn}^\dagger)$$

$$= \frac{\sigma_{trn}^2\hat{\gamma}^2}{\hat{\tau}^2}\operatorname{Tr}(\hat{h}^T\hat{h}) + 2\frac{\sigma_{trn}^3\|\hat{t}\|^2\hat{\gamma}}{\hat{\tau}^2}\operatorname{Tr}(\hat{h}^T\hat{k}^T\hat{A}_{trn}^\dagger) + \frac{\sigma_{trn}^4\|\hat{t}\|^4}{\hat{\tau}^2}\operatorname{Tr}((\hat{A}_{trn}^\dagger)^T\hat{k}\hat{k}^T\hat{A}_{trn}^\dagger).$$

Where the last inequality is true due to the fact that $\|u\|^2 = 1$. $\square$

### E.3.4 Step 4: Estimate With Random Matrix Theory

**Lemma 5.** *Let $A$ be a $p \times q$ matrix and let $\hat{A} = \begin{bmatrix} A & \mu I \end{bmatrix} \in \mathbb{R}^{p \times q + p}$. Suppose $A = U\Sigma V^T$ be the singular value decomposition of $A$. If $\hat{A} = \hat{U}\hat{\Sigma}\hat{V}^T$ is the singular value decomposition of $\hat{A}$, then $\hat{U} = U$ and if $p < q$*

$$\hat{\Sigma} = \begin{bmatrix} \sqrt{\sigma_1(A)^2 + \mu^2} & 0 & \cdots & 0 \\ 0 & \sqrt{\sigma_2(A)^2 + \mu^2} & & 0 \\ \vdots & & \ddots & \vdots \\ 0 & 0 & \cdots & \sqrt{\sigma_p(A)^2 + \mu^2} \end{bmatrix} \in \mathbb{R}^{p \times p},$$

*and*

$$\hat{V} = \begin{bmatrix} V_{1:p}\Sigma\hat{\Sigma}^{-1} \\ \mu U\hat{\Sigma}^{-1} \end{bmatrix} \in \mathbb{R}^{q + p \times p}.$$

*Here $V_{1:p}$ are the first $p$ columns of $V$.*

*Proof.* Since $p < q$, we have that $U \in \mathbb{R}^{p \times p}$, $\Sigma \in \mathbb{R}^{p \times p}$ are invertible. Here also consider the form of the SVD in which $V^T \in \mathbb{R}^{p \times q}$.

We start by nothing that $\hat{U}\hat{\Sigma}^2\hat{U}^T = \hat{A}\hat{A}^T = AA^T + \mu^2 I = U(\Sigma^2 + \mu^2 I_p)U^T$. Thus, we immediately see that $\sigma_i(\hat{A})^2 = \sigma_i(A)^2 + \mu^2$ and that $\hat{U} = U$.

Finally, we see,

$$\hat{V}^T = \hat{\Sigma}^{-1}U^T\hat{A} = \begin{bmatrix} \hat{\Sigma}^{-1}\Sigma V_{1:p}^T & \mu\hat{\Sigma}^{-1}U^T \end{bmatrix}$$

$\square$

**Lemma 6.** *Let $A$ be a $p \times q$ matrix and let $\hat{A} = \begin{bmatrix} A & \mu I \end{bmatrix} \in \mathbb{R}^{p \times q + p}$. Suppose $A = U\Sigma V^T$ be the singular value decomposition of $A$. If $\hat{A} = \hat{U}\hat{\Sigma}\hat{V}^T$ is the singular value decomposition of $\hat{A}$, then $\hat{U} = U$ and if $p > q$*

$$\hat{\Sigma} = \begin{bmatrix} \sqrt{\sigma_1(A)^2 + \mu^2} & 0 & \cdots & 0 & & \cdots & 0 \\ 0 & \sqrt{\sigma_2(A)^2 + \mu^2} & & 0 & & & \\ \vdots & & \ddots & \vdots & & & \vdots \\ 0 & 0 & \cdots & \sqrt{\sigma_q(A)^2 + \mu^2} & & & 0 \\ & & & & \mu & & \\ \vdots & & & & & \ddots & 0 \\ 0 & 0 & \cdots & 0 & \cdots & 0 & \mu \end{bmatrix} \in \mathbb{R}^{p \times p}.$$

*Here we will denote the upper left $q \times q$ block by $C$. Further,*

$$\hat{V} = \begin{bmatrix} V\Sigma_{1:q,1:q}^T C^{-1} & 0 \\ \mu U_{1:q}C^{-1} & U_{q+1:p} \end{bmatrix} \in \mathbb{R}^{q + p \times p}.$$

*Proof.* Since $p > q$, we have that $U \in \mathbb{R}^{p \times p}$ and we have that $\Sigma \in \mathbb{R}^{p \times q}$. Here $V^T \in \mathbb{R}^{q \times q}$ is invertible.

We start with nothing,

$$\hat{U}\hat{\Sigma}^2\hat{U}^T = \hat{A}\hat{A}^T = AA^T + \mu^2 I = U \left( \begin{bmatrix} \Sigma_{1:q,1:q}^2 & 0 \\ 0 & 0_{q-p} \end{bmatrix} + \mu^2 I_q \right) U^T.$$

Thus, we immediately see that for $i = 1, \ldots, p$ $\sigma_i(\hat{A})^2 = \sigma_i(A)^2 + \mu^2$ and for $i = p + 1, \ldots, q$, we have that $\sigma_i(\hat{A})^2 = \mu^2$ and that $\hat{U} = U$.

Then, we see,

$$\hat{V}^T = \hat{\Sigma}^{-1}U^T\hat{A} = \begin{bmatrix} \hat{\Sigma}^{-1}\Sigma V^T & \mu\hat{\Sigma}^{-1}U^T \end{bmatrix}.$$

Note that $\Sigma$ has 0 for the last $p - q$ entries. Thus,

$$\hat{\Sigma}^{-1}\Sigma V = \begin{bmatrix} C^{-1}\Sigma_{1:q,1:q}V \\ 0_{q-p,q} \end{bmatrix}.$$

Similarly, due to the structure of $\hat{\Sigma}$, we see,

$$\mu\hat{\Sigma}^{-1}U^T = [\mu C^{-1}U_{1:q}^T \quad \mu\frac{1}{\mu}U_{q+1:p}^T].$$

□

**Lemma 7.** *Suppose $A$ is an $p$ by $q$ matrix such that $p < q$, the entries of $A$ are independent and have mean 0, variance $1/p$, and bounded fourth moment. Let $c = p/q$. Let $\hat{A} = [A \quad \mu I] \in \mathbb{R}^{p \times q+p}$. Let $W_p = \hat{A}\hat{A}^T$ and let $W_q = \hat{A}^T\hat{A}$. Suppose $\lambda_p$ is a random non-zero eigenvalue from the largest $p$ eigenvalues of $W_p$, and $\lambda_q$ is a random non-zero eigenvalue of $W_q$. Then*

1. $\mathbb{E}\left[\frac{1}{\lambda_p}\right] = \mathbb{E}\left[\frac{1}{\lambda_q}\right] = \frac{\sqrt{(1+\mu^2c-c)^2+4\mu^2c^2}-1-\mu^2c+c}{2\mu^2c} + o(1).$

2. $\mathbb{E}\left[\frac{1}{\lambda_p^2}\right] = \mathbb{E}\left[\frac{1}{\lambda_q^2}\right] = \frac{\mu^2c^2+c^2+\mu^2c-2c+1}{2\mu^4c\sqrt{4\mu^2c^2+(1-c+\mu^2c)^2}} + \frac{1}{2\mu^4}\left(1-\frac{1}{c}\right) + o(1).$

*Proof.* First, we note that the non-zero eigenvalues of $W_p$ and $W_q$ are the same. Hence we focus on $W_p$. $W_p$ is nearly a Wishart matrix but is not normalized by the correct value. However, $cW_p$ does have the correct normalization.

Due to the assumptions on $A$, we have that the eigenvalues of $cAA^T$ converge to the Marchenko-Pastur. Hence since the eigenvalues of $cW_p$ are

$$(c\lambda_p)_i = c\sigma_i(A)^2 + c\mu^2,$$

we can estimate them by estimating $c\sigma_i(A)^2$ with the Marchenko-Pastur [44, 50–53]. In particular, we want the expectation of the inverse. We need to use the Stieljes transform. We know that if $m_c(z)$ is the Stieljes transform for the Marchenko-Pastur with shape parameter $c$, then if $\lambda$ is sampled from the Marchenko-Pastur distribution, then

$$m_c(z) = \mathbb{E}_\lambda\left[\frac{1}{\lambda - z}\right].$$

Thus, we have that the expected inverse of the eigenvalue can be approximated $m(-c\mu^2)$. We know that the Steiljes transform:

$$m_c(z) = -\frac{1-z-c-\sqrt{(1-z-c)^2-4cz}}{-2zc}.$$

Thus, we have,

$$\mathbb{E}\left[\frac{1}{c\lambda_p}\right] = m(-c\mu^2) = \frac{\sqrt{(1+\mu^2c-c)^2+4\mu^2c^2}-1-\mu^2c+c}{2\mu^2c^2}.$$

Canceling $1/c$ from both sides, we get,

$$\mathbb{E}\left[\frac{1}{\lambda_p}\right] = \frac{\sqrt{(1+\mu^2c-c)^2+4\mu^2c^2}-1-\mu^2c+c}{2\mu^2c}.$$

Then for the estimate of $\mathbb{E}\left[1/\lambda_p^2\right]$, we need to compute the derivative of the $m_c(z)$ and evaluate it at $-c\mu^2$. Hence, we see,

$$m_c'(z) = \frac{(c-z+\sqrt{-4cz+(1-c-z)^2}-1)(c+z+\sqrt{-4cz+(1-c-z)^2}-1)}{4cz^2\sqrt{-4cz+(1-c-z)^2}}.$$

Thus,

$$\mathbb{E}\left[\frac{1}{c^2\lambda_p^2}\right] = m_c'(-c\mu^2)$$

$$= \frac{(c+\mu^2c+\sqrt{4\mu^2c^2+(1-c+\mu^2c)^2}-1)(c-\mu^2c+\sqrt{4\mu^2c^2+(1-c+\mu^2c)^2}-1)}{4\mu^4c^3\sqrt{4\mu^2c^2+(1-c+\mu^2c)^2}}.$$

Canceling the $1/c^2$ from both sides, we get,

$$\mathbb{E}\left[\frac{1}{\lambda_p^2}\right] = \frac{(c + \mu^2 c + \sqrt{4\mu^2 c^2 + (1 - c + \mu^2 c)^2} - 1)(c - \mu^2 c + \sqrt{4\mu^2 c^2 + (1 - c + \mu^2 c)^2} - 1)}{4\mu^4 c \sqrt{4\mu^2 c^2 + (1 - c + \mu^2 c)^2}}.$$

Multiplying out and simplifying

$$\mathbb{E}\left[\frac{1}{\lambda_p^2}\right] = \frac{\mu^2 c^2 + c^2 + \mu^2 c - 2c + 1}{2\mu^4 c \sqrt{4\mu^2 c^2 + (1 - c + \mu^2 c)^2}} + \frac{1}{2\mu^4}\left(1 - \frac{1}{c}\right).$$

$\square$

**Lemma 8.** *Suppose $A$ is an $p$ by $q$ matrix such that $p > q$, the entries of $A$ are independent and have mean 0, variance $1/p$, and bounded fourth moment. Let $c = p/q$. Let $\hat{A} = [A \quad \mu I] \in \mathbb{R}^{p \times q + p}$. Let $W_p = \hat{A}\hat{A}^T$ and let $W_q = \hat{A}^T \hat{A}$. Suppose $\lambda_p$ is a random non-zero eigenvalue of $W_p$, and $\lambda_q$ is a random eigenvalue from the largest $q$ eigenvalues of $W_q$. Then*

*1.* $\mathbb{E}\left[\frac{1}{\lambda_q}\right] = \mathbb{E}\left[\frac{1}{\lambda_p}\right] = \frac{\sqrt{4\mu^2 c + (-1 + c + \mu^2 c)^2} - c - \mu^2 c + 1}{2\mu^2} + o(1).$

*2.* $\mathbb{E}\left[\frac{1}{\lambda_q^2}\right] = \mathbb{E}\left[\frac{1}{\lambda_p^2}\right] = \frac{1 - 2c + c^2 + \mu^2 c + \mu^2 c^2}{2\mu^4 \sqrt{4\mu^2 c + (-1 + c + \mu^2 c)^2}} + (1 - c)\frac{1}{2\mu^4} + o(1).$

*Proof.* First, we note that the non-zero eigenvalues of $W_p$ and $W_q$ are the same. Hence we focus on $W_p$. Due to the assumptions on $A$, we have that the eigenvalues of $A^T A$ converge to the Marchenko-Pastur with shape $c^{-1}$. Hence if $\lambda_p$ is one of the first $q$ eigenvalues of $W_p$, we see,

$$\mathbb{E}\left[\frac{1}{\lambda_p}\right] = m_{c^{-1}}(\mu^2) = \frac{\sqrt{(1 + \mu^2 - 1/c)^2 + 4\mu^2/c} - 1 - \mu^2 + 1/c}{2\mu^2/c}.$$

Then for the estimate of $\mathbb{E}\left[1/\lambda_p^2\right]$, we need to compute the derivative of the $m_{c^{-1}}(z)$ and evaluate it at $-\mu^2$. Hence, we see,

$$\mathbb{E}\left[\frac{1}{\lambda_p^2}\right] = \frac{(1/c + \mu^2 + \sqrt{4\mu^2/c + (1 - 1/c + \mu^2)^2} - 1)(1/c - \mu^2 + \sqrt{4\mu^2/c + (1 - 1/c + \mu^2)^2} - 1)}{4\mu^4/c \sqrt{4\mu^2/c + (1 - 1/c + \mu^2)^2}}$$

$$= \frac{(1 + \mu^2 c + c\sqrt{4\mu^2/c + (1 - 1/c + \mu^2)^2} - c)(1 - \mu^2 c + c\sqrt{4\mu^2/c + (1 - 1/c + \mu^2)^2} - c)}{4\mu^4 c \sqrt{4\mu^2/c + (1 - 1/c + \mu^2)^2}}$$

$$= \frac{(1 + \mu^2 c + \sqrt{4\mu^2 c + (-1 + c + \mu^2 c)^2} - c)(1 - \mu^2 c + \sqrt{4\mu^2 c + (-1 + c + \mu^2 c)^2} - c)}{4\mu^4 \sqrt{4\mu^2 c + (-1 + c + \mu^2 c)^2}}$$

This can be further simplified to

$$\frac{1 - 2c + c^2 + \mu^2 c + \mu^2 c^2}{2\mu^4 \sqrt{4\mu^2 c + (-1 + c + \mu^2 c)^2}} + (1 - c)\frac{1}{2\mu^4} + o(1)$$

$\square$

We will also need to estimate some other terms.

**Lemma 9.** *Suppose $A$ is an $p$ by $q$ matrix such that the entries of $A$ are independent and have mean 0, variance $1/p$, and bounded fourth moment. Let $\hat{A} = [A \quad \mu I] \in \mathbb{R}^{p \times q + p}$. Let $W_p = \hat{A}\hat{A}^T$ and let $W_q = \hat{A}^T \hat{A}$. Suppose $\lambda_p, \lambda_q$ are random non-zero eigenvalues of $W_p, W_q$ from the largest $\min(p, q)$ eigenvalues of $W_p, W_q$. Then*

*1. If $p > q$,* $\mathbb{E}\left[\frac{\lambda_p - \mu^2}{\lambda_p}\right] = c\left(\frac{1}{2} + \frac{1 + \mu^2 c - \sqrt{(-1 + c + \mu^2 c)^2 + 4\mu^2 c}}{2c}\right) + o(1).$

*2. If $p < q$,* $\mathbb{E}\left[\frac{\lambda_q - \mu^2}{\lambda_q}\right] = \frac{1}{2} + \frac{1 + \mu^2 c - \sqrt{(1 - c + \mu^2 c)^2 + 4c^2 \mu^2}}{2c} + o(1).$

3. If $p > q$, $\mathbb{E}\left[\frac{\lambda_p - \mu^2}{\lambda_p^2}\right] = c\left(\frac{1+c+\mu^2 c}{2\sqrt{(-1+c+\mu^2 c)^2 + 4\mu^2 c}} - \frac{1}{2}\right) + o(1)$.

4. If $p < q$, $\mathbb{E}\left[\frac{\lambda_q - \mu^2}{\lambda_q^2}\right] = \frac{1+c+\mu^2 c}{2\sqrt{(1-c+c\mu^2)^2 + 4c^2\mu^2}} - \frac{1}{2} + o(1)$.

*Proof.* Notice that if $\lambda$ is an eigenvalue of $A$ (so unshifted).

$$\frac{\lambda}{\lambda + \mu^2} = 1 - \frac{\mu^2}{\lambda + \mu^2} \text{ and } \frac{\lambda}{(\lambda + \mu^2)^2} = \frac{1}{\lambda + \mu^2} - \frac{\mu^2}{(\lambda + \mu^2)^2}$$

Then use Lemmas 7, and 8 to finish the proof. $\qquad\square$

**Bounding the Variance.**

**Lemma 10.** *Let $\eta_n$ be a uniform measure on $n$ numbers $a_1, \ldots, a_n$ such that $\eta^n \to \eta$ weakly in probability. Then for any bounded continuous function $f$*

$$\frac{1}{n}\sum_{i=1}^{n-1} f(a_i) \to \mathbb{E}_{x\sim\eta}[f(x)].$$

*Proof.* Using weak convergence

$$\frac{1}{n}\sum_{i=1}^{n} f(a_i) \to \mathbb{E}_{x\sim\eta}[f(x)].$$

Then using the boundedness of $f$, we get,

$$\frac{1}{n}\sum_{i=1}^{n-1} f(a_i) - \frac{1}{n}\sum_{i=1}^{n} f(a_i) = -\frac{1}{n}f(a_n) \to 0.$$

$\qquad\square$

**Lemma 11.** *Let $\eta_n$ be a uniform measure on $n$ numbers $a_1, \ldots, a_n$ such that $\eta_n \to \eta$ weakly in probability. Let $s$ be a uniformly random unit vector in $\mathbb{R}^m$ independent of $\eta_n$. Suppose $n/m \to \zeta \in (0,1]$. Then for any bounded function $f$,*

$$\mathbb{E}_s\left[\sum_{i=1}^{n} s_i^2 f(a_i)\right] \to \zeta\mathbb{E}_{x\sim\eta}[f(x)]$$

*and*

$$\mathbb{E}_s\left[\left(\sum_{i=1}^{n} s_i^2 f(a_i)\right)^2\right] - \mathbb{E}_s\left[\sum_{i=1}^{n} s_i^2 f(a_i)\right]^2 \to 0.$$

*Proof.* The first limit comes directly from weak convergence.

For the second, notice,

$$\left(\sum_{i=1}^{n} s_i^2 f(a_i)\right)^2 = \sum_{i=1}^{n} s_i^4 f(a_i)^2 + \sum_{i\neq j} s_i^2 s_j^2 f(a_i) f(a_j) = \sum_{i=1}^{n} s_i^4 f(a_i)^2 + \sum_{i=1}^{n} s_i^2 f(a_i)\sum_{j\neq i} s_j^2 f(a_j).$$

Taking the expectation with respect to $s$ we get,

$$\mathbb{E}_s\left[\left(\sum_{i=1}^{n} s_i^2 f(a_i)\right)^2\right] = \frac{1}{m^2 + O(m)}\sum_{i=1}^{n} f(a_i)^2 + \frac{1}{m^2 + O(m)}\sum_{i=1}^{n} f(a_i)\sum_{j\neq i} f(a_j)$$

Then using Lemma 10 for any fixed $i$, we have,

$$\frac{1}{m}\sum_{j\neq i} f(a_j) \to \zeta\mathbb{E}_{x\sim\eta}[f(x)].$$

Thus, as $n \to \infty$, we have,

$$\mathbb{E}_s\left[\left(\sum_{i=1}^n s_i^2 f(a_i)\right)^2\right] \to \zeta^2 \mathbb{E}_{x\sim\eta}[f(x)]^2.$$

Then since

$$\mathbb{E}_s\left[\sum_{i=1}^n s_i^2 f(a_i)\right]^2 \to \zeta^2 \mathbb{E}_{x\sim\eta}[f(x)]^2.$$

Thus, the variance goes to zero. $\qquad\square$

The interpretation of the above Lemma is that the variance of the sum decays to zero as $m \to \infty$.

**Lemma 12.** *Suppose $A$ is an $p$ by $q$ matrix such that the entries of $A$ are independent and have mean 0, variance $1/p$, and bounded fourth moment. Let $\hat{A} = [A \quad \mu I] \in \mathbb{R}^{p \times q+p}$. Let $x \in \mathbb{R}^p$ and $\hat{y} \in \mathbb{R}^{p+q}$ be unit norm vectors such that $\hat{y}^T = [y^T \quad 0_p]$. Then*

1. *If $p < q$, then $\mathbb{E}[\text{Tr}(x^T(\hat{A}\hat{A}^T)^\dagger x] = \frac{\sqrt{(1-c+\mu^2 c)^2 + 4\mu^2 c^2} - 1 - \mu^2 c + c}{2\mu^2 c} + o(1).$*

2. *If $p > q$, then $\mathbb{E}[\text{Tr}(x^T(\hat{A}\hat{A}^T)^\dagger x] = \frac{\sqrt{(-1+c+\mu^2 c)^2 + 4\mu^2 c} - 1 - \mu^2 c + c}{2\mu^2 c} + o(1).$*

3. *If $p < q$, then $\mathbb{E}[\text{Tr}(\hat{y}^T(\hat{A}^T\hat{A})^\dagger \hat{y}] = c\left(\frac{1+c+\mu^2 c}{2\sqrt{(1-c+\mu^2 c)^2 + 4c^2\mu^2}} - \frac{1}{2}\right) + o(1).$*

4. *If $p > q$, then $\mathbb{E}[\text{Tr}(\hat{y}^T(\hat{A}^T\hat{A})^\dagger \hat{y}] = c\left(\frac{1+c+\mu^2 c}{2\sqrt{(-1+c+\mu^2 c)^2 + 4\mu^2 c}} - \frac{1}{2}\right) + o(1).$*

*The variance of each above is $o(1)$.*

*Proof.* Let us start with $p < q$.

Let $\hat{A} = \hat{U}\hat{\Sigma}\hat{V}^T$, where $\hat{\Sigma}$ is $p \times p$. Then we see,

$$(\hat{A}\hat{A}^T)^\dagger = \hat{U}\hat{\Sigma}^{-2}\hat{U}^T.$$

Where $\hat{U}$ is uniformly random. Thus similar to [24], we can use Lemma 7 to get,

$$\mathbb{E}[\text{Tr}(x^T(\hat{A}\hat{A}^T)^\dagger x] = \frac{\sqrt{(1 + \mu^2 c - c)^2 + 4\mu^2 c^2} - 1 - \mu^2 c + c}{2\mu^2 c} + o(1).$$

On the other hand, for $p > q$, we have that only the first $q$ eigenvalues have the expectation in Lemma 8 The other $p - q$ are equal to $\frac{1}{\mu^2}$. Thus, we see,

$$\mathbb{E}[\text{Tr}(x^T(\hat{A}\hat{A}^T)^\dagger x] = \frac{1}{c}\left(\frac{\sqrt{4\mu^2 c + (-1 + c + \mu^2 c)^2} - c - \mu^2 c + 1}{2\mu^2} + o(1)\right) + \left(1 - \frac{1}{c}\right)\frac{1}{\mu^2}$$

$$= \frac{\sqrt{4\mu^2 c + (-1 + c + \mu^2 c)^2} + c - \mu^2 c - 1}{2c\mu^2}.$$

Again let us first consider the case when $p < q$. Then we have,

$$(\hat{A}^T\hat{A})^\dagger = \hat{V}\hat{\Sigma}^{-2}\hat{V}^T = \begin{bmatrix} V_{1:p}\Sigma\hat{\Sigma}^{-1} \\ \mu U\hat{\Sigma}^{-1} \end{bmatrix} \hat{\Sigma}^{-2} \begin{bmatrix} \hat{\Sigma}^{-1}\Sigma V_{1:p}^T & \mu\hat{\Sigma}^{-1}U^T \end{bmatrix}.$$

Since $\hat{y}$ has zeros in the last $p$ coordinates, we see,

$$\hat{y}^T(\hat{A}^T\hat{A})^\dagger \hat{y} = y^T V_{1:p}\Sigma\hat{\Sigma}^{-4}\Sigma V_{1:p}^T y.$$

Thus, we can use Lemma 9 to estimate this as,

$$c\left(\frac{1 + c + \mu^2 c}{2\sqrt{(1 - c + c\mu^2)^2 + 4c^2\mu^2}} - \frac{1}{2}\right) + o(1).$$

The extra factor of $c$ comes from the sum of $p$ coordinates of a uniformly unit vector in $q$ dimensional space. And for $p > q$, we have that the estimate is

$$c\left(\frac{1+c+\mu^2 c}{2\sqrt{(1+\mu^2-1/c)^2+4\mu^2/c}} - \frac{1}{2}\right) + o(1).$$

For the variance term, use Lemma 11. For three of the cases, the limiting distribution is the Marchenko-Pastur distribution. For the other case, the limiting measure is a mixture of the Marchenko-Pastur and a dirac delta at $1/\mu^2$. □

The rest of the lemmas in this section are used to compute the mean and variance of the various terms that appear in the formula of $W_{opt}$.

**Lemma 13.** *We have that*

$$\mathbb{E}_{A_{trn}}\left[\|\hat{h}\|^2\right] = \begin{cases} c\left(\frac{1+c+\mu^2 c}{2\sqrt{(1-c+\mu^2 c)^2+4\mu^2 c^2}} - \frac{1}{2}\right) + o(1) & c < 1 \\ c\left(\frac{1+c+\mu^2 c}{2\sqrt{(-1+c+\mu^2 c)^2+4\mu^2 c}} - \frac{1}{2}\right) + o(1) & c > 1 \end{cases}$$

*and that* $\mathbb{V}(\|\hat{h}\|^2) = o(1)$.

*Proof.* Here we see that

$$\|\hat{h}\|^2 = \text{Tr}(\hat{v}_{trn}^T(\hat{A}_{trn}^T\hat{A}_{trn})^\dagger \hat{v}_{trn}^T).$$

Thus, using the Lemma 12 we get that if $c < 1$

$$\mathbb{E}[\|\hat{h}\|^2] = c\left(\frac{1+c+\mu^2 c}{2\sqrt{(1-c+\mu^2 c)^2+4\mu^2 c^2}} - \frac{1}{2}\right) + o(1)$$

and if $c > 1$

$$\mathbb{E}[\|\hat{h}\|^2] = c\left(\frac{1+c+\mu^2 c}{2\sqrt{(-1+c+\mu^2 c)^2+4\mu^2 c}} - \frac{1}{2}\right) + o(1).$$

□

**Lemma 14.** *We have*

$$\mathbb{E}_{A_{trn}}\left[\|\hat{k}\|^2\right] = \begin{cases} \frac{\sqrt{(1-c+\mu^2 c)^2+4\mu^2 c^2}-1-\mu^2 c+c}{2\mu^2 c} + o(1) & c < 1 \\ \frac{\sqrt{(-1+c+\mu^2 c)^2+4\mu^2 c}-1-\mu^2 c+c}{2\mu^2 c} + o(1) & c > 1 \end{cases}$$

*and that* $\mathbb{V}(\|\hat{k}\|^2) = o(1)$.

*Proof.* Since $\hat{k} = \hat{A}_{trn}^\dagger u$, we have that

$$\|\hat{k}\|^2 = \text{Tr}(u^T(\hat{A}_{trn}\hat{A}_{trn}^T)^\dagger u).$$

According to the Lemma 12, if $c < 1$

$$\mathbb{E}[\|\hat{k}\|^2] = \frac{\sqrt{(1-c+\mu^2 c)^2+4\mu^2 c^2}-1-\mu^2 c+c}{2\mu^2 c} + o(1)$$

and if $c > 1$

$$\mathbb{E}[\|\hat{k}\|^2] = \frac{\sqrt{(-1+c+\mu^2 c)^2+4\mu^2 c}-1-\mu^2 c+c}{2\mu^2 c} + o(1).$$

□

**Lemma 15.** *We have that*

$$\mathbb{E}_{A_{trn}}\left[\|\hat{t}\|^2\right] = \begin{cases} \frac{1}{2}\left(1-c-\mu^2 c+\sqrt{(1-c+\mu^2 c)^2+4c^2\mu^2}\right) + o(1) & c < 1 \\ \frac{1}{2}\left(1-c-\mu^2 c+\sqrt{(-1+c+\mu^2 c)^2+4\mu^2 c}\right) + o(1) & c > 1 \end{cases}$$

*and we have that* $\mathbb{V}(\|\hat{t}\|^2) = o(1)$

*Proof.* Here we see that $\hat{t} = \hat{v}_{trn}(I - \hat{A}_{trn}^{\dagger}\hat{A}_{trn})$. Thus, we see that

$$\|\hat{t}\|^2 = \|v_{trn}\|^2 - \hat{v}_{trn}^T\hat{A}_{trn}^{\dagger}\hat{A}_{trn}\hat{v}_{trn} = 1 - \hat{v}_{trn}^T\hat{A}_{trn}^{\dagger}\hat{A}_{trn}\hat{v}_{trn}.$$

If $\hat{V} \in \mathbb{R}^{p+q \times p+q}$, we have that

$$\hat{A}_{trn}^{\dagger}\hat{A}_{trn} = \hat{V}\begin{bmatrix} I_p & 0 \\ 0 & 0_q \end{bmatrix}\hat{V}^T.$$

Then if $p < q$ using Lemma 6 and the fact that the last $p$ coordinates of $\hat{v}_{trn}$ are 0, we see that

$$\hat{v}_{trn}^T\hat{A}_{trn}^{\dagger}\hat{A}_{trn}\hat{v}_{trn} = v_{trn}^T V_{1:p}\Sigma\hat{\Sigma}^{-2}\Sigma V_{1:p}^T v_{trn}.$$

Then using Lemma 9 to estimate the middle diagonal matrix, we get that

$$\mathbb{E}[\|\hat{t}\|^2] = 1 - c\left(\frac{1}{2} + \frac{1 + \mu^2 c - \sqrt{(1 + \mu^2 c - c)^2 + 4c^2\mu^2}}{2c}\right)$$

$$= \frac{1}{2}\left(1 - c - \mu^2 c + \sqrt{(1 - c + \mu^2 c)^2 + 4c^2\mu^2}\right) + o(1).$$

Similarly for $c > 1$, we have that

$$\mathbb{E}[\|\hat{t}\|^2] = 1 - \left(\frac{1}{2} + \frac{c + \mu^2 c - c\sqrt{(1 + \mu^2 - 1/c)^2 + 4\mu^2/c}}{2}\right) + o(1)$$

$$= \frac{1}{2}\left(1 - c - \mu^2 c + \sqrt{(-1 + c + \mu^2 c)^2 + 4\mu^2 c}\right) + o(1).$$

The variance of $\hat{A}_{trn}^{\dagger}\hat{A}_{trn}$ is also $o(1)$ using Lemma 11. $\qquad\square$

**Lemma 16.** *We have that* $\mathbb{E}_{A_{trn}}[\hat{\gamma}] = 1$ *and* $\mathbb{V}(\gamma) = O(\sigma_{trn}^2/d)$.

*Proof.* Noting that $\hat{A} = U\hat{\Sigma}\hat{V}^T$, we have that

$$\hat{\gamma} = 1 + \sigma_{trn}\hat{v}_{trn}^T\hat{A}_{trn}^{\dagger}u = 1 + \sigma_{trn}\sum_{i=1}^{\min(n,d)}\sigma_i(\hat{A})^{-1}\hat{a}_i b_i.$$

Here $\hat{a}^T = \hat{v}_{trn}^T\hat{V}$ and $b = U^T u$. $U$ is a uniformly random rotation matrix that is independent of $\hat{\Sigma}$ and $\hat{V}$. Thus, taking the expectation with respect to $A_{trn}$, we get that the expectation is equal to zero.

For the variance, let us first consider the case when $c < 1$. For this case, we have that

$$\hat{V} = \begin{bmatrix} V_{1:d}\Sigma\hat{\Sigma}^{-1} \\ \mu U\hat{\Sigma}^{-1} \end{bmatrix}.$$

Thus, letting $a^T = v_{trn}^T V_{1:d}$, we get that

$$\hat{\gamma} = 1 + \sum_{i=1}^{d}\frac{\sigma_i(A)}{\sigma_i^2(A) + \mu^2}a_i b_i.$$

Squaring and taking the expectation, we see that

$$\mathbb{E}[\gamma^2] = 1 + \frac{\sigma_{trn}^2}{n}\mathbb{E}_{\lambda\sim\mu_c}\left[\frac{\lambda}{(\lambda + \mu^2)^2}\right] + o\left(\frac{\sigma_{trn}^2}{n}\right).$$

Similarly for $c > 1$, we have that

$$\mathbb{E}[\gamma^2] = 1 + \frac{\sigma_{trn}^2}{d}\mathbb{E}_{\lambda\sim\mu_c}\left[\frac{\lambda}{(\lambda + \mu^2)^2}\right] + o\left(\frac{\sigma_{trn}^2}{d}\right).$$

$\qquad\square$

**Lemma 17.** *We have that*

$$\mathbb{E}\left[\mathrm{Tr}((\hat{A}_{trn}^\dagger)^T \hat{k}\hat{k}^T \hat{A}_{trn}^\dagger)\right] = \mathbb{E}\left[\rho\right] = \begin{cases} \frac{\mu^2 c^2 + c^2 + \mu^2 c - 2c + 1}{2\mu^4 c\sqrt{4\mu^2 c^2 + (1-c+\mu^2 c)^2}} + \frac{1}{2\mu^4}\left(1 - \frac{1}{c}\right) + o(1) & c < 1 \\ \frac{1 - 2c + c^2 + \mu^2 c + \mu^2 c^2}{2\mu^4 c\sqrt{4\mu^2 c + (-1+c+\mu^2 c)^2}} + \left(1 - \frac{1}{c}\right)\frac{1}{2\mu^4} + o(1) & c > 1 \end{cases}$$

*and that* $\mathbb{V}(\rho) = o(1)$.

*Proof.* Here we have that

$$\rho = \mathrm{Tr}(\hat{k}^T(\hat{A}_{trn}^T \hat{A}_{trn})^\dagger \hat{k}) = \mathrm{Tr}(u^T(\hat{A}_{trn}\hat{A}_{trn}^T)^\dagger(\hat{A}_{trn}\hat{A}_{trn}^T)^\dagger u).$$

We first notice that

$$(\hat{A}_{trn}\hat{A}_{trn}^T)^\dagger(\hat{A}_{trn}\hat{A}_{trn}^T)^\dagger = \hat{U}^T\hat{\Sigma}^2\hat{U}.$$

Thus using Lemmas 7 and 8, we see that if $c < 1$

$$\mathbb{E}[\rho] = \frac{\mu^2 c^2 + c^2 + \mu^2 c - 2c + 1}{2\mu^4 c\sqrt{4\mu^2 c^2 + (1-c+\mu^2 c)^2}} + \frac{1}{2\mu^4}\left(1 - \frac{1}{c}\right)$$

and if $c > 1$

$$\mathbb{E}[\rho] = \frac{1}{c}\left(\frac{1 - 2c + c^2 + \mu^2 c + \mu^2 c^2}{2\mu^4\sqrt{4\mu^2 c + (-1+c+\mu^2 c)^2}} + (1-c)\frac{1}{2\mu^4}\right) + \left(1 - \frac{1}{c}\right)\frac{1}{\mu^4}$$

$$= \frac{1 - 2c + c^2 + \mu^2 c + \mu^2 c^2}{2\mu^4 c\sqrt{4\mu^2 c + (-1+c+\mu^2 c)^2}} + \left(1 - \frac{1}{c}\right)\frac{1}{2\mu^4}.$$

The variance being $o(1)$ comes from Lemma 11 again. $\qquad\square$

**Lemma 18.** *We have that*

$$\mathbb{E}_{A_{trn}}\left[\mathrm{Tr}(\hat{h}^T \hat{k}^T \hat{A}_{trn}^\dagger)\right] = 0$$

*and the variance is* $o(1)$.

*Proof.* Letting $\hat{A} = U\hat{\Sigma}\hat{V}^T$, we get that

$$\mathrm{Tr}(\hat{h}^T \hat{k}^T \hat{A}^T) = u^T U\hat{\Sigma}^{-3}\hat{V}^T \hat{v}_{trn}^T.$$

Then again since $U$ is uniformly random and independent of $\hat{\Sigma}$ and $\hat{V}$, the expectation is equal to zero. The variance is computed similarly to Lemma 16. $\qquad\square$

### E.3.5 Step 5: Putting it together

**Lemma 19.** *We have that*

$$\mathbb{E}\left[\frac{\tau}{\sigma_{trn}^2}\right] = \begin{cases} \frac{1}{\sigma_{trn}^2} + \frac{1}{2}\left(1 + \mu^2 c + c - \sqrt{(1 - c + \mu^2 c)^2 + 4\mu^2 c^2}\right) + o(1) & c < 1 \\ \frac{1}{\sigma_{trn}^2} + \frac{1}{2}\left(1 + \mu^2 c + c - \sqrt{(-1 + c + \mu^2 c)^2 + 4\mu^2 c}\right) + o(1) & c > 1 \end{cases}$$

*and that* $\mathbb{V}(\tau/\sigma_{trn}^2) = o(1)$.

*Proof.* Using the fact that all of the quantities concentrate, we can use the previous estimates. Specifically, we use that

$$|\mathbb{E}[XY] - \mathbb{E}[X]\mathbb{E}[Y]| \leq \sqrt{\mathbb{V}[X]\mathbb{V}[Y]}.$$

Thus, since our variances decay, we can use the product of the expectations. Further,

$$|\mathbb{V}[XY]| = |\mathbb{V}[X]\mathbb{V}[Y] + \mathbb{E}[X]^2\mathbb{V}[Y] + \mathbb{E}[Y]^2\mathbb{V}[X] - 2\mathbb{E}[X]\mathbb{E}[Y]\mathrm{Cov}(X,Y) + \mathrm{Cov}(X^2,Y^2) - \mathrm{Cov}(X,Y)^2|$$

$$\leq |\mathbb{V}[X]\mathbb{V}[Y] + \mathbb{E}[X]^2\mathbb{V}[Y] + \mathbb{E}[Y]^2\mathbb{V}[X]| + 2|\mathbb{E}[X]\mathbb{E}[Y]|\sqrt{\mathbb{V}[X]\mathbb{V}[Y]} + |\mathbb{V}[X]\mathbb{V}[Y]| + |\sqrt{\mathbb{V}[X^2]\mathbb{V}[Y^2]}|.$$

Thus, since the variances individually go to 0, we see that the variance of the product also goes to 0. Then using Lemma 15 and 14, we have that if $c < 1$

$$\mathbb{E}\left[\|\hat{t}\|^2\|\hat{k}\|^2\right] = \frac{1}{2}\left(1 + \mu^2 c + c - \sqrt{(1 - c + \mu^2 c)^2 + 4\mu^2 c^2}\right) + o(1)$$

and $\mathbb{V}(\|\hat{t}\|^2 \|\hat{k}\|^2) = o(1)$. Then since

$$|\mathbb{V}[X + Y]| \leq |\mathbb{V}[X] + \mathbb{V}[Y]| + 2\sqrt{\mathbb{V}[X]\mathbb{V}[Y]}$$

we have that using Lemma 16, that if $c < 1$

$$\mathbb{E}\left[\frac{\tau}{\sigma_{trn}^2}\right] = \frac{1}{\sigma_{trn}^2} + \frac{1}{2}\left(1 + \mu^2 c + c - \sqrt{(1 - c + \mu^2 c)^2 + 4\mu^2 c^2}\right) + o(1)$$

and that that variance is $o(1)$. If $c > 1$

$$\mathbb{E}\left[\frac{\tau}{\sigma_{trn}^2}\right] = \frac{1}{\sigma_{trn}^2} + \frac{1}{2}\left(1 + \mu^2 c + c - \sqrt{(-1 + c + \mu^2 c)^2 + 4\mu^2 c}\right) + o(1).$$

$\square$

**Lemma 20.** *We have that*

$$\mathbb{E}_{A_{trn}}\left[\frac{1}{\sigma_{trn}^2}\|\hat{h}\|^2 + \|\hat{t}\|^4 \rho\right] = \begin{cases} \frac{c(1+\sigma_{trn}^{-2})}{2}\left(\frac{\mu^2 c + c + 1}{\sqrt{(1-c+\mu^2 c)^2 + 4\mu^2 c^2}} - 1\right) + o(1) & c < 1 \\ \frac{c(1+\sigma_{trn}^{-2})}{2}\left(\frac{\mu^2 c + c + 1}{\sqrt{(-1+c+\mu^2 c)^2 + 4\mu^2 c}} - 1\right) + o(1) & c > 1 \end{cases}$$

*and that the variance is $o(1)$.*

*Proof.* Similar to Lemma 19, we can multiply the expectations since the variances are small. For $c < 1$, simplifying, we get that

$$\mathbb{E}_{A_{trn}}\left[\frac{1}{\sigma_{trn}^2}\|\hat{h}\|^2 + \|\hat{t}\|^4 \rho\right] = \frac{c(1 + \sigma_{trn}^{-2})}{2}\left(\frac{\mu^2 c + c + 1}{\sqrt{(1 - c + \mu^2 c)^2 + 4\mu^2 c^2}} - 1\right) + o(1)$$

and if $c > 1$, we get that

$$\mathbb{E}_{A_{trn}}\left[\frac{1}{\sigma_{trn}^2}\|\hat{h}\|^2 + \|\hat{t}\|^4 \rho\right] = \frac{c(1 + \sigma_{trn}^{-2})}{2}\left(\frac{\mu^2 c + c + 1}{\sqrt{(-1 + c + \mu^2 c)^2 + 4\mu^2 c}} - 1\right) + o(1)$$

and the variance decays since the variances decay individually. $\square$

**Lemma 21.** *We have that*

$$\mathbb{E}_{A_{trn}}\left[\|W_{opt}\|_F^2\right] = \frac{\sigma_{trn}^4}{\tau^2}\begin{cases} \frac{c(1+\sigma_{trn}^{-2})}{2}\left(\frac{\mu^2 c + c + 1}{\sqrt{(1-c+\mu^2 c)^2 + 4\mu^2 c^2}} - 1\right) + o(1) & c < 1 \\ \frac{c(1+\sigma_{trn}^{-2})}{2}\left(\frac{\mu^2 c + c + 1}{\sqrt{(-1+c+\mu^2 c)^2 + 4\mu^2 c}} - 1\right) + o(1) & c > 1 \end{cases}$$

*and that $\mathbb{V}(\|W_{opt}\|_F^2) = o(1)$.*

*Proof.* Follows immediately from Lemmas 4, 17, 18, and 20. $\square$

**Theorem 1** (Generalization Error Formula). *Suppose the training data $X_{trn}$ and test data $X_{tst}$ satisfy Assumption 3 and the noise $A_{trn}, A_{tst}$ satisfy Assumption 4. Let $\mu$ be the regularization parameter. Then for the under-parameterized regime (i.e., $c < 1$) for the solution $W_{opt}$ to Problem 1, the generalization error or risk given by Equation 2 is given by*

$$\mathcal{R}(c, \mu) = \frac{c\sigma_{trn}^2(\sigma_{trn}^2 + 1))}{2d\tau^2}\frac{1 + c + \mu^2 c}{\sqrt{(1 - c + \mu^2 c)^2 + 4\mu^2 c^2}} - \tau^{-2}\frac{c\sigma_{trn}^2(\sigma_{trn}^2 + 1))}{2d} + \tau^{-2}\frac{\sigma_{tst}^2}{n_{tst}} + o\left(\frac{1}{d}\right),$$

*where*

$$\frac{1}{\tau} = \frac{2\|\beta^T u\|}{2 + \sigma_{trn}^2(1 + c + \mu^2 c - \sqrt{(1 - c + \mu^2 c) + 4\mu^2 c^2})}.$$

*Proof.* Rewriting $\frac{\hat{\gamma}^2}{\tau^2}$ as $\frac{\hat{\gamma}^2/\sigma_{trn}^4}{\tau^2/\sigma_{trn}^4}$, we can the concentration from Lemmas 16 and 19. Then using Lemma 21 we get the needed result. $\square$

**Theorem 6.** *For the over-parameterized case, we have that the generalization error is given by*

$$\mathcal{R}(c,\mu) = \tau^{-2}\left(\frac{\sigma_{tst}^2}{N_{tst}} + \frac{c\sigma_{trn}^2(\sigma_{trn}^2+1))}{2d}\left(\frac{1+c+\mu^2c}{\sqrt{(-1+c+\mu^2c)^2+4\mu^2c}}-1\right)\right)+o\left(\frac{1}{d}\right),$$

*where* $\tau^{-1} = \dfrac{2}{2+\sigma_{trn}^2(1+c+\mu^2c-\sqrt{(-1+c+\mu^2c)+4\mu^2c})}.$

*Proof.* Rewriting $\frac{\hat{\gamma}^2}{\tau^2}$ as $\frac{\hat{\gamma}^2/\sigma_{trn}^4}{\tau^2/\sigma_{trn}^4}$, we can the concentration from Lemmas 16 and 19. Then using Lemma 21 we get the needed result. $\qquad\square$

### E.4 Proof of Theorem 2

**Theorem 2** (Under-Parameterized Peak). *Let* $\mu \in \mathbb{R}_{>0}$, $\sigma_{trn}^2 = n = d/c$ *and* $\sigma_{tst}^2 = n_{tst}$, *and* $d$ *is sufficiently large, so that the error term* $o(1/d)$ *is small, then the risk* $\mathcal{R}(c)$ *from Theorem 1, as a function of* $c$, *has a local maximum in the under-parameterized regime at* $c = \frac{1}{1+\mu^2}$.

*Proof.* First, we compute the derivative of the risk. We do so using SymPy and get the following expression.

$$\frac{4c\left(4c\mu^2+(\mu^2-1)\left(c\mu^2-c+1\right)-(\mu^2+1)\sqrt{4c^2\mu^2+(c\mu^2-c+1)^2}\right)}{d\left(4c^2\mu^2+(c\mu^2-c+1)^2\right)\left(c\mu^2+c-\sqrt{4c^2\mu^2+(c\mu^2-c+1)^2}+1\right)^2}$$

$$+\frac{2c\left((\mu^2+1)\left(4c^2\mu^2+(c\mu^2-c+1)^2\right)-(4c\mu^2+(\mu^2-1)\left(c\mu^2-c+1\right))\left(c\mu^2+c+1\right)\right)}{d\left(4c^2\mu^2+(c\mu^2-c+1)^2\right)^{\frac{3}{2}}\left(c\mu^2+c-\sqrt{4c^2\mu^2+(c\mu^2-c+1)^2}+1\right)^2}$$

$$+\frac{2\left(\left(4c^2\mu^2+(c\mu^2-c+1)^2\right)^3\left(c\mu^2+c-\sqrt{4c^2\mu^2+(c\mu^2-c+1)^2}+1\right)^6\right)}{M\left(4c^2\mu^2+(c\mu^2-c+1)^2\right)^{\frac{7}{2}}\left(c\mu^2+c-\sqrt{4c^2\mu^2+(c\mu^2-c+1)^2}+1\right)^7}$$

We can then compute the limit as $c \to 0^+$. Again using SymPy we see that

$$\lim_{c\to 0^+}\frac{\partial}{\partial c}\mathcal{R}(c,\mu^2;\sigma_{trn}^2=d/c)=\frac{1}{d}>0.$$

Let

$$T(c,\mu) = c^2\mu^4+2c^2\mu^2+c^2+2c\mu^2-2c+1$$

To find the critical point, we shall write the derivative as one fraction of the form

$$\partial_c\mathcal{R}(c,\mu) = -2\frac{(c\mu^2+c-1)P(c,\mu,d,T)}{Q(c,\mu,d,T)}.$$

Here we see that

$$P(c,\mu,d,T) = c^2\mu^4+2c^2\mu^2+c^2+2c\mu^2+1-(c\mu+c+1)\sqrt{T}$$

and

$$Q(c,\mu,d,T) = d(c\mu^2+c+1-\sqrt{T})^2T^{3/2}$$

We can see that the numerator is zero at $c = (1+\mu^2)^{-1}$. To evaluate the denominator at this point, we first get that

$$T((1+\mu^2)^{-1},\mu^2) = \frac{4\mu^2}{\mu^2+1}<4$$

Thus, we see that the denominator is

$$d\left(2 - \frac{2\mu}{\sqrt{\mu^2 + 1}}\right)^2 \left(\frac{4\mu^2}{\mu^2 + 1}\right)^{3/2}$$

Since $T((1 + \mu^2)^{-1}, \mu^2 < 4$, we have that

$$2 - \frac{2\mu}{\sqrt{\mu^2 + 1}} > 0$$

Hence the denominator is non-zero. Thus, we see that $c = (1 + \mu^2)^{-1}$ is a critical point.

Next we want to show that this point is a local maximum. To do so, we compute the second derivative and evaluate at $c = (1 + \mu^2)^{-1}$. Using SymPy, we get that the value of the second derivative at this point is

$$\frac{(\mu^2 + 1)^4 \cdot (4\mu^3 + 3\mu - 4\mu^2\sqrt{\mu^2 + 1} - \sqrt{\mu^2 + 1})}{8d \cdot \mu^3 \cdot (\mu^2 - \mu\sqrt{\mu^2 + 1} + 1)^3}$$

To see that it is a maximum, we need to show that the above is negative. We begin by showing that the denominator is positive. Since $8d\mu^3 > 0$, we only need to look at the second term. Using arithmetic mean and geometric mean inequality, we see that

$$\mu\sqrt{\mu^2 + 1} = \sqrt{\mu^2(\mu^2 + 1)} \leq \frac{\mu^2 + \mu^2 + 1}{2} = \mu^2 + \frac{1}{2} < \mu^2 + 1$$

Hence the denominator is positive. To show that the numerator is negative, we have the following

$$4\mu^3 + 3\mu - 4\mu^2\sqrt{\mu^2 + 1} - \sqrt{\mu^2 + 1} < 0 \iff 4\mu^2 + 3 < (4\mu^2 + 1)\frac{\sqrt{\mu^2 + 1}}{\mu}$$

$$\iff 16\mu^4 + 9 + 24\mu^2 < (16\mu^4 + 1 + 8\mu^2) \cdot \left(1 + \frac{1}{\mu^2}\right)$$

$$\iff 16\mu^2 + 8 < \frac{1}{\mu^2}(16\mu^4 + 8\mu^2 + 1)$$

$$\iff 0 < \frac{1}{\mu^2}$$

Where we are allowed to square both sides because both quantities are non-negative. Thus, we see get the needed result. $\square$

### E.5 Proof of Theorem 3

**Theorem 3** ($\|W_{opt}\|_F$ Peak). *If $\sigma_{tst} = \sqrt{n_{tst}}$, $\sigma_{trn} = \sqrt{n}$ and $\mu$ is such that $p(\mu) < 0$, then for fixed $n$ that is sufficiently large enough, we have that $\mathbb{E}\left[\|W_{opt}\|_F\right]$ versus $c = d/n$ curve has a local maximum in the under-parameterized regime at $c = (\mu^2 + 1)^{-1}$.*

*Proof.* Here we note that the expression for the norm of $W_{opt}$ is given by Lemma 21. We follow the same proof structure as Theorem 2. Differentiating with respect to $c$, we see that the numerator is if the form

$$(c\mu^2 + c - 1)P(c, \mu, T)$$

When the denominator is

$$(c\mu^2 + c + 1 - \sqrt{(T)})^7 T^{7/2}$$

Where is as is in the proof of Theorem 2. Again at when $c = 1/(\mu^2 + 1)$. We see that the denominator is positive because $\sqrt{T} < 2$. Hence again, we have a critical point $c = (1 + \mu^2)^{-1}$. $\square$

### E.6 Proof of Theorem 5

**Theorem 5** (Training Error). *Let $\tau$ be as in Theorem 1. The training error for $c < 1$ is given by*

$$\mathbb{E}_{A_{trn}}[\|X_{trn} - W_{opt}(X_{trn} + A_{trn})\|_F^2] = \tau^{-2}\left(\sigma_{trn}^2(1 - c \cdot T_1) + \sigma_{trn}^4 T_2\right) + o(1),$$

*where* $T_1 = \dfrac{\mu^2}{2}\left(\dfrac{1+c+\mu^2 c}{\sqrt{(1-c+\mu^2 c)^2 + 4\mu^2 c^2}} - 1\right) + \dfrac{1}{2} + \dfrac{1+\mu^2 c - \sqrt{(1-c+\mu^2 c)^2 + 4c^2\mu^2}}{2c},$

*and*

$$T_2 = (\mu^2 c + c - 1 - \sqrt{(1-c+\mu^2 c)^2 + 4c^2\mu^2})^2 \left(\frac{\mu^2 c + c + 1}{2\sqrt{(1-c+\mu^2 c)^2 + 4c^2\mu^2}} + \frac{1}{2}\right).$$

*Proof.* Note that we have:

$$\mathbb{E}_{A_{trn}}\left[\frac{\|X_{trn} - W_{opt}Y_{trn}\|_F^2}{n}\right] = \frac{1}{n}\mathbb{E}_{A_{trn}}\left[\|X_{trn} - W_{opt}(X_{trn} + A_{trn}))\|_F^2\right]$$

$$= \frac{1}{n}\mathbb{E}[\|X_{trn} - W_{opt}X_{trn}\|^2] + \frac{1}{n}\mathbb{E}[\|W_{opt}A_{trn}\|^2]$$

$$+ \frac{2}{n}\mathbb{E}\left[\text{Tr}((X_{trn} - W_{opt}X_{trn})^T W_{opt}A_{trn})\right].$$

First, by Lemma 2, we have $X_{trn} - W_{opt}X_{trn} = \frac{\hat{\gamma}}{\hat{\tau}}X_{trn}$. Then, $\mathbb{E}[\|X_{trn} - W_{opt}X_{trn}\|^2] = \frac{\hat{\gamma}^2}{\hat{\tau}^2}\mathbb{E}[\|X_{trn}\|^2] = \frac{\hat{\gamma}^2 \sigma_{trn}^2}{\hat{\tau}^2}$. Then, let us look at the $\mathbb{E}_{A_{trn}}[\|W_{opt}A_{trn}\|_F^2]$ term.

$$\mathbb{E}_{A_{trn}}[\|W_{opt}A_{trn}\|_F^2] = \mathbb{E}[\text{Tr}(A_{trn}^T W_{opt}^T W_{opt}A_{trn})]$$

$$= \frac{\sigma_{trn}^2 \hat{\gamma}^2}{\hat{\tau}^2}\mathbb{E}[\text{Tr}(A_{trn}^T \hat{h}^T u^T u\hat{h}A_{trn})]$$

$$+ \frac{\sigma_{trn}^3 \hat{\gamma}\|\hat{t}\|^2}{\hat{\tau}^2}\mathbb{E}[\text{Tr}(A_{trn}^T \hat{h}^T u^T u\hat{k}^T \hat{A}_{trn}^\dagger A_{trn})]$$

$$+ \frac{\sigma_{trn}^3 \hat{\beta}\|\hat{t}\|^2}{\hat{\tau}^2}\mathbb{E}[\text{Tr}(A_{trn}^T (\hat{A}_{trn}^\dagger)^T \hat{k}u^T u\hat{h}A_{trn})]$$

$$+ \frac{\sigma_{trn}^4 \|\hat{t}\|^4}{\hat{\tau}^2}\mathbb{E}[\text{Tr}(A_{trn}^T (\hat{A}_{trn}^\dagger)^T \hat{k}u^T u\hat{k}^T \hat{A}_{trn}^\dagger A_{trn})]$$

$$= \frac{\sigma_{trn}^2 \hat{\gamma}^2}{\hat{\tau}^2}\mathbb{E}[\text{Tr}(\hat{h}A_{trn}A_{trn}^T \hat{h}^T)]$$

$$+ \frac{\sigma_{trn}^3 \hat{\gamma}\|\hat{t}\|^2}{\hat{\tau}^2}\mathbb{E}[\text{Tr}(\hat{k}^T \hat{A}_{trn}^\dagger A_{trn}A_{trn}^T \hat{h}^T)]$$

$$+ \frac{\sigma_{trn}^3 \hat{\gamma}\|\hat{t}\|^2}{\hat{\tau}^2}\mathbb{E}[\text{Tr}(\hat{h}A_{trn}A_{trn}^T (\hat{A}_{trn}^\dagger)^T \hat{k})]$$

$$+ \frac{\sigma_{trn}^4 \|\hat{t}\|^4}{\hat{\tau}^2}\mathbb{E}[\text{Tr}(\hat{k}^T \hat{A}_{trn}^\dagger A_{trn}A_{trn}^T (\hat{A}_{trn}^\dagger)^T \hat{k})]$$

$$= \frac{\sigma_{trn}^2 \hat{\gamma}^2}{\hat{\tau}^2}\mathbb{E}[\text{Tr}(\hat{v}_{trn}^T \hat{A}_{trn}^\dagger A_{trn}A_{trn}^T (\hat{A}_{trn}^\dagger)^T \hat{v}_{trn}^T)]$$

$$+ \frac{\sigma_{trn}^3 \hat{\gamma}\|\hat{t}\|^2}{\hat{\tau}^2}\mathbb{E}[\text{Tr}(u^T (\hat{A}_{trn}^\dagger)^T \hat{A}_{trn}^\dagger A_{trn}A_{trn}^T (\hat{A}_{trn}^\dagger)^T \hat{v}_{trn}^T)]$$

$$+ \frac{\sigma_{trn}^3 \hat{\gamma}\|\hat{t}\|^2}{\hat{\tau}^2}\mathbb{E}[\text{Tr}(\hat{v}_{trn}^T \hat{A}_{trn}^\dagger A_{trn}A_{trn}^T (\hat{A}_{trn}^\dagger)^T \hat{A}_{trn}^\dagger u)]$$

$$+ \frac{\sigma_{trn}^4 \|\hat{t}\|^4}{\hat{\tau}^2}\mathbb{E}[\text{Tr}(u^T (\hat{A}_{trn}^\dagger)^T \hat{A}_{trn}^\dagger A_{trn}A_{trn}^T (\hat{A}_{trn}^\dagger)^T \hat{A}_{trn}^\dagger u)]$$

$$= \frac{\sigma_{trn}^2 \hat{\gamma}^2}{\hat{\tau}^2}\mathbb{E}[\text{Tr}(\hat{v}_{trn}^T \hat{A}_{trn}^\dagger A_{trn}A_{trn}^T (\hat{A}_{trn}^\dagger)^T \hat{v}_{trn}^T)]$$

$$+ \frac{\sigma_{trn}^4 \|\hat{t}\|^4}{\hat{\tau}^2}\mathbb{E}[\text{Tr}(u^T (\hat{A}_{trn}^\dagger)^T \hat{A}_{trn}^\dagger A_{trn}A_{trn}^T (\hat{A}_{trn}^\dagger)^T \hat{A}_{trn}^\dagger u)].$$

Then, we look at the $\text{Tr}((X_{trn} - W_{opt}X_{trn})^T W_{opt} A_{trn})$ term. By Lemma 2, we have $X_{trn} - W_{opt}X_{trn} = \frac{\hat{\gamma}}{\hat{\tau}} X_{trn}$. Then,

$$
\begin{aligned}
\frac{\hat{\gamma}}{\hat{\tau}} \text{Tr}(X_{trn}^T W_{opt} A_{trn}) &= \frac{\hat{\gamma}}{\hat{\tau}} \text{Tr}\left( X_{trn}^T \left( \frac{\sigma_{trn}\hat{\gamma}}{\hat{\tau}} u\hat{h} + \frac{\sigma_{trn}^2 \|\hat{t}\|^2}{\hat{\tau}} u\hat{k}^T \hat{A}_{trn}^\dagger \right) A_{trn} \right) \\
&= \frac{\sigma_{trn}\hat{\gamma}^2}{\hat{\tau}^2} \text{Tr}\left( X_{trn}^T u\hat{h} A_{trn} \right) \\
&\quad + \frac{\sigma_{trn}^2 \hat{\gamma} \|\hat{t}\|^2}{\hat{\tau}^2} \text{Tr}\left( X_{trn}^T u\hat{k}^T \hat{A}_{trn}^\dagger A_{trn} \right) \\
&= \frac{\sigma_{trn}\hat{\gamma}^2}{\hat{\tau}^2} \text{Tr}\left( \sigma_{trn} v_{trn} \hat{v}_{trn}^T \hat{A}_{trn}^\dagger A_{trn} \right) \\
&\quad + \frac{\sigma_{trn}^2 \hat{\gamma} \|\hat{t}\|^2}{\hat{\tau}^2} \text{Tr}\left( \sigma_{trn} v_{trn} u^T (\hat{A}_{trn}^\dagger)^T \hat{A}_{trn}^\dagger A_{trn} \right) \\
&= \frac{\sigma_{trn}^2 \hat{\gamma}^2}{\hat{\tau}^2} \text{Tr}\left( \hat{v}_{trn}^T \hat{A}_{trn}^\dagger A_{trn} v_{trn} \right) \\
&\quad + \frac{\sigma_{trn}^3 \hat{\gamma} \|\hat{t}\|^2}{\hat{\tau}^2} \text{Tr}\left( u^T (\hat{A}_{trn}^\dagger)^T \hat{A}_{trn}^\dagger A_{trn} v_{trn} \right) \\
&= \frac{\sigma_{trn}^2 \hat{\gamma}^2}{\hat{\tau}^2} \text{Tr}\left( \hat{v}_{trn}^T \hat{A}_{trn}^\dagger A_{trn} v_{trn} \right).
\end{aligned}
$$

In conclusion, we have the training error:

$$
\begin{aligned}
\mathbb{E}_{A_{trn}} \left[ \frac{\|X_{trn} - W_{opt}Y_{trn}\|_F^2}{n} \right] &= \frac{\hat{\gamma}^2 \sigma_{trn}^2}{n\hat{\tau}^2} + \frac{\sigma_{trn}^2 \hat{\gamma}^2}{n\hat{\tau}^2} \mathbb{E}[\text{Tr}(\hat{v}_{trn}^T \hat{A}_{trn}^\dagger A_{trn} A_{trn}^T (\hat{A}_{trn}^\dagger)^T \hat{v}_{trn}^T)] \\
&\quad + \frac{\sigma_{trn}^4 \|\hat{t}\|^4}{n\hat{\tau}^2} \mathbb{E}[\text{Tr}(u^T (\hat{A}_{trn}^\dagger)^T \hat{A}_{trn}^\dagger A_{trn} A_{trn}^T (\hat{A}_{trn}^\dagger)^T \hat{A}_{trn}^\dagger u)] \\
&\quad + 2\frac{\sigma_{trn}^2 \hat{\gamma}^2}{n\hat{\tau}^2} \mathbb{E}\left[ \text{Tr}\left( \hat{v}_{trn}^T \hat{A}_{trn}^\dagger A_{trn} v_{trn} \right) \right].
\end{aligned}
$$

Now we estimate the above terms using random matrix theory. Here we focus on the $c < 1$ case. For $c < 1$, we note that
$$
\hat{A}_{trn}^\dagger A_{trn} A_{trn}^T (\hat{A}_{trn}^\dagger)^T = \hat{V}\hat{\Sigma}^{-1}\Sigma\Sigma^T \hat{\Sigma}^{-1} \hat{V}^T.
$$

Thus, for $c < 1$
$$
\hat{v}_{trn}^T \hat{A}_{trn}^\dagger A_{trn} A_{trn}^T (\hat{A}_{trn}^\dagger)^T \hat{v}_{trn} = \sum_{i=1}^{d} a_i^2 \frac{\sigma_i(A)^4}{(\sigma_i(A)^2 + \mu^2)^2}
$$

where $a^T = v_{trn}^T V_{1:d}$. Taking the expectation, and using Lemma 9 we get that

$$
\mathbb{E}_{A_{trn}} \left[ \hat{v}_{trn}^T \hat{A}_{trn}^\dagger A_{trn} A_{trn}^T (\hat{A}_{trn}^\dagger)^T \hat{v}_{trn} \right] =
$$
$$
c\left( \frac{1}{2} + \frac{1 + \mu^2 c - \sqrt{(1 - c + \mu^2 c)^2 + 4c^2\mu^2}}{2c} + \mu^2 \left( \frac{1 + c + \mu^2 c}{2\sqrt{(1 - c + c\mu^2)^2 + 4c^2\mu^2}} - \frac{1}{2} \right) \right) + o(1).
$$

Using Lemma 11, we see that the variance is $o(1)$. Similarly, we have that
$$
(\hat{A}_{trn}^\dagger)^T \hat{A}_{trn}^\dagger A_{trn} A_{trn}^T (\hat{A}_{trn}^\dagger)^T \hat{A}_{trn}^\dagger = U\hat{\Sigma}^{-2}\Sigma\Sigma^T \hat{\Sigma}^{-2}U^T.
$$

Thus, again, using a similar argument, we see that

$$
\mathbb{E}_{A_{trn}} \left[ \text{Tr}(u^T (\hat{A}_{trn}^\dagger)^T \hat{A}_{trn}^\dagger A_{trn} A_{trn}^T (\hat{A}_{trn}^\dagger)^T \hat{A}_{trn}^\dagger u) \right] = \frac{1 + c + \mu^2 c}{2\sqrt{(1 - c + c\mu^2)^2 + 4c^2\mu^2}} - \frac{1}{2} + o(1)
$$

and again using Lemma 11, the variance is $o(1)$. Finally,

$$
\hat{A}_{trn}^\dagger A_{trn} = \hat{V}\hat{\Sigma}^{-1}\Sigma V.
$$

Thus,

$$\text{Tr}(\hat{v}_{trn}^T \hat{A}_{trn}^\dagger A_{trn} v_{trn} = \sum_{i=1}^d a_i^2 \frac{\sigma_i(A)^2}{\sigma_i(A)^2 + \mu^2}.$$

Thus, using Lemma 9, we get that

$$\mathbb{E}_{A_{trn}}\left[\text{Tr}(\hat{v}_{trn}^T \hat{A}_{trn}^\dagger A_{trn} v_{trn}\right] = \frac{1}{2} + \frac{1 + \mu^2 c - \sqrt{(1-c+\mu^2 c)^2 + 4c^2\mu^2}}{2c} + o(1)$$

and using Lemma 11, the variance is $o(1)$. Then, similar to the proof of Theorem 1, we can simplify the above expression to get the final result. $\qquad\square$

## E.7 Proof of Proposition 1

**Proposition 1** (Optimal $\sigma_{trn}$). *The optimal value of $\sigma_{trn}^2$ for $c < 1$ is given by*

$$\sigma_{trn}^2 = \frac{\sigma_{tst}^2 d[2c(\mu^2+1)^2 - 2T(c\mu^2 + c + 1) + 2(c\mu^2 - 2c + 1)] + N_{tst}(\mu^2 c^2 + c^2 + 1 - T)}{N_{tst}(c^3(\mu^2+1)^2 - T(\mu^2 c^2 + c^2 - 1) - 2c^2 - 1)}.$$

*Proof.* Let $\sigma := \sigma_{trn}^2$ and

$$F = \tau^{-2}\left(\frac{\sigma_{tst}^2}{N_{tst}} + \frac{1}{d}(\sigma\|\hat{h}\|_2^2 + \sigma^2\|\hat{t}\|_2^4\rho)\right).$$

Notice that only $\tau$ is a function of $\sigma$, $\|\hat{h}\|_2^2$, $\|\hat{t}\|_2^2$, and $\|\hat{k}\|_2^2$ are all functions of $\mu$. Then

$$\begin{aligned}
\frac{\partial F}{\partial \sigma} &= \tau^{-2}\frac{1}{d}(\|\hat{h}\|_2^2 + 2\sigma\|\hat{t}\|_2^4\rho) - 2\tau^{-3}\frac{\partial \tau}{\partial \sigma}\left(\frac{\sigma_{tst}^2}{N_{tst}} + \frac{1}{d}\left(\sigma\|\hat{h}\|_2^2 + \sigma^2\|\hat{t}\|_2^4\rho\right)\right) \\
&= \tau^{-2}\frac{1}{d}(\|\hat{h}\|_2^2 + 2\sigma\|\hat{t}\|_2^4\rho) - 2\tau^{-3}\|\hat{t}\|_2^2\|\hat{k}\|_2^2\left(\frac{\sigma_{tst}^2}{N_{tst}} + \frac{1}{d}(\sigma\|\hat{h}\|_2^2 + \sigma^2\|\hat{t}\|_2^4\rho)\right) \\
&= \tau^{-2}\left(\frac{1}{d}(\|\hat{h}\|_2^2 + 2\sigma\|\hat{t}\|_2^4\rho) - 2\tau^{-1}\|\hat{t}\|_2^2\|\hat{k}\|_2^2\left(\frac{\sigma_{tst}^2}{N_{tst}} + \frac{1}{d}(\sigma\|\hat{h}\|_2^2 + \sigma^2\|\hat{t}\|_2^4\rho)\right)\right).
\end{aligned}$$

The optimal $\sigma^*$ satisfies $\frac{\partial F}{\partial \sigma}|_{\sigma=\sigma^*} = 0$. Thus, we can solve the equation

$$\tau^{-2} = 0 \quad \text{or} \quad \frac{1}{d}(\|\hat{h}\|_2^2 + 2\sigma\|\hat{t}\|_2^4\rho) - 2\tau^{-1}\|\hat{t}\|_2^2\|\hat{k}\|_2^2\left(\frac{\sigma_{tst}^2}{N_{tst}} + \frac{1}{d}(\sigma\|\hat{h}\|_2^2 + \sigma^2\|\hat{t}\|_2^4\rho)\right).$$

Let $\alpha := \|\hat{t}\|_2^2\|\hat{k}\|_2^2$, $\delta := d\frac{\sigma_{tst}^2}{N_{tst}}$. Then

$$\tau^{-2} = 0 \implies \sigma = -\frac{1}{\|t\|_2^2\|k\|_2^2}.$$

Notice that $\sigma < 0$ implies $\sigma_{trn}$ is an imaginary number, something we don't want. Thus, we look at the other expression.

$$\begin{aligned}
0 &= \frac{1}{d}(\|\hat{h}\|_2^2 + 2\sigma\|\hat{t}\|_2^4\rho) - 2\tau^{-1}\|\hat{t}\|_2^2\|k\|_2^2\left(\frac{\sigma_{tst}^2}{N_{tst}} + \frac{1}{d}(\sigma\|\hat{h}\|_2^2 + \sigma^2\|\hat{t}\|_2^4\rho)\right) \\
&= \frac{1}{d}(\|\hat{h}\|_2^2 + 2\sigma\|\hat{t}\|_2^4\rho) - 2\tau^{-1}\alpha\left(\frac{\delta}{d} + \frac{1}{d}(\sigma\|\hat{h}\|_2^2 + \sigma^2\|\hat{t}\|_2^4\rho)\right). && [\alpha = \|\hat{t}\|_2^2\|\hat{k}\|_2^2]
\end{aligned}$$

Then multiplying through by $d$ and $\tau$

$$
\begin{aligned}
0 &= (1 + \alpha\sigma)(\|\hat{h}\|_2^2 + 2\sigma\|\hat{t}\|_2^4\rho) - 2\alpha(\delta + \sigma\|\hat{h}\|_2^2 + \sigma^2\|\hat{t}\|_2^4\rho) && [\tau = 1 + \alpha\sigma] \\
&= \|\hat{h}\|_2^2 + 2\|\hat{t}\|_2^4\rho\sigma + \alpha\|\hat{h}\|_2^2\sigma + 2\alpha\|\hat{t}\|_2^4\rho\sigma^2 - 2\alpha\delta - 2\alpha\|\hat{h}\|_2^2\sigma - 2\alpha\|\hat{t}\|_2^4\rho\sigma^2 \\
&= \|\hat{h}\|_2^2 + 2\|\hat{t}\|_2^4\rho\sigma + \alpha\|\hat{h}\|_2^2\sigma - 2\alpha\delta - 2\alpha\|\hat{h}\|_2^2\sigma.
\end{aligned}
$$

Then solving for $\sigma$, we get that

$$
\sigma = \frac{2\alpha\delta - \|\hat{h}\|^2}{2\|t\|^4\rho - \alpha\|\hat{h}\|^2} = \frac{2d\|\hat{t}\|_2^2\|\hat{k}\|_2^2\sigma_{tst}^2 - \|\hat{h}\|^2 N_{tst}}{N_{tst}(2\|\hat{t}\|_2^4\rho - \|\hat{t}\|_2^2\|\hat{k}\|_2^2\|\hat{h}\|_2^2)}.
$$

Then we use the random matrix theory lemmas to estimate this quantity. $\qquad\square$

# F Proof of low-rank case

Similar to the proof in [37], we conducted the low rank case.

We begin by defining some notation. Let $\hat{X}_{trn} = U\Sigma_{trn}V_{trn}^T$. Here $U$ is $d \times r$ with $U^T U = I$, $\Sigma_{trn}$ is $r \times r$, and $V_{trn}$ is $r \times (d + N)$. All of the following matrices are full rank.

1. $\hat{X}_{trn}$ and $\hat{X}_{tst}$ is $d \times (d + N)$ with rank $r$. $\hat{X}_{trn} = [X_{trn} \quad 0]$
2. $\hat{X}_{trn} = U\Sigma V$, by the singular value decomposition. Let $\hat{X}_{tst} = UL$.
3. $U$ is $d \times r$ with $U^T U = I_{r \times r}$. $V^T$ is is $r \times (d + N)$.
4. $\Sigma_{trn}$ is $r \times r$, with rank $r$.
5. $\hat{A}_{trn} = [A_{trn} \quad \mu I]$.
6. $\hat{A}_{trn}$ is $d \times (N + d)$ with rank $d$.

7. $\hat{A}_{trn}^\dagger \hat{A}_{trn}$ is $(N + d) \times (N + d)$.
8. $H$ is $r \times (N + d)$, with rank $r$.
9. $K$ is $(N + d) \times r$, with rank $r$.
10. $Z$ is $r \times r$, with rank $r$.
11. $H_1$ is $r \times r$, with rank $r$.
12. $\hat{A}_{trn} = \eta_{trn}\hat{U}\hat{\Sigma}\hat{V}^T$.
13. $\hat{U}$ is $d \times d$ unitary.
14. $\hat{\Sigma}$ is $d \times d$.

For rank $r$ data and $r < N$, with $c = \frac{d}{N}$, the following is true.

1. We denote the minimum norm linear denoiser $W_{opt}$ by just $W$ in this subsection. It is given by
$$W_{opt} = -U\Sigma_{trn}H_1^{-1}K^T\hat{A}_{trn}^\dagger + U\Sigma_{trn}H_1^{-1}Z^T(QQ^T)^{-1}H$$

2. The test error when $X_{tst} = UL$ is given by
$$\mathbb{E}_{\hat{A}_{trn}}\left[\frac{1}{N_{tst}}\|U\Sigma_{trn}H_1^{-1}Z^T(QQ^T)^{-1}\Sigma_{trn}^{-1}L\|_F^2 + \frac{\sigma_{tst}^2}{d}\|W_{opt}\|_F^2\right],$$

where $Q = V^T(I - \hat{A}_{trn}^\dagger\hat{A}_{trn})$, $H = V_{trn}^T\hat{A}_{trn}^\dagger$,1 $K = -\hat{A}_{trn}^\dagger U\Sigma_{trn}$, $Z = I + V_{trn}^T\hat{A}_{trn}^\dagger U\Sigma_{trn}$, $H_1 = K^TK + Z^T(QQ^T)^{-1}Z$.

For $c < 1$, we have that if $d < N$ then
$$\mathbb{E}[\Sigma_{trn}^{-1}K^TK\Sigma_{trn}^{-1}] = \left(\frac{\sqrt{(1 + \mu^2c - c)^2 + 4\mu^2c^2} - 1 - \mu^2c + c}{2\mu^2c} + o(1)\right)I_r$$

and if $d > N$ then
$$\mathbb{E}[\Sigma_{trn}^{-1}K^TK\Sigma_{trn}^{-1}] = \left(\frac{\sqrt{4\mu^2c + (-1 + c + \mu^2c)^2} - 1 - \mu^2c + c}{2\mu^2c} + o(1)\right)I_r.$$

When $d < N$ then
$$\mathbb{E}[\Sigma_{trn}^{-1}K^T\hat{A}_{trn}^\dagger(\hat{A}_{trn}^\dagger)^TK\Sigma_{trn}^{-1}] = \frac{\mu^2c^2 + c^2 + \mu^2c - 2c + 1}{2\mu^4c\sqrt{4\mu^2c^2 + (1 - c + \mu^2c)^2}}I_r + \frac{1}{2\mu^4}\left(1 - \frac{1}{c}\right)I_r + o(1),$$

if $d > N$ then
$$\mathbb{E}[\Sigma_{trn}^{-1}K^T\hat{A}_{trn}^\dagger(\hat{A}_{trn}^\dagger)^TK\Sigma_{trn}^{-1}] = \frac{1 - 2c + c^2 + \mu^2c + \mu^2c^2}{2c\mu^4\sqrt{4\mu^2c + (-1 + c + \mu^2c)^2}}I_r + (1 - \frac{1}{c})\frac{1}{2\mu^4}I_r + o(1).$$

We have that

$$\mathbb{E}[QQ^T] = c\left(\frac{1}{2} + \frac{1 + \mu^2 c - \sqrt{(-1 + c + \mu^2 c)^2 + 4\mu^2 c}}{2c}\right) + o(1).$$

and

$$\mathbb{E}[(QQ^T)^{-1}] = \frac{2}{1 + \mu^2 c + c - \sqrt{(-1 + c + \mu^2 c)^2 + 4\mu^2 c}} + o(1).$$

When $d < N$ we have that

$$\mathbb{E}[HH^T] = c\left(\frac{1 + c + \mu^2 c}{2\sqrt{(1 - c + \mu^2 c)^2 + 4c^2\mu^2}} - \frac{1}{2}\right) I_r + o(1)$$

and when $d > N$, we have

$$\mathbb{E}[HH^T] = c\left(\frac{1 + c + \mu^2 c}{2\sqrt{(-1 + c + \mu^2 c)^2 + 4\mu^2 c}} - \frac{1}{2}\right) I_r + o(1).$$

When $d < N$, we have

$$\mathbb{E}[\|W\|_F^2] = \left(\frac{\mu^2 c^2 + c^2 + \mu^2 c - 2c + 1}{2\mu^4 c\sqrt{4\mu^2 c^2 + (1 - c + \mu^2 c)^2}} + \frac{1}{2\mu^4}\left(1 - \frac{1}{c}\right)\right)$$

$$\mathrm{Tr}\left(\left(\frac{\sqrt{(1 + \mu^2 c - c)^2 + 4\mu^2 c^2} - 1 - \mu^2 c + c}{2\mu^2 c} I_r + \frac{2}{1 + \mu^2 c + c - \sqrt{(-1 + c + \mu^2 c)^2 + 4\mu^2 c}}\Sigma^{-2}\right)^{-2}\right)$$

$$+ \left(\frac{2}{1 + \mu^2 c + c - \sqrt{(-1 + c + \mu^2 c)^2 + 4\mu^2 c}}\right)^2 c\left(\frac{1 + c + \mu^2 c}{2\sqrt{(1 - c + \mu^2 c)^2 + 4c^2\mu^2}} - \frac{1}{2}\right) \mathrm{Tr}(\Sigma^{-2})$$

$$\mathrm{Tr}\left(\left(\frac{\sqrt{(1 + \mu^2 c - c)^2 + 4\mu^2 c^2} - 1 - \mu^2 c + c}{2\mu^2 c} I_r + \frac{2}{1 + \mu^2 c + c - \sqrt{(-1 + c + \mu^2 c)^2 + 4\mu^2 c}}\Sigma^{-2}\right)^{-2}\right)$$

$$+ o(1),$$

when $d > N$ this is estimated by

$$\mathbb{E}[\|W\|_F^2] = \left(\frac{1 - 2c + c^2 + \mu^2 c + \mu^2 c^2}{2c\mu^4\sqrt{4\mu^2 c + (-1 + c + \mu^2 c)^2}} + (1 - \frac{1}{c})\frac{1}{2\mu^4}\right)$$

$$\mathrm{Tr}\left(\left(\frac{\sqrt{4\mu^2 c + (-1 + c + \mu^2 c)^2} - 1 - \mu^2 c + c}{2\mu^2 c} I_r + \frac{2}{1 + \mu^2 c + c - \sqrt{(-1 + c + \mu^2 c)^2 + 4\mu^2 c}}\Sigma^{-2}\right)^{-2}\right)$$

$$+ \left(\frac{2}{1 + \mu^2 c + c - \sqrt{(-1 + c + \mu^2 c)^2 + 4\mu^2 c}}\right)^2 c\left(\frac{1 + c + \mu^2 c}{2\sqrt{(-1 + c + \mu^2 c)^2 + 4\mu^2 c}} - \frac{1}{2}\right) \mathrm{Tr}(\Sigma^{-2})$$

$$\mathrm{Tr}\left(\left(\frac{\sqrt{4\mu^2 c + (-1 + c + \mu^2 c)^2} - 1 - \mu^2 c + c}{2\mu^2 c} I_r + \frac{2}{1 + \mu^2 c + c - \sqrt{(-1 + c + \mu^2 c)^2 + 4\mu^2 c}}\Sigma^{-2}\right)^{-2}\right)$$

$$+ o(1).$$

When $d < N$ the test error $\mathcal{R}(W, X_{tst})$ for $W = W_{opt}$ is given by

$$\mathcal{R}(W, X_{tst}) = \frac{1}{N_{tst}} \text{Tr}(( \frac{\sqrt{4\mu^2 c + (-1 + c + \mu^2 c)^2} - 1 - \mu^2 c + c}{2\mu^2 c} I_r$$

$$+ \frac{2}{1 + \mu^2 c + c - \sqrt{(-1 + c + \mu^2 c)^2 + 4\mu^2 c}} \Sigma^{-2})^{-2})$$

$$\frac{2}{1 + \mu^2 c + c - \sqrt{(-1 + c + \mu^2 c)^2 + 4\mu^2 c}} \text{Tr}(\Sigma^{-2})$$

$$+ \frac{\sigma_{tst}^2}{d} \left( \frac{\mu^2 c^2 + c^2 + \mu^2 c - 2c + 1}{2\mu^4 c \sqrt{4\mu^2 c^2 + (1 - c + \mu^2 c)^2}} + \frac{1}{2\mu^4}\left(1 - \frac{1}{c}\right) \right)$$

$$\text{Tr}\left( \left( \frac{\sqrt{(1 + \mu^2 c - c)^2 + 4\mu^2 c^2} - 1 - \mu^2 c + c}{2\mu^2 c} I_r + \frac{2}{1 + \mu^2 c + c - \sqrt{(-1 + c + \mu^2 c)^2 + 4\mu^2 c}} \Sigma^{-2} \right)^{-2} \right)$$

$$+ \frac{\sigma_{tst}^2}{d} \left( \frac{2}{1 + \mu^2 c + c - \sqrt{(-1 + c + \mu^2 c)^2 + 4\mu^2 c}} \right)^2 c \left( \frac{1 + c + \mu^2 c}{2\sqrt{(1 - c + \mu^2 c)^2 + 4c^2\mu^2}} - \frac{1}{2} \right) \text{Tr}(\Sigma^{-2})$$

$$\text{Tr}\left( \left( \frac{\sqrt{(1 + \mu^2 c - c)^2 + 4\mu^2 c^2} - 1 - \mu^2 c + c}{2\mu^2 c} I_r + \frac{2}{1 + \mu^2 c + c - \sqrt{(-1 + c + \mu^2 c)^2 + 4\mu^2 c}} \Sigma^{-2} \right)^{-2} \right)$$

$$+ o(1),$$

when $d > N$ this is estimated by

$$\mathcal{R}(W, X_{tst}) = \frac{1}{N_{tst}} \text{Tr}(( \frac{\sqrt{4\mu^2 c + (-1 + c + \mu^2 c)^2} - 1 - \mu^2 c + c}{2\mu^2 c} I_r$$

$$+ \frac{2}{1 + \mu^2 c + c - \sqrt{(-1 + c + \mu^2 c)^2 + 4\mu^2 c}} \Sigma^{-2})^{-2})$$

$$\frac{2}{1 + \mu^2 c + c - \sqrt{(-1 + c + \mu^2 c)^2 + 4\mu^2 c}} \text{Tr}(\Sigma^{-2})$$

$$+ \frac{\sigma_{tst}^2}{d} \left( \frac{1 - 2c + c^2 + \mu^2 c + \mu^2 c^2}{2c\mu^4 \sqrt{4\mu^2 c + (-1 + c + \mu^2 c)^2}} + (1 - \frac{1}{c})\frac{1}{2\mu^4} \right)$$

$$\text{Tr}\left( \left( \frac{\sqrt{4\mu^2 c + (-1 + c + \mu^2 c)^2} - 1 - \mu^2 c + c}{2\mu^2 c} I_r + \frac{2}{1 + \mu^2 c + c - \sqrt{(-1 + c + \mu^2 c)^2 + 4\mu^2 c}} \Sigma^{-2} \right)^{-2} \right)$$

$$+ \frac{\sigma_{tst}^2}{d} \left( \frac{2}{1 + \mu^2 c + c - \sqrt{(-1 + c + \mu^2 c)^2 + 4\mu^2 c}} \right)^2 c \left( \frac{1 + c + \mu^2 c}{2\sqrt{(-1 + c + \mu^2 c)^2 + 4\mu^2 c}} - \frac{1}{2} \right) \text{Tr}(\Sigma^{-2})$$

$$\text{Tr}\left( \left( \frac{\sqrt{4\mu^2 c + (-1 + c + \mu^2 c)^2} - 1 - \mu^2 c + c}{2\mu^2 c} I_r + \frac{2}{1 + \mu^2 c + c - \sqrt{(-1 + c + \mu^2 c)^2 + 4\mu^2 c}} \Sigma^{-2} \right)^{-2} \right)$$

$$+ o(1).$$

# G   Experiments

All experiments were conducted using Pytorch and run on Google Colab using an A100 GPU. For each empirical data point, we did at least 100 trials. The maximum number of trials for any experiment was 20000 trials.

For each configuration of the parameters, $N_{trn}, N_{tst}, d, \sigma_{trn}, \sigma_{tst}$, and $\mu$. For each trial, we sampled $u, v_{trn}, v_{tst}$ uniformly at random from the appropriate dimensional sphere. We also sampled new training and test noise for each trial.

For the data scaling regime, we kept $d = 1000$ and for the parameter scaling regime, we kept $N_{trn} = 1000$. For all experiments, $N_{tst} = 1000$.

