# OpenReview forum: "Least Squares Regression Can Exhibit Under-Parameterized Double Descent"
_NeurIPS.cc/2024/Conference — NeurIPS 2024 poster_

### Official Review · Reviewer_SLpr · 2024-07-04

**Soundness:** 4
**Presentation:** 2
**Contribution:** 3
**Rating:** 7
**Confidence:** 3

**Summary:**

The paper aims to understand the phenomenon of double descent in regression, and helps complement existing knowledge about the phenomenon by proving that double descent can occur even in the under-parametrized regime, going against previous intuition. They also prove that the peak in the norm of the estimator does not imply a peak in the risk.

**Strengths:**

Quality and Clarity: Contextualization relative to prior work is good, Table 1, for example, provides a very concise and insightful summary of existing results.

Originality: The paper considers an original viewpoint on the problem of double-descent, namely they realize that violating either Assumptions 1 or 2 can cause the peak to move into the under-parametrized region. To the best of my knowledge, this had not been noticed before.

Significance: This result is significant, as it addresses the prominent double-descent phenomenon in machine learning, which is an important primitive for understanding generalization and other properties of estimators operating on high-dimensional data. They also prove that a peak in the norm of the estimator does not imply a peak in risk, and this is important as it goes against some of the intuition provided in earlier works.

**Weaknesses:**

Clarity: Although the contextualization relative to prior work is good, I find that the paper lacks in clarity. In particular, it is not exceptionally well written, and leaves some sections with much to be desired in terms of exposition. Specific examples are:

- Table 1: What does 1/Low mean? The superscript 3 leads to nowhere.
- Middle of page four: “Hence, this is controlled by 1. The alignment…, 2. The spectrum”. What exactly is being controlled here? And why exactly is it controlled by 1. and 2. ? This is not immediate to me, and I think this needs to be made more precise.
- Over-use of italics in the introduction makes it hard to know what to focus on. I would recommend maximum one italicized sentence per paragraph.
- Section 4.1: be more precise about the model, I don’t know what it is at this point. I later identify that $X + A$ represents the spiked data, but this must be made clear earlier on.  I find“Let A be the noise matrix” to not be clear enough.
- Theorem 1: a comment on the proof technique for this theorem would be helpful, even if you have already mentioned it previously (or given the intuition). Also, the interpretation of Theorem 1 is not clear until we read Theorem 2, could you interpret Theorem 1 a bit more and specifically identify what it says that is not said in Theorem 2? Is Theorem 2 just a corollary of Theorem 1?
- The first sentence of the abstract is not clear. Overall, the abstract may need to be rewritted in a more professional manner, not referring to previous works as "believing" in something, but something more precise.

**Questions:**

Figure 2: I see a clear peak, but I do not see an initial “descent”. Is there something I am missing here? Why does the model have best generalization at $c = 0$? I am guessing that the focus of the study is on the peak, although that is not the full picture of double descent as I am not seeing the initial descent. That is okay, but just make clear why there is no initial descent.

**Limitations:**

Yes

---

> ### Author Rebuttal · Authors · 2024-08-05
>
> We thank the reviewer for recognizing the paper's good contextualization, its original viewpoint, and its significant result.
>
> We now address the reviewer's concerns.
>
> > Clarity: Although the contextualization relative to prior work is good, I find that the paper lacks in clarity. In particular, it is not exceptionally well written, and leaves some sections with much to be desired in terms of exposition. Specific examples are:
>
> We thank the reviewer for pointing out these concerns. We shall address them in the revised version of the paper.
>
> > Table 1: What does 1/Low mean? The superscript 3 leads to nowhere.
>
> Subscript 3 points to Appendices A and F, which present a conjectured formula for the low-rank case with experimental evidence. We shall update the table only to say 1, the rank considered in the main text.
>
> > Middle of page four: “Hence, this is controlled by 1. The alignment…, 2. The spectrum”. What exactly is being controlled here? And why exactly is it controlled by 1. and 2. ? This is not immediate to me, and I think this needs to be made more precise.
>
> We apologize there was a typo in the equation, it should read $\sum_{i=1}^r \frac{(y^T V)_I^2}{\sigma_i^2}$.
>
> This quantity depends on two things: the product $y^TV$, which we call the alignment, and $\sigma_i^2$, which is the spectrum.
>
> > Over-use of italics in the introduction makes it hard to know what to focus on. I would recommend maximum one italicized sentence per paragraph.
>
> We thank the reviewer for the feedback and shall reduce the number of italicized phrases.
>
> > Section 4.1: be more precise about the model, I don’t know what it is at this point. I later identify that $X + A$ represents the spiked data, but this must be made clear earlier on. I find“Let A be the noise matrix” to not be clear enough.
>
> We shall add a definition for the spiked model in section 4.1
>
> > Theorem 1: a comment on the proof technique for this theorem would be helpful, even if you have already mentioned it previously (or given the intuition).
>
> We shall add the following small proof sketch to the paper.
>
> *Sketch:* The main steps for the proof are as follows. First, we use the results from [49], which is like a Sherman Morrison formula but for pseudoinverses. Following this, we rewrite the error as a sum and product of various dependent quadratic forms. We then use ideas from random matrix theory and concentration of measure to show that each quadratic form concentrates on a deterministic number that depends on the Stieljtes transform of the limiting empirical spectral distribution. We then show that the product/sums of dependent forms also concentrate. This gives us the error rate as well.
>
> > Also, the interpretation of Theorem 1 is not clear until we read Theorem 2, could you interpret Theorem 1 a bit more and specifically identify what it says that is not said in Theorem 2?
>
> Theorem 1 provides the complete error curve as a function of $c$, not just where the maximum is. Prior works such as [24,48] have shown that noise on the independent variable acts as a regularizer. Hence, we have two regularizers for the problem - the noise and the ridge regularization. It is interesting to explore their tradeoffs. We do this in Appendix D. We shall expand on this in the main text.
>
> > Is Theorem 2 just a corollary of Theorem 1?
>
> Theorem 2 can be viewed as a corollary of Theorem 1. It follows by taking the leading terms in Theorem 1 and doing calculus (compute the first derivative to find the critical point and the second to check that it is a maximum).
>
> > The first sentence of the abstract is not clear. Overall, the abstract may need to be rewritted in a more professional manner, not referring to previous works as "believing" in something, but something more precise.
>
> We thank the reviewer for pointing out this concern and shall change the phrasing to make it clearer.
>
> > Figure 2: I see a clear peak, but I do not see an initial “descent”. Is there something I am missing here? Why does the model have best generalization at $c = 0$? I am guessing that the focus of the study is on the peak, although that is not the full picture of double descent as I am not seeing the initial descent. That is okay, but just make clear why there is no initial descent.
>
> The reviewer is correct that there is no initial descent. This is actually quite common for linear models. See Hastie et al (2020) for many such examples.
>
> Whether the error is minimum at $c = 0$ vs $c = \infty$ is quite interesting and related to benign overfitting. $c~0$ is the case when $n >> d$. Hence, we have a lot of data points. Hence, classically, we hope for consistent estimators. Hence, we can expect to do well.
>
> The $c ~\infty$ case is when $d >> n$. This is the largely overparameterized case. If the global minimum is here, this suggests that the model exhibits benign overfitting.
>
> For our examples, benign overfitting seems absent, and hence, the minimum is at $c = 0$.
>
> We hope this has addressed the reviewers' concerns and improved their opinion about our work. If the reviewer has further concerns, please let us know,

---

> > ### Comment · Reviewer_SLpr · 2024-08-10
> >
> > Thank you to the authors for addressing my questions, they have helped me better understand the paper.

---

### Official Review · Reviewer_RQj2 · 2024-07-11

**Soundness:** 3
**Presentation:** 3
**Contribution:** 3
**Rating:** 7
**Confidence:** 3

**Summary:**

The authors explore the double descent phenomenon, postulating that the location of the peak (that separates the "classical" and the "modern" interpolating regime) depends on the properties of the spectrum and the eigenvectors of the sample covariance. In particular, the authors show that the violation of one of two assumptions (assumption 1: Alignment of y and right singular vectors of X; assumption 2: Stieljtes Transform Peak Assumption) can move the peak from the interpolation point into the under-parameterized regime. They also present two simple examples that exhibit double descent in the under-parameterized regime and do not seem to occur for reasons provided in prior work.

**Strengths:**

- The paper tackles a very important research topic, and tries to understand the reasons behind the location of the peak in double descent.
- The work seems rigorous and the contributions relevant.
- Overall, the paper is well-written and reasonably clear.

**Weaknesses:**

- The conclusions of the paper are very brief and, from my point of view, not very informative (see section 6 in the paper). In relation to this, I also perceive a certain imbalance in the weight of the two examples provided: while the first (the one related to "Alignment Mismatch") occupies 3 pages of the work, the second example ("Shifting Local Maximum for Stieljtes Transform as a Function of c") is addressed more hastily (one page).
- The volume of information provided by the paper is very high. From this point of view, I think it would be positive to recapitulate and indicate clearly, and in a simple and intuitive way, the way in which the risk curves shown throughout the paper are created (Figures 2, 4 and 6). The same applies to Figure 3: what ablation experiments do the authors refer to?

**Questions:**

- What do the authors exactly mean by input and output noise?
- In Table 1 (page 3), where are the footnotes related with numbers 3 and 4?
- In Figure 3, what ablation experiments do the authors refer to?

**Limitations:**

- In my opinion, the authors do not sufficiently discuss the limitations of the work performed. In fact, in the NeurIPS Paper Checklist, they only state that "We believe the main purpose of the paper is to show that a certain phenomenon exists and are very careful with our assumptions."

---

> ### Author Rebuttal · Authors · 2024-08-04
>
> We thank the reviewer for finding that the paper tackles a very important research topic, is rigorous with relevant contributions, and is well written. We now address the reviewer's concerns.
>
> > The conclusions of the paper are very brief and, from my point of view, not very informative (see section 6 in the paper). In relation to this, I also perceive a certain imbalance in the weight of the two examples provided: while the first (the one related to "Alignment Mismatch") occupies 3 pages of the work, the second example ("Shifting Local Maximum for Stieljtes Transform as a Function of c") is addressed more hastily (one page).
>
> We agree that the conclusions are short. In the final version of the paper, we shall increase the length of the conclusions to make it more substantive while balancing the two examples. In particular, we will add the stronger version of Theorem 4 mentioned in the response to reviewer 4cXR.
>
> > The volume of information provided by the paper is very high. From this point of view, I think it would be positive to recapitulate and indicate clearly, and in a simple and intuitive way, the way in which the risk curves shown throughout the paper are created (Figures 2, 4 and 6). The same applies to Figure 3: what ablation experiments do the authors refer to?
>
> Thank you for the feedback. We shall do so.
>
> The ablation experiment is described on lines 212 to 218, where we break the misalignment in two ways.
>
> First, we consider the unregularized problem and artificially shift the spectrum of the noise matrix. This results in the shifted noise matrix having the same spectrum as the effective spectrum in the regularized problem. However, the alignment wasn't broken, so we see the peak at 1. Figure 3 (left).
>
> Second, for the regularized problem, we artificially fixed the alignment and noticed that the peak is now in the over-parameterized regime.
>
> Hence, the experiment validates the theory that the location of the peak was due to the misalignment.
>
> > What do the authors exactly mean by input and output noise?
>
> Consider a linear function $y = \beta^T x$. This is a function that we are trying to fit. However, if we had access to exactly the correct inputs $x$ and outputs $y$, then solving the problem would be easy, and we would never see a double descent. To see double descent, we need to introduce noise into the problem. This can be done in two ways.
>
> 1. Output noise. That is, we do not receive the responses $y$, but noisy versions, so $y = \beta^T x + noise$.
> 2. Input noise. Now, instead of getting noisy $y$ measurements. We get the true measurements $y$ but receive noisy inputs $x$.
>
> > In Table 1 (page 3), where are the footnotes related with numbers 3 and 4?
>
> We apologize for this.
>
> Footnote 3 points to Appendix A and F, which discuss the low-rank version of Theorem 1. Appendix A provides numerical experiments to verify the formula, and Appendix F has a statistical physics-type derivatization for the error.
>
> Footnote 4 states that [41] only considers optimal regularization.
>
> > In Figure 3, what ablation experiments do the authors refer to?
>
> The ablation experiment is described on lines 212 to 218, where we break the misalignment in two ways.
>
> First, we consider the unregularized problem and artificially shift the spectrum of the noise matrix. This results in the shifted noise matrix having the same spectrum as the effective spectrum in the regularized problem. However, the alignment wasn't broken, so we see the peak at 1. Figure 3 (left).
>
> We hope that this has addressed the concerns of the reviewers and improved their opinion about our work. If the reviewer has further concerns, please let us know,

---

> > ### Comment · Reviewer_RQj2 · 2024-08-09
> > **Score raised**
> >
> > Dear authors,
> >
> > I've read your responses to my comments, as well as your responses to all other reviewers' comments, and I've increased my score (moving from "6: Weak Accept" to "7: Accept"). I thank you for your detailed reply.
> >
> > Best

---

> > > ### Author Response · Authors · 2024-08-10
> > >
> > > Dear reviewer,
> > >
> > > Thank you for the feedback and help in improving the paper. We also thank the reviewer for increasing their score.

---

### Official Review · Reviewer_LfUR · 2024-07-11

**Soundness:** 3
**Presentation:** 2
**Contribution:** 2
**Rating:** 5
**Confidence:** 3

**Summary:**

In this paper, the authors focus on the generalization performance of linear least squares regression and show the existence of double descent generalization curve  in the under-parameterization regime.
In particular, the authors argue, in the linear model in (1) under study (which is slightly different from standard linear models in the literature, but well motivated), that the generalization risk can have a peak in the under-paramererized regime that is due to the alignment between singular space of data and the target, and spectrum of data covariance, instead of the raw dimension ratio or the explosion of the estimator norm.
Some numerical results are provided to support the theoretical analysis.

**Strengths:**

The problem under study is of significance.
The message of this paper looks interesting. But I find it a bit hard to really understand the results and contribution, see my comments below.

**Weaknesses:**

While this paper looks interesting, it is a bit hard for me to really understand the results and contribution.
I think making precise the dimension settings (relation between $n, d, n_{tst})$ will address this issue, see some of my detailed comments below.
Another issue is the contribution: while Theorem 1 is rather general, the discussion thereafter seems all special cases: For example Theorem 2 is a special case, and the results in Sec 4.3 and 4.5 are essentially numerical. The discussions in Sec 5 is interesting but again a very special setting (mixture model of multivariate Gaussian and a fixed direction) without any motivation. It is thus difficult for me to evaluate the significance of this work.

**Questions:**

* line 35: when summarizing the contribution of this work, it would be helpful to forward point to the corresponding theoretical result and/or definition, for the sake of a precise statement of the technical result or the definition (for example, the spiked covariate model).
* it seems that the footnotes 3 and 4 are missing?
* Equation after line 123: I am a bit confused here. What is the purpose here? Is $\hat \beta$ still the min norm solution, then what is $\beta$?
* To make Definition 1 more rigorous, perhaps say here that the convergence of ESD holds in a weak sense as $k \to \infty$, or something like that?
* Theorem 1: perhaps say somewhere in the theorem that this result holds in the asymptotic setting as $n,d,n_{tst}$ going to infinity at the same pace?
* Honestly, I do not understand this result. It seems to me that my previous comment is wrong, and that the result in Theorem 1 does NOT hold in the limit of $n,d,n_{tst} \to \infty$ together, or at least, $n_{tst}$ and $n$ can be much larger than $d$. Some specifications and discussions are needed here.
* Theorem 2 looks interesting. Could the authors comment more on this? For example, note that taking $\mu = 0$ is (more or less) similar to the ridgeless case in the literature. There, according to Theorem 2, we should have a at $c = 1$, as in accordance with "classical" double descent.  So, should we understand Theorem 2 as an extension of "classical" double descent to the regularized setting? Or is this due to the model in (1) and (2)?
* line 207 -208: $\hat \Sigma^T \hat \Sigma = \Sigma^T \Sigma + \mu^2$ a typo here?

---
I thank the authors for their detailed reply, which helps me better understand their theoretical results and their contribution I increase my score accordingly.

I, nonetheless, feel that there are a few typos that need be fixed and clarifications needed, in the current version of the paper.

**Limitations:**

I do not see any potential negative social impact of this work.

---

> ### Author Rebuttal · Authors · 2024-08-04
>
> We thank the reviewer for finding that the problem we study is significant and that the findings are interesting. We now address the reviewer's concerns.
>
> > Another issue is the contribution: while Theorem 1 is rather general ... It is thus difficult for me to evaluate the significance of this work.
>
> The paper's main purpose is to attempt to understand the reasons peaks occur in the excess risk curve and their locations. Prior work has phrased this as occurring due to overparameterization and the location being on the boundary of the under and overparameterized regimes. Additionally, some prior work [32] has criticized parameter counts as a measure of complexity, saying that parameter counts might overestimate the true complexity. Hence, shifting the boundary between under and over-parameterized to the **right, not the left**.
>
> We question this narrative. In particular, we show that the location of the peak is determined by technical reasons in random matrix theory **and not because it is the boundary between over and under-parameterized**. To show this, we present the two conditions in **Section 3 and, in particular, lines 145 to 148 highlight them again.** The rest of the paper then presents two examples where each example violates exactly one of the conditions, and we show that the peak moves to the left.
>
> In particular, our examples let us move the peak to the left and control its location.
>
> > Scaling of $d$, $n$, and $n_{tst}$
>
> This is a great question. Our results are valid for **any** $n$, $d$, and $n_{tst}$. However, they are only meaningful when $n$ and $d$ are large, and $d$ and $n$ are proportional. Note that we assume that $\sigma^2_{tst} = O(n_{tst})$. The last error is mostly independent of the scaling of $n_{tst}$.
>
> This can be seen as follows. The primary proof technique is concentration of measure. In particular, we want to show that the risk concentrates. Theorem 1 can be interpreted as saying when $c = d/n$ that (grouping the first and second term together)
>
> $$ \left|\mathcal{R}(c, \mu) - \frac{\sigma\_{trn}^2(\sigma\_{trn}^2 + 1)}{2d\tau^2} \left( \frac{c(1+c+\mu^2 c)}{\sqrt{(1-c+\mu^2 c)^2+4\mu^2 c^2}} - 1\right)  - \frac{\sigma\_{tst}^2}{\tau^2 n\_{tst}} \right| = o\left(\frac{1}{d}\right) = o\left(\frac{1}{n}\right) $$
>
> Hence, we show that the error concentrates around the above expression with an error of order $o\left(\frac{1}{d}\right)$. Hence if $d,n$ are small, the error might be large, however, if $d,n$ are large then the error will be small.
>
> To make sure this is meaningful, suppose $\sigma_{trn}^2 = \Theta(n) = \Theta(d)$ (we are assuming $n$ and $d$ are proportional) and $\sigma_{tst}^2 = \Theta(n_{tst})$. Then we see that $\tau^2 = \Theta(n^2) = \Theta(d^2)$. Then we see that the first term $\frac{\sigma_{trn}^2(\sigma_{trn}^2 + 1)}{2d\tau^2} \left( \frac{c(1+c+\mu^2 c)}{\sqrt{(1-c+\mu^2c)^2+4\mu^2c^2}} - 1\right)$ is $\Theta(1/d) = \Theta(1/n)$. The final term $\frac{\sigma_{tst}^2}{\tau^2 n_{tst}}$ of order $\Theta(1/d^2)$. Thus, the whole expression is of order $\Theta(1/d)$, while the error goes to zero faster.
>
> > line 35: when summarizing the contribution of this work, it would be helpful to forward point to the corresponding theoretical result and/or definition, for the sake of a precise statement of the technical result or the definition (for example, the spiked covariate model).
>
> Thank you for this suggestion. We had forward pointers to the Theorems; however, we shall do this for the definitions as well.
>
> > Footnote links
>
> We apologize for this.
>
> Footnote 3 points to Appendix A and F, which discuss the low-rank version of Theorem 1. Appendix A provides numerical experiments to verify the formula, and Appendix F has a statistical physics-type derivatization for the error.
>
> Footnote 4 states that [41] only considers optimal regularization.
>
> > Line 123
>
> Apologies, it should be $\hat{\beta}$ on line 123 and the equation should be $\frac{(y^TV)_i^2}{\sigma_i^2}$
>
> > Definition 1
>
> Thank you, we shall add the phrase.
>
> > Theorem 1: perhaps say somewhere in the theorem that this result holds ...
>
> and
>
> > Honestly, I do not understand this result...
>
> Please see our response on the scaling on $d$ and $n$. *It is imperative that we clarify any concerns about the theoretical results. Please let us know if something is still unclear.*
>
> > Theorem 2 looks interesting...
>
> Setting $\mu = 0$ exactly recovers the result from [24], so it is an extension of that result. It is not a direct extension of the results in, say, Dobriban and Wager, Hastie et al., or Bartlett et al. Those settings are different.
>
> In our setting we have low dimensional signal, the response depends on the signal and have noise on the inputs. In the setting of Dobriban and Wager, Hastie et al., or Bartlett et al., we have full dimensional signal and noise on the outputs. The regularized extension for these works can be seen in [11,23]. Here, we see double descent for certain $\mu$, and for the optimal $\mu$, we do not. **However, the peak does not move!**. This and more prior work are summarized in Table 1.
>
> In our work, the peak moves $\mu$, which is quite surprising. *Classical wisdom would states* that increasing $\mu$ would *increase* the regularization, hence would *decrease* the complexity of the model. Hence, we would need *a larger* number of parameters to overfit. Hence, the peak would move to the *right* in the overparameterized regime.
>
> However, we show that the peak moves to **left**!
>
> > Line 207-208
>
> Apologies, that is a typo. It should read $\hat{\Sigma}^T \hat{\Sigma} = \Sigma^T \Sigma + \mu^2 I$.
>
>
> We thank the reviewer again for their detailed comments. We hope that our response addressed all of the concerns. We would be eager to continue the conversation if there are any more concerns.
>
> We hope that the response has improved the reviewer's opinion on our work.

---

> > ### Comment · Reviewer_LfUR · 2024-08-09
> >
> > I thank the authors for their detailed reply, which helps me better understand their theoretical results and their contribution
> > I increase my score accordingly.
> > I, nonetheless, feel that there are a few typos that need be fixed and clarifications needed, in the current version of the paper.

---

> > > ### Author Response · Authors · 2024-08-10
> > >
> > > We thank the reviewer again for their feedback and will incorporate the same.

---

### Official Review · Reviewer_4cXR · 2024-07-13

**Soundness:** 3
**Presentation:** 3
**Contribution:** 2
**Rating:** 6
**Confidence:** 3

**Summary:**

The paper considers the problem of linear least squares. Its main contribution is presenting two examples of double descent in the under-parameterized regime.

**Strengths:**

- The paper is well written: Related works are sufficiently discussed (to my knowledge); the introduction is well-motivated and easy to follow; theorems are often followed by examples, figures, and illustrations helping the reader understand the results.
- The problem the paper investigates and the perspective the paper takes is quite interesting. While the mainstream research in the field focuses on double descent in the case of over-parameterization, the paper analyzes under-parametrization in-depth and presents several results that improve one's understanding of double descent.

**Weaknesses:**

- The paper takes unconventional notations that make the paper more challenging to penetrate. For example, it uses row vector notations and writes vector-matrix multiplication $\beta^\top X$ rather than the more common matrix-vector multiplication $X^\top \beta$. Sometimes I also found that the notations of singular vector $u$ and regularization parameter $\mu$ can be confusing as they look similar.
- The two examples the paper offered are indeed examples. The reason is that the paper's assumptions are quite strong. For example, Assumption 3 assumes the test and training data matrices are both of rank $1$, and Theorem 4 has the orthogonality assumption $\beta^\top z=0$ which greatly simplifies the model and analysis.

**Questions:**

I have no questions. It should be noted that I am not an expert in the exact area of double descent. I am not very familiar with the proof techniques used in the literature and am unable to make comments on technical depth.

**Limitations:**

See above.

---

> ### Author Rebuttal · Authors · 2024-08-04
>
> We thank the reviewer for finding our paper well-written and our perspective interesting. We now address the reviewer's concerns.
>
> > Notation
>
> We shall fix $\beta^T X$ to $X^T\beta$ to align the paper better with the prior convention. We shall change the font for the $u$ to clarify the difference.
>
> > Strong Assumptions
>
> We agree with the reviewer that some of our assumptions are strong. However, we believe that this helps us highlight the dependence of the location on the peak on 1) the alignment between $y$ and $V$ and 2) The peak in the Stieljtes transforms curve at 0.
>
> However, here we discuss some relaxations of the assumptions.
>
> ------------
>
> For rank 1 assumptions, we refer the reviewer to Appendix A and Appendix F, where we provide a conjectured formula for general rank $r$  data as well as numerical verification of the conjecture.
>
> -------------
>
> For the orthogonality assumption, this is primarily to simplify the proof. **Please note that in the numerical simulations displayed on the left of Figure 7, $\beta$ is not orthogonal to $z$.** This version of the result can be proved.
>
> We will update the paper with the result without assuming orthogonality. Here are the proof changes sketched out.
>
> If we do not make this assumption, Then at the bottom of page 19, we would have an additional term:
>
> $$ \frac{\|v\|^2}{1+\|v\|^2 z^T(AA^T)^{-1}z} \beta^T z z^T(AA^T)^{-1} $$
>
> Thus, on line 565, when we compute the excess risk, instead of
>
> $$\mathbb{E}\left[\left\|\left\|\xi^T \begin{bmatrix} A^T \\ vz^T \end{bmatrix} \left(AA^T + \|v\|zz^T\right)^{-1} \right\|\right\|^2 | X \right],$$
>
> we would have
>
> $$\mathbb{E}\left[\left\|\left\|\xi^T \begin{bmatrix} A^T \\ vz^T \end{bmatrix} \left(AA^T + \|v\|zz^T\right)^{-1} +  \frac{\|v\|^2}{1+\|v\|^2 x^T(AA^T)^{-1}z} \beta^T z z^T(AA^T)^{-1}\right\|\right\|^2 | X \right]$$
>
> Then, we can expand the term norm into parts and see that the cross terms are zero due to $\xi$ being independent from $z, A$ and having mean zero. The first term would be
>
> $$\mathbb{E}\left[\left\|\left\|\xi^T \begin{bmatrix} A^T \\ vz^T \end{bmatrix} \left(AA^T + \|v\|zz^T\right)^{-1} \right\|\right\|^2 | X \right],$$
>
> which have already shown how to compute the expectation. For the second term, to understand the contribution of the norm of this term, we would need to compute
>
> $$  \left(\frac{\|v\|^2}{1+\|v\|^2 z^T(AA^T)^{-1}z}\right)^2 \text{Tr}(z^T(AA^T)^{-2} z) $$
>
> Using the estimates from lines 568-570 and the same concentration results from [24], we see that this concentrates around
>
> $$ \left(\frac{\|v\|^2}{1+\|v\|^2\mathbb{E}\_{\lambda \sim \nu}\left[\frac{1}{\lambda}\right]}\right)^2 \mathbb{E}\_{\lambda \sim \nu}\left[\frac{1}{\lambda^2}\right] $$
>
> Then, since the covariance for the Gaussian part is $\frac{\pi_1}{d} I$. Multiplying, we get that this term goes to zero as $d \to \infty$. Hence, the error formula is unchanged.
>
> ------------
>
> We hope that this has addressed the concerns of the reviewers and improved their opinion about our work. If the reviewer has further concerns, please let us know.

---

> ### Comment · Reviewer_4cXR · 2024-08-11
> **Reply**
>
> Dear authors, thank you for your reply. It has nicely addressed my concerns. I have thus increased my score by 1.

---

### Official Review · Reviewer_4Q1q · 2024-07-13

**Soundness:** 3
**Presentation:** 3
**Contribution:** 2
**Rating:** 6
**Confidence:** 2

**Summary:**

The authors show several facts about the double descent phenomena for the linear regression model with L2 loss and Frobenius norm regularization. They show taking different assumption from previous work moves the peak of the risk of the problem from the interpolation point into the under-parameterized regime. Provided theorems concretely describe the position of peak risk, including separating the risk into several terms: bias and the norm of the estimator. The authors also provide derivations (appendix) and experiments.

**Strengths:**

Their analysis is thorough. Especially, reasoning change of trends of risks in Figure 4 through the term of the norm of the estimator supports the reliability of the results. They clarify proof steps in the appendix. Experiments provided to support each theorem also look thorough.

**Weaknesses:**

The authors results which do not coincide with prior theory are based on different assumptions. Can the authors discuss how wider cases covered by their assumption?

**Questions:**

Line 190: Is the assumption "$d$ is sufficiently large" used only to assume $o\left(1/d\right)=0$ in the Equation between Lines 177 & 178? If so, clarifying it in the main text would be better.



Does $\left\|W_{opt}\right\|_F$ in Line 240 indicate its expectation?



I recommend the authors to double check the overall text and equations. Here is a list of errata & typos.

Note that from the equation between Line 119 and Line 120, $\hat \beta\in\\mathbb{R}^{d\times k}$.

* Line 120: $\hat\beta = y X^\dagger$ -> $\hat \beta = \left(X^\dagger\right)^Ty^T$

* Displayed Equation between Lines 123-124: The denominator seems wrong. The following should be correct:

$$
\left\|\hat\beta\right\|^2 = \sum_{i=1}^{\mathrm{rank}\left(X\right)}\frac{\left(y V\right)_i^2}{\sigma_i^2}
$$

* Line 129: It would be better to add a couple of words emphasizing $\Sigma_n^{1/2}$ indicates the diagonal matrix $\Sigma$ in $X=U\Sigma V^T$ for readability. $\Sigma_n^{1/2}z_i$ looks like a summation symbol. I first thought it is a typo of $\sum_{i=1}^{1/2}z_i$.

* Displayed equation between Lines 137-138: $z\in \mathbb{C}\setminus J$ -> $\zeta\in \mathbb{C}\setminus J$

* Lines 239-240: Is "for $n$ large enough and $d=cn$" a typo of "for $n$ and large enough $d=cn$"?

**Limitations:**

Does the previous work (peak at interpolation point, using Assumptions 1 and 2) also rely on Assumptions 3 and 4? Or are you using just a different set (A 2 & 3 & 4) of assumptions which is a neither necessary nor sufficient condition of one of the previous work (A 1 & 2)? It should be stated more clearly.

---

> ### Author Rebuttal · Authors · 2024-08-04
>
> We thank the reviewer for finding our analysis thorough, especially the reasoning for the trends. We now address the reviewer's concerns.
>
> > The authors results which do not coincide with prior theory are based on different assumptions. Can the authors discuss how wider cases covered by their assumption?
>
> We believe that much prior work assumes Assumptions A1 (alignment between $y$ and $V$) and A2 (Stieljtes Transform Peak). **In our paper, we want to construct examples that violate these two assumptions.** Hence, the example in Section 4 makes assumptions A3 and A4 (not A1 and A2), while the example in Section 5 is a mixture of a high-dimensional Gaussian and a one-dimensional Gaussian.
>
> Assumptions A3 and A4 exist in prior work such as [24,37]. These works do have a peak at $c=1$. In our paper, setting $\mu = 0$ recovers [24]. Appendix A, F has the corresponding result for which setting $\mu = 0$ recovers [37].
>
> The mixture example can be viewed as an extension of prior theory presented in papers such as Dobriban and Wager, and Hastie et al., by taking data distributions that satisfy their assumptions and creating a mixture with a distribution supported on a lower dimensional set. Setting $\pi_1 = 1$ (i.e., removing the low dimensional component) recovers an example that satisfies their assumptions
>
> Finally, we would like to highlight some prominent examples that explicitly assume A1 and A2 or their assumptions imply A1 and A2 are satisfied.
>
> **Dobriban and Wager, 2015**
>
> *A1*
>
> Assume that $\beta$ (and hence $y$) has an isotropic distribution (Assumption B in the arxiv version). Hence, $y^T V$ has an isotropic distribution. Hence, they assume A1.
>
> *A2*
>
> The paper assumes assumption A and that the operator norm of the expected sample covariance is uniformly bounded. These assumptions then satisfy the assumptions of Theorem 1.1 in [BZ08], which implies that the Stieljtes transform of the limiting distribution satisfies
>
> $$ m(z) = \int \frac{1}{t(1-c-czm) - z} dH(t), $$
>
> where $H$ is the limiting distribution of the population covariance matrix, our assumption A2 is concerned about the function $c \mapsto m(0)$. Here, we see that
>
> $$ m(0) = \int \frac{1}{t(1-c)} dH(t) $$
>
> This function is clearly maximum when $c = 1$. Hence, it satisfies assumption A2.
>
> **Hastie, Montanari, Rosset, Tibshirani, 2020**
>
> *A1*
>
> They assume that data $x_i \sim \mathcal{N}(0,\Sigma)$. If the $x_i$ are the columns of $X = U\Sigma V$ (as is the notation in our paper). Then $XX^T/n = U\Sigma\Sigma^T U^T / n$ is the sample covariance. Hence, it is not dependent on $V$. Hence, this implies that the distribution of $V$ doesn't impact the sample covariance matrix. Hence, we believe this is similar to assuming that $V$ is uniform. Hence, assuming $y^T V $ is istropic.
>
> *A2*
>
> Similar to the Dobriban paper, their data satisfies the assumption of Theorem 1.1 in [BZ08].
>
> [BZ08] Zhidong Bai and Wang Zhou. Large sample covariance matrices without independence structures in columns.
>
> > Line 190: Is the assumption " is sufficiently large" used only to assume $o(1/d) = 0$ in the Equation between Lines 177 & 178? If so, clarifying it in the main text would be better.
>
> The reviewer is right. We shall clarify this.
>
> > Does $\|W_{opt}\|$ in Line 240 indicate its expectation?
>
> Yes, thank for yo pointing this out, we shall correct this.
>
> > I recommend the authors to double check the overall text and equations. Here is a list of errata & typos.
>
> We thank the reviewer for the typos found. We shall correct them.
>
> > Does the previous work (peak at interpolation point, using Assumptions 1 and 2) also rely on Assumptions 3 and 4? Or are you using just a different set (A 2 & 3 & 4) of assumptions which is a neither necessary nor sufficient condition of one of the previous work (A 1 & 2)? It should be stated more clearly.
>
> Some prior work assumes A3 and A4 (such as [24,37], setting $\mu = 0$ in our result recovers those results); however, other prior work does not. The key fact is that A1 and A2 are true in prior work in these prior works.
>
> ---------
>
> We hope that we have addressed all of the reviewers' concerns and questions and improved their opinion of our work. If the reviewer has further concerns, please let us know.

---

> > ### Comment · Reviewer_4Q1q · 2024-08-14
> >
> > Thank you for the detailed response to the questions. It help me to understand this work more. I have been increased my score by 1.

---

### Author Rebuttal · Authors · 2024-08-06

We thank the reviewers for their time, effort, and feedback.

The paper's main purpose is to attempt to understand the reasons peaks occur in the excess risk curve and their locations. Prior work has phrased this as occurring due to overparameterization and the location being on the boundary of the under and overparameterized regimes. Additionally, some prior work [32] has criticized parameter counts as a measure of complexity, saying that parameter counts might overestimate the true complexity. Hence, shifting the boundary between under and over-parameterized to the right, not the left.

We question this narrative. In particular, we show that the location of the peak is determined by technical reasons in random matrix theory and not because it is the boundary between over and under-parameterized. To show this, we present the two conditions. The rest of the paper then presents two examples where each example violates exactly one of the conditions, and we show that the peak moves to the left.

We would also like to summarize the strengths of the paper as highlighted by the reviewers. We address the concerns of the reviewers in the individual responses.

1. Paper studies as Important Problem. \
    a. Reviewer LfUR - "The problem under study is of significance."\
    b. Reviewer RQj2 - "The paper tackles a very important research topic, and tries to understand the reasons behind the location of the peak in double descent."\
    c. Reviewer SLpr - "addresses the prominent double-descent phenomenon in machine learning, which is an important primitive for understanding generalization and other properties of estimators operating on high-dimensional data"



2. The persecpective of the paper is new, interesting, and rigorous. \
    a. Reviewer 4Q1q - "Their analysis is thorough.", "Experiments provided to support each theorem also look thorough."\
    b. Reviewer 4cXR - "The problem the paper investigates and the perspective the paper takes is quite interesting.", " presents several results that improve one's understanding of double descent."\
    c. Reviewer RQj2 - "The work seems rigorous and the contributions relevant."\
    d. Reviewer SLpr - "The paper considers an original viewpoint on the problem of double-descent", "This result is significant"

We hope that our responses address the reviewers concerns. If there are concerns that have not been addressed or need further discussion please let us know.

---

### Decision · Program_Chairs · 2024-09-25

**Decision:**

Accept (poster)

**Comment:**

The paper investigates the double descent phenomenon in linear regression. Specifically, the authors present two examples of double descent in the under-parameterized regime. The reviewers appreciated the contribution, found the paper well-written and clear, and liked how it positions itself with respect to previous work. Therefore, I recommend acceptance.